# Deep learning of contagion dynamics on complex networks

Charles Murphy [1,2], Edward Laurence[1,2] & Antoine Allard [1,2✉]

Forecasting the evolution of contagion dynamics is still an open problem to which mechanistic models only offer a partial answer. To remain mathematically or computationally tractable, these models must rely on simplifying assumptions, thereby limiting the quantitative accuracy of their predictions and the complexity of the dynamics they can model. Here, we propose a complementary approach based on deep learning where effective local mechanisms governing a dynamic on a network are learned from time series data. Our graph neural network architecture makes very few assumptions about the dynamics, and we demonstrate its accuracy using different contagion dynamics of increasing complexity. By allowing simulations on arbitrary network structures, our approach makes it possible to explore the properties of the learned dynamics beyond the training data. Finally, we illustrate the applicability of our approach using real data of the COVID-19 outbreak in Spain. Our results demonstrate how deep learning offers a new and complementary perspective to build effective models of contagion dynamics on networks.

[1] Département de physique, de génie physique et d'optique, Université Laval, Québec, Québec, Canada. [2] Centre interdisciplinaire en modélisation mathématique, Université Laval, Québec, Québec, Canada. ✉email: antoine.allard@phy.ulaval.ca

Our capacity to prevent or contain outbreaks of infectious diseases is directly linked to our ability to accurately model contagion dynamics. Since the seminal work of Kermack and McKendrick almost a century ago[1], a variety of models incorporating ever more sophisticated contagion mechanisms has been proposed[2–5]. These mechanistic models have provided invaluable insights about how infectious diseases spread, and have thereby contributed to the design of better public health policies. However, several challenges remain unresolved, which call for contributions from new modeling approaches[6–8].

For instance, many complex contagion processes involve the nontrivial interaction of several pathogens[9–12], and some social contagion phenomena, like the spread of misinformation, require to go beyond pairwise interactions between individuals[13–15]. Also, while qualitatively informative, the forecasts of most mechanistic models lack quantitative accuracy[16]. Indeed, most models are constructed from a handful of mechanisms which can hardly reproduce the intricacies of real complex contagion dynamics. One approach to these challenges is to complexify the models by adding more detailed and sophisticated mechanisms. However, mechanistic models become rapidly intractable as new mechanisms are added. Moreover, models with higher complexity require the specification of a large number of parameters whose values can be difficult to infer from limited data.

There has been a recent gain of interest towards using machine learning to address the issue of the often-limiting complexity of mechanistic models[12,17–23]. This new kind of approach aims at training predictive models directly from observational time series data. These data-driven models are then used for various tasks such as making accurate predictions[19,21], gaining useful intuitions about complex phenomena[12] and discovering new patterns from which better mechanisms can be designed[17,18]. Although these approaches were originally designed for regularly structured data, this new paradigm is now being applied to epidemics spreading on networked systems[24,25], and more generally to dynamical systems[26–28]. Meanwhile, the machine learning community has dedicated a considerable amount of attention to deep learning on networks, structure learning and graph neural networks (GNN)[29–31]. Recent works showed great promise for GNN in the context of community detection[32], link prediction[33], network inference[34], as well as for the discovery of new materials and drugs[35,36]. Yet, others have pointed out the inherent limitations of a majority of GNN architectures in distinguishing certain network structures[31], in turn limiting their learning capabilities. Hence, while recent advances and results[37–40] suggest that GNN could be prime candidates for building effective data-driven dynamical models on networks, it remains to be shown if, how and when GNN can be applied to dynamics learning problems.

In this paper, we show how GNN, usually used for structure learning, can also be used to model contagion dynamics on complex networks. Our contribution is threefold. First, we design a training procedure and an appropriate GNN architecture capable of representing a wide range of dynamics with very few assumptions. Second, we demonstrate the validity of our approach using various contagion dynamics of increasing complexity on networks of different natures, as well as real epidemiological data. Finally, we show how our approach can provide predictions for previously unseen network structures, therefore allowing the exploration of the properties of the learned dynamics beyond the training data. Our work generalizes the idea of constructing dynamical models from regularly structured data to arbitrary network structures, and suggests that our approach could be accurately extended to many other classes of dynamical processes.

## Results

In our approach, we assume that an unknown dynamical process, denoted $\mathcal{M}$, takes place on a known network structure—or ensemble of networks—, denoted $G = (\mathcal{V}, \mathcal{E}; \boldsymbol{\Phi}, \boldsymbol{\Omega})$, where $\mathcal{V} = \{v_1, \cdots, v_N\}$ is the node set and $\mathcal{E} = \{e_{ij} | v_j$ is connected to $v_i \wedge (v_i, v_j) \in \mathcal{V}^2\}$ is the edge set. We also assume that the network(s) contains metadata, taking the form of node and edge attributes denoted $\Phi_i = (\phi_1(v_i), \cdots, \phi_Q(v_i))$ for node $v_i$ and $\Omega_{ij} = (\omega_1(e_{ij}), \cdots, \omega_P(e_{ij}))$ for edge $e_{ij}$, respectively, where $\phi_q : \mathcal{V} \to \mathbb{R}$ and $\omega_p : \mathcal{E} \to \mathbb{R}$. These metadata can take various forms like node characteristics or edge weights. We also denote the node and edge attribute matrices $\boldsymbol{\Phi} = (\Phi_i | v_i \in \mathcal{V})$ and $\boldsymbol{\Omega} = (\Omega_{ij} | e_{ij} \in \mathcal{E})$, respectively.

Next, we assume that the dynamics $\mathcal{M}$ has generated a time series $D$ on the network $G$. This time series takes the form of a pair of consecutive snapshots $D = (\boldsymbol{X}, \boldsymbol{Y})$ with $\boldsymbol{X} = (X_1, \cdots, X_T)$ and $\boldsymbol{Y} = (Y_1, \cdots, Y_T)$, where $X_t \in \mathcal{S}^{|\mathcal{V}|}$ is the state of the nodes at time $t$, $Y_t \in \mathcal{R}^{|\mathcal{V}|}$ is the outcome of $\mathcal{M}$ defined as

$$Y_t = \mathcal{M}(X_t, G) , \tag{1}$$

$\mathcal{S}$ is the set of possible node states, and $\mathcal{R}$ is the set of possible node outcomes. This way of defining $D$ allows us to formally concatenate multiple realizations of the dynamics in a single dataset. Additionally, the elements $x_i(t) = (X_t)_i$ and $y_i(t) = (Y_t)_i$ correspond to the state of node $v_i$ at time $t$ and its outcome, respectively. Typically, we consider that the outcome $y_i(t)$ is simply the state of node $v_i$ after transitioning from state $x_i(t)$. In this case, we have $\mathcal{S} = \mathcal{R}$ and $x_i(t + \Delta t) = y_i(t)$ where $\Delta t$ is the length of the time steps. However, if $\mathcal{S}$ is a discrete set—i.e. finite and countable—$y_i(t)$ is a transition probability vector conditioned on $x_i(t)$ from which the following state, $x_i(t + \Delta t)$, will be sampled. The element $(y_i(t))_m$ corresponds to the probability that node $v_i$ evolves to state $m \in \mathcal{S}$ given that it was previously in state $x_i(t)$—i.e. $\mathcal{R} = [0, 1]^{|\mathcal{S}|}$. When $\mathcal{M}$ is a stochastic dynamics, we do not typically have access to the transition probabilities $y_i(t)$ directly, but rather to the observed outcome state—e.g. $x_i(t + \Delta t)$ in the event where $\boldsymbol{X}$ is temporally ordered—, we therefore define the observed outcome $\tilde{y}_i(t)$ as

$$(\tilde{y}_i(t))_m = \delta(x_i(t + \Delta t), m) , \quad \forall m \in \mathcal{S} \tag{2}$$

where $\delta(x, y)$ is the Kronecker delta. Finally, we assume that $\mathcal{M}$ acts on $X_t$ locally and identically at all times, according to the structure of $G$. In other words, knowing the state $x_i$ as well as the states of all the neighbors of $v_i$, the outcome $y_i$ is computed using a time independent function $f$ identical for all nodes

$$y_i = f\left(x_i, \Phi_i, x_{\mathcal{N}_i}, \Phi_{\mathcal{N}_i}, \Omega_{i\mathcal{N}_i}\right) , \tag{3}$$

where $x_{\mathcal{N}_i} = (x_j | v_j \in \mathcal{N}_i)$ denotes the states of the neighbors, $\mathcal{N}_i = \{v_j | e_{ij} \in \mathcal{E}\}$ is the set of the neighbors, $\Phi_{\mathcal{N}_i} = \{\Phi_j | v_j \in \mathcal{N}_i\}$ and $\Omega_{i\mathcal{N}_i} = \{\Omega_{ij} | v_j \in \mathcal{N}_i\}$. As a result, we impose a notion of locality where the underlying dynamics is time invariant and invariant under the permutation of the node labels in $G$, under the assumption that the node and edge attributes are left invariant.

Our objective is to build a model $\hat{\mathcal{M}}$, parametrized by a GNN with a set of tunable parameters $\boldsymbol{\Theta}$ and trained on the observed dataset $D$ to mimic $\mathcal{M}$ given $G$, such that

$$\hat{\mathcal{M}}(X'_t, G'; \boldsymbol{\Theta}) \approx \mathcal{M}(X'_t, G') , \tag{4}$$

for all states $X'_t$ and all networks $G'$. The architecture of $\hat{\mathcal{M}}$, detailed in Section "Graph neural network and training details", is designed to act locally similarly to $\mathcal{M}$. In this case, the locality is imposed by a modified attention mechanism inspired by ref. [41].

The advantage of imposing locality allows our architecture to be *inductive*: If the GNN is trained on a wide range of local structures—i.e. nodes with different neighborhood sizes (or degrees) and states—it can then be used on any other networks within that range. This suggests that the topology of $G$ will have a strong impact on the quality of the trained models, an intuition that is confirmed below. Simiarly to Eq. (3), we can write each individual node outcome computed by the GNN using a function $\hat{f}$ such that

$$\hat{y}_i = \hat{f}\left(x_i, \Phi_i, x_{\mathcal{N}_i}, \Phi_{\mathcal{N}_i}, \Omega_{i\mathcal{N}_i}; \Theta\right) \qquad (5)$$

where $\hat{y}_i$ is the outcome of node $v_i$ predicted by $\hat{\mathcal{M}}$.

The objective described by Eq. (4) must be encoded into a global loss function, denoted $\mathcal{L}(\Theta)$. Like the outcome functions, $\mathcal{L}(\Theta)$ can be decomposed locally, where the local losses of each node $L(y_i, \hat{y}_i)$ are arithmetically averaged over all possible node inputs $(x_i, \Phi_i, x_{\mathcal{N}_i}, \Phi_{\mathcal{N}_i}, \Omega_{i\mathcal{N}_i})$, where $y_i$ and $\hat{y}_i$ are given by Eqs. (3) and (5), respectively. By using an arithmetic mean to the evaluation of $\mathcal{L}(\Theta)$, we assume that the node inputs are equally important and uniformly distributed. Consequently, the model should be trained equally well on all of them. This consideration is critical because in practice we only have access to a finite number of inputs in $D$ and $G$, for which the node input distribution, denoted $\rho(k_i, x_i, \Phi_i, x_{\mathcal{N}_i}, \Phi_{\mathcal{N}_i}, \Omega_{i\mathcal{N}_i})$, is typically far from being uniform. Hence, in order to train effective models, we recalibrate the inputs using the following global loss

$$\mathcal{L}(\Theta) = \sum_{t \in \mathcal{T}'} \sum_{v_i \in \mathcal{V}'(t)} \frac{w_i(t)}{Z'} L\left(y_i(t), \hat{y}_i(t)\right) \qquad (6)$$

where $w_i(t)$ is a weight assigned to node $v_i$ at time $t$, and $Z' = \sum_{t \in \mathcal{T}'} \sum_{v_i \in \mathcal{V}'(t)} w_i(t)$ is a normalization factor. Here, the training node set $\mathcal{V}'(t) \subseteq \mathcal{V}$ and the training time set $\mathcal{T}' \subseteq [1, T]$ allow us to partition the training dataset for validation and testing when required.

The choice of weights needs to reflect the importance of each node at each time. Because we wish to lower the influence of overrepresented inputs and increase that of rare inputs, a sound choice of weights is

$$w_i(t) \propto \rho\left(k_i, x_i, \Phi_i, x_{\mathcal{N}_i}, \Phi_{\mathcal{N}_i}, \Omega_{i\mathcal{N}_i}\right)^{-\lambda} \qquad (7)$$

where $k_i$ is the degree of node $v_i$ in $G$, and $0 \leq \lambda \leq 1$ is an hyperparameter. Equation (7) is an ideal choice, because it corresponds to a principled importance sampling approximation of Eq. (6)[42], which is relaxed via the exponent $\lambda$. We obtain a pure importance sampling scheme when $\lambda = 1$. Note that the weights can rarely be exactly computed using Eq. (7), because the distribution $\rho$ is typically computationally intensive to obtain from data, especially for continuous $\mathcal{S}$ with metadata. We illustrate various ways to evaluate the weights in Sec. "Importance weights" and Sec. I in the Supplementary Information.

We now illustrate the accuracy of our approach by applying it to four types of synthetic dynamics of various natures (see Sec. "Dynamics" for details on the dynamics). We first consider a *simple contagion* dynamics: The discrete-time susceptible-infected-susceptible (SIS) dynamics. In this dynamics, nodes are either susceptible ($S$) or infected ($I$) by some disease, i.e. $\mathcal{S} = \{S, I\} = \{0, 1\}$, and transition between each state stochastically according to an infection probability function $\alpha(\ell)$, where $\ell$ is the number of infected neighbors of a node, and a constant recovery probability $\beta$. A notable feature of simple contagion dynamics is that susceptible nodes get infected by the disease through their infected neighbors independently. This reflects the assumption that disease transmission behaves identically whether a person has a large number of infected neighbors or not.

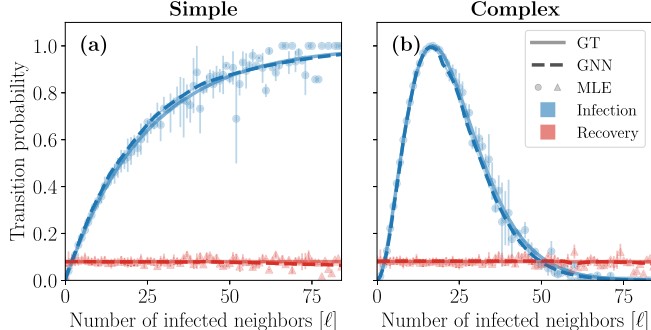

**Fig. 1 Predictions of GNN trained on a Barabási-Albert random network [72] (BA). a** Transition probabilities of the simple contagion dynamics. **b** Transition probabilities of the complex contagion dynamics. The solid and dashed lines correspond to the transition probabilities of the dynamics used to generate the training data (labeled GT for "ground truth"), and predicted by the GNN, respectively. Symbols correspond to the maximum likelihood estimation (MLE) of the transition probabilities computed from the dataset $D$. The colors indicate the type of transition: infection ($S \rightarrow I$) in blue and recovery ($I \rightarrow S$) in red. The standard deviations, as a result of averaging the outcomes given $\ell$, are shown using a colored area around the lines (typically narrower than the width of the lines) and using vertical bars for the symbols.

Second, we relax this assumption by considering a *complex contagion* dynamics with a nonmonotonic infection function $\alpha(\ell)$ where the aforementioned transmission events are no longer independent[14]. This contagion dynamics has an interesting interpretation in the context of the propagation of a social behavior, where the local popularity of a behavior (large $\ell$) hinders its adoption. The independent transmission assumption can also be lifted when multiple diseases are interacting[10]. Thus, we also consider an asymmetric *interacting contagion* dynamics with two diseases. In this case, $\mathcal{S} = \{S_1S_2, I_1S_2, S_1I_2, I_1I_2\} = \{0, 1, 2, 3\}$ where $U_1V_2$ corresponds to a state where a node is in state $U$ with respect to the first disease and in state $V$ with respect to the second disease. The interaction between the diseases happens via a coupling that is active only when a node is infected by at least one disease, otherwise it behaves identically to the simple contagion dynamics. This coupling may increase or decrease the virulence of the other disease.

Whereas the previously presented dynamics capture various features of contagion phenomena, real datasets containing this level of detail about the interactions among individuals are rare[43–45]. A class of dynamics for which dataset are easier to find is that of mass-action *metapopulation* dynamics[46–49], where the status of the individuals are gathered by geographical regions. These dynamics typically evolve on the weighted networks of the individuals' mobility between regions and the state of a region consists in the number of people that are in each individual health state. As a fourth case study, we consider a type of deterministic metapopulation dynamics where the population size is constant and where people can either be susceptible ($S$), infected ($I$) or recovered from the disease ($R$). As a result, we define the state of the node as three-dimensional vectors specifying the fraction of people in each state—i.e. $\mathcal{S} = \mathcal{R} = [0, 1]^3$.

Figure 1 shows the GNN predictions for the infection and recovery probabilities of the simple and complex contagion dynamics as a function of the number of infected neighbors $\ell$. We then compare these probabilities with their ground truths, i.e. Eq. (20) using Eqs. (18)–(22) for the infection functions. We also show the maximum likelihood estimators (MLE) of the transition probabilities computed from the fraction of nodes in state $x$ and

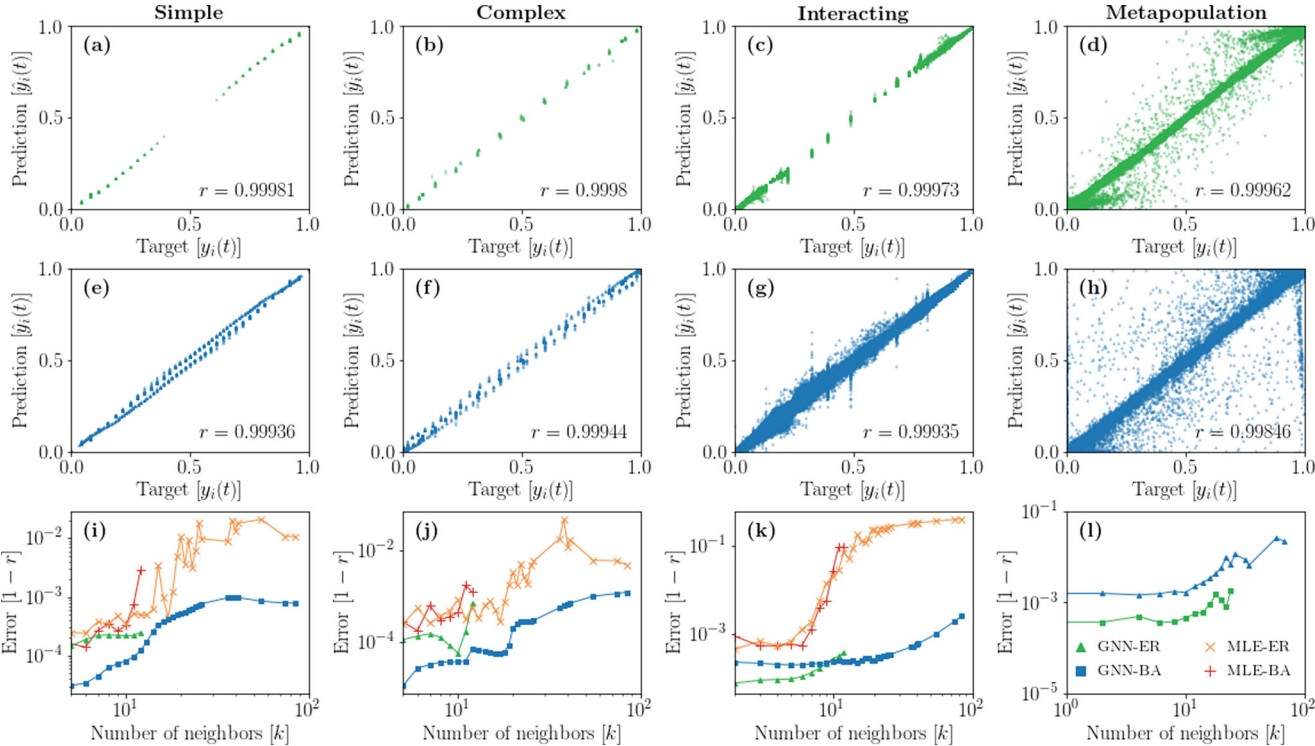

**Fig. 2 GNN learning performance on different dynamics and structures. a**, **b**, **c**, **d** GNN trained on Erdős–Rényi networks (ER). **e**, **f**, **g**, **h** GNN trained on Barabási-Albert networks (BA). **i**, **j**, **k**, **l** Error as a function of the number of neighbors. Each point shown on the panels **a–h** corresponds to a different pair $(y_i(t), \hat{y}_i(t))$ in the complete dataset $D$. We also indicate the Pearson coefficient $r$ on each panel to measure the correlation between the predictions and the targets and use it as a global performance measure. The panels **i–l** show the errors $(1 − r)$ as a function of the number of neighbors for GNN trained on ER and BA networks, and those of the corresponding MLE. These errors are obtained from the Pearson coefficients computed from subsets of the prediction-target pairs where all nodes have degree $k$.

with $\ell$ infected neighbors that transitioned to state $y$ in the complete dataset $D$. The MLE, which are typically used in this kind of inference problem[50], stands as a reference to benchmark the performance of our approach.

We find that the GNN learns remarkably well the transition probabilities of the simple and complex contagion dynamics. In fact, the predictions of the GNN seem to be systematically smoother than the ones provided by the MLE. This is because the MLE is computed for each individual pair $(x, \ell)$ from disjoint subsets of the training dataset. This implies that a large number of samples of each pair $(x, \ell)$ is needed for the MLE to be accurate; a condition rarely met in realistic settings, especially for high degree nodes. This also means that the MLE cannot be used directly to interpolate beyond the pairs $(x, \ell)$ present in the training dataset, in sharp contrast with the GNN which, by definition, can interpolate within the dataset $D$. Furthermore, all of its parameters are hierarchically involved during training, meaning that the GNN benefits from any sample to improve all of its predictions, which are then smoother and more consistent. Further still, we found that not all GNN architectures can reproduce the level of accuracy obtained in Fig. 1 (see Sec. III F of the Supplementary Information). In fact, we showed that many standard GNN aggregation mechanisms are ineffective at learning the simple and complex contagion dynamics, most likely because they were specifically designed with structure learning in mind rather than dynamics learning.

It is worth mentioning that the GNN is not specifically designed nor trained to compute the transition probabilities as a function of a single variable, namely the number of infected $\ell$. In reality, the GNN computes its outcome from the complete

multivariate state of the neighbors of a node. The interacting contagion and the metapopulation dynamics, unlike the simple and complex contagions, are examples of such multivariate cases. Their outcome is thus harder to visualize in a representation similar to Fig. 1. Figure 2a–h address this issue by comparing each of the GNN predictions $\hat{y}_i(t)$ with its corresponding target $y_i(t)$ in the dataset $D$. We quantify the global performance of the models in different scenarios, for the different dynamics and underlying network structures, using the Pearson correlation coefficient $r$ between the predictions and targets (see Sec. "Graph neural network and training details"). We also compute the error, defined from the Pearson coefficient as $1 − r$ for each degree class $k$ (i.e. between the predictions and targets of only the nodes of degree $k$). This allows us to quantify the GNN performance for every local structure.

Figures 2i–k confirm that the GNN provides more accurate predictions than the MLE in general and across all degrees. This is especially true in the case of the interacting contagion, where the accuracy of the MLE seems to deteriorate rapidly for large degree nodes. This is a consequence of how scarce the inputs are for this dynamics compared to both the simple and complex contagion dynamics for training datasets of the same size, and of how fast the size of the set of possible inputs scales, thereby quickly rendering MLE completely ineffective for small training datasets. The GNN, on the other hand, is less affected by the scarcity of the data, since any sample improves its global performance, as discussed above.

Figure 2 also exposes the crucial role of the network $G$ on which the dynamics evolves in the global performance of the GNN. Namely, the heterogeneous degree distributions of

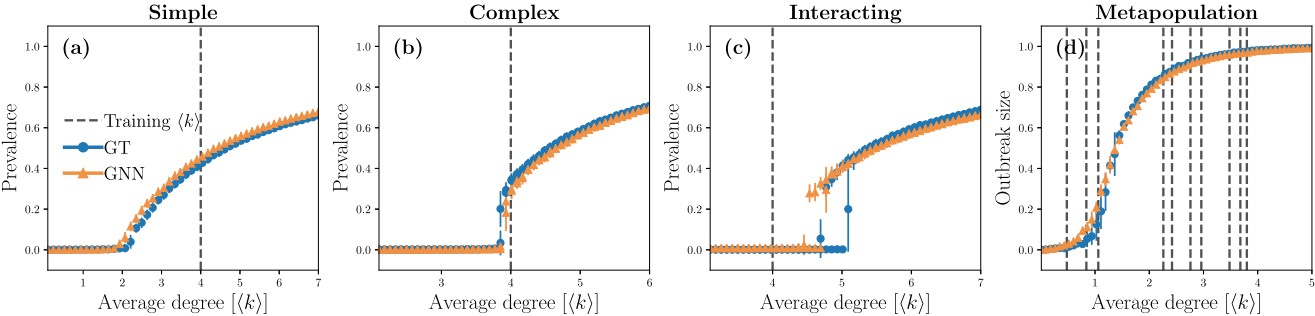

**Fig. 3 Bifurcation diagrams of the trained GNN. a** Simple contagion dynamics. **b** Complex contagion dynamics. **c** Interacting contagion dynamics. **d** Metapopulation dynamics. In these experiments, we used Poisson networks composed of $|\mathcal{V}| = 2000$ nodes with different average degrees $\langle k \rangle$. The prevalence is defined as the average fraction of nodes that are asymptotically infected by at least one disease and the outbreak size corresponds to the average fraction of nodes that have recovered. These quantities are obtained from numerical simulations using the "ground truth" (GT) dynamics (blue circles) and the GNN trained on Barabási-Albert networks (orange triangles). The error bars correspond to the standard deviations of these numerical simulations. The trained GNN used are the same ones as those used for Fig. 2. As a reference, we also indicate with dashed lines the value(s) of average degree $\langle k \rangle$ corresponding to the network(s) on which the GNN were trained. On **d**, more than one value of $\langle k \rangle$ appear as multiple networks with different average degrees were used to train the GNN (see Sec. "Graph neural networks and training details").

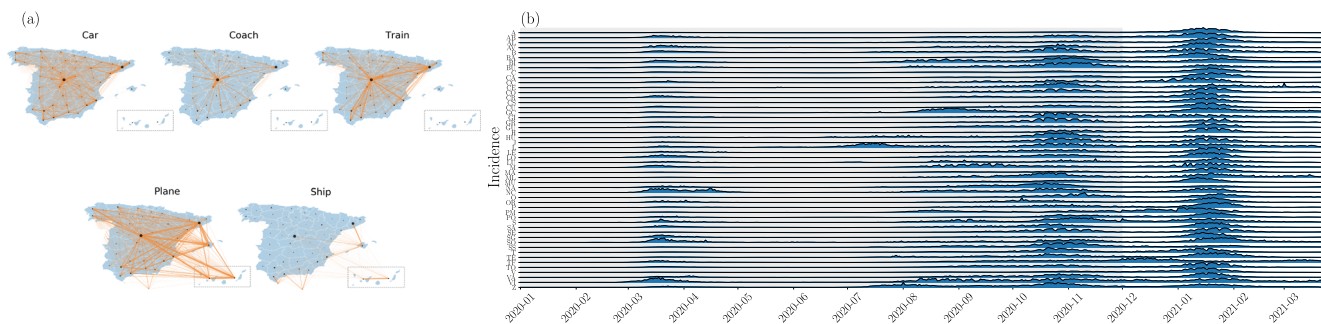

**Fig. 4 Spain COVID-19 dataset. a** Spain mobility multiplex network[54]. The thickness of the edges is proportional to the average number of people transitioning between all connected node pairs. The size of the nodes is proportional to the population $N_i$ living in the province. **b** Time series of the incidence for the 52 provinces of Spain between January 2020 and March 2021[53]. Each province is identified by its corresponding ISO code. Each incidence time series has been rescaled by its maximum value for the purpose of visualization. The shaded area indicates the training and validation datasets (in-sample) from January 1st 2020 to December 1st 2021. The remaining dataset is used for testing.

Barabási-Albert networks (BA)—or any heterogeneous degree distribution—offer a wider range of degrees than those of homogeneous Erdős-Rényi networks (ER). We can take advantage of this heterogeneity to train GNN models that generalize well across a larger range of local structures, as seen in Fig. 2i–l (see also Sec. III of the Supplementary Information). However, the predictions on BA networks are not systematically always better for low degrees than those on ER networks, as seen in the interacting and metapopulation cases. This nonetheless suggests a wide applicability of our approach for real complex systems, whose underlying network structures recurrently exhibit a heterogeneous degree distribution[51].

We now test the trained GNN on unseen network structures by recovering the bifurcation diagrams of the four dynamics. In the infinite-size limit $|\mathcal{V}| \to \infty$, these dynamics have two possible long-term outcomes: the absorbing state where the diseases quickly die out, and the endemic/epidemic state in which a macroscopic fraction of nodes remains (endemic) or has been infected over time (epidemic)[4,10,52]. These possible long-term outcomes exchange stability during a phase transition which is continuous for the simple contagion and metapopulation dynamics, and discontinuous for the complex and interacting contagion dynamics. The position of the phase transition depends on the parameters of the dynamics as well as on the topology of the network. Note that for the interacting contagion dynamics, the stability of absorbing and endemic states do not change at the

same point, giving rise to a bistable regime where both states are stable.

Figure 3 shows the different bifurcation diagrams obtained by performing numerical simulations with the trained GNN models [using Eq. (5)] while varying the average degree of networks, on which the GNN has not been trained. Quantitatively, the predictions are again strikingly accurate—essentially perfect for the simple and complex contagion dynamics—which is remarkable given that the bifurcation diagrams were obtained on networks the GNN had never seen before. These results illustrate how insights can be gained about the underlying process concerning the existence of phase transitions and their order, among other things. They also suggest how the GNN can be used for diverse applications, such as predicting the dynamics under various network structures (e.g. designing intervention strategies that affect the way individuals interact and are therefore connected).

Finally, we illustrate the applicability of our approach by training our GNN model using the evolution of COVID-19 in Spain between January 1st 2020 and March 27th 2021 (see Fig. 4). This dataset consists of the daily number of new cases (i.e. incidence) for each of the 50 provinces of Spain as well as Ceuta and Melilla[53]. We also use a network of the mobility flow recorded in 2018[54] as a proxy to model the interaction network between these 52 regions. This network is multiplex—each layer corresponding to a different mode of transportation—, directed and weighted (average daily mobility flow).

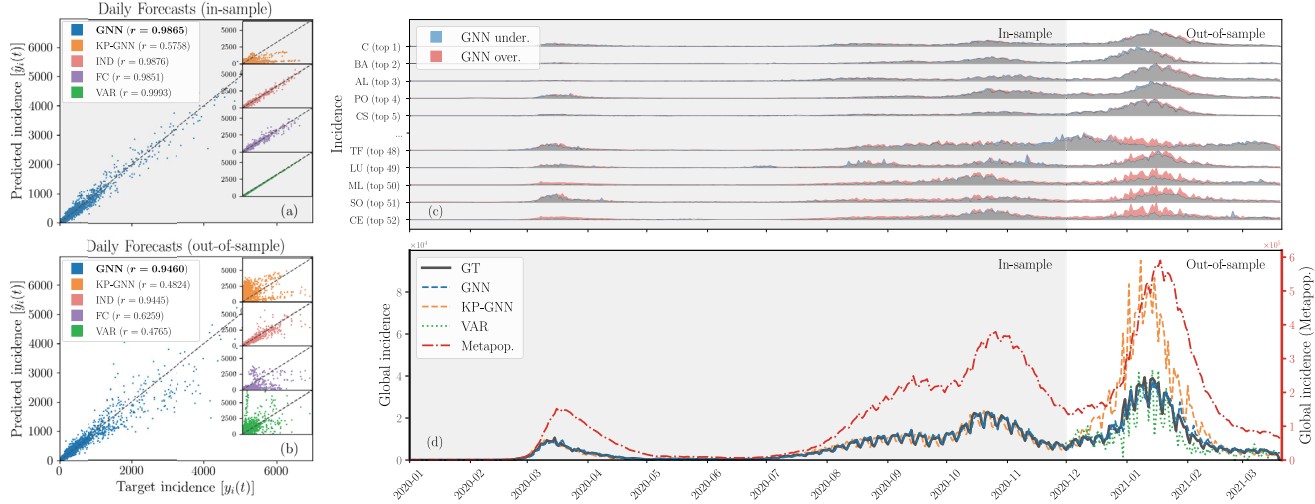

**Fig. 5 Learning the Spain COVID-19 dataset. a, b** Comparison between the targets and the predictions in the in-sample and the out-of-sample datasets for our GNN model (blue) and for other models (KP-GNN in orange, IND in pink, FC in purple and VAR in green; see main text). The accuracy of the predictions is quantified by the Pearson correlation coefficient provided in the legend. **c** Forecasts by our GNN model for individual time series of the provincial daily incidence compared with the ground truth. Underestimation and overestimation are respectively shown in blue and red. Each time series has been rescaled as in Fig. 4b and are ordered according to mean square error of the GNN's predictions. **d** Forecasts for the global incidence (sum of the daily incidence in every province). The solid gray line indicates the ground truth (GT); the dashed blue line, the dashed orange line and dotted green line show the forecast of our GNN model, of KP-GNN and of VAR, respectively. We also show the forecast of an equivalent metapopulation model (red dash-dotted line) which has its own scale (red axis on the right) to improve the visualization; the other lines share the same axis on the left. Similarly to Fig. 4, we differentiate the in-sample from the out-of-sample forecasts using a shaded background.

We compare the performance of our approach with that of different baselines: Four data-driven techniques—three competing neural network architectures[37,55] and a linear vector autoregressive model (VAR)[56,57]—, and an equivalent mechanistic metapopulation model (Metapop.) driven by a simple contagion mechanism[58]. Among the three neural network architectures, we used the model of ref. [37] (KP-GNN) that has been used to predict the evolution of COVID-19 in the US. In a way of an ablation study, the other two GNN architectures embody the assumptions that the nodes of the networks are mutually independent (IND), or that the nodes are arbitrarily codependent (FC) in a way that is learned by the neural network. Note that there exists a wide variety of GNN architectures designed to handle dynamics *of* networks[38]—networks whose topology evolves over time—but that these architectures are not typically adapted for learning dynamics *on* networks (see Sec. III F of the Supplementary Information). Finally, we used the parameters of ref. [59] for the metapopulation model. Section "COVID-19 outbreak in Spain" provides more details on the baselines.

Figure 5 shows that all data-driven models can generate highly accurate in-sample predictions, with the exception of the KP-GNN model which appears to have a hard time learning the dynamics, possibly because of its aggregation mechanism (see Sec. IIIF of the Supplementary Information). This further substantiates the idea that many GNN architectures designed for structure learning, like the graph convolutional network[60] at the core the KP-GNN model, are suboptimal for dynamics learning problems. However, the other architectures do not appear to have the same capability to generalize the dynamics out-of-sample: The FC and the VAR models, especially, tend to overfit more than the GNN and the IND models. While this was expected for the linear VAR model, the FC model overfits because it is granted too much freedom in the way it learns how the nodes interact with one another. Interestingly, the IND and the GNN models seem to perform similarly, which hints at the possibility that the specifics of the mobility network might not have contributed significantly

to the dynamics. This is perhaps not surprising since social distancing and confinement measures were in place during the period covered by the dataset. Indeed, our results indicate that the global effective dynamics was mostly driven by internal processes within each individual province, rather than driven by the interaction between them. This last observation suggests that our GNN model is robust to spurious connections in the interaction network.

Finally, Fig. 5d shows that the metapopulation model is systematically overestimating the incidence by over an order of magnitude. Again, this is likely due to the confinement measures in place during that period which were not reflected in the original parameters of the dynamics[59]. Additional mechanisms accounting for this interplay between social restrictions and the prevalence of the disease—e.g. complex contagion mechanisms[12] or time-dependent parameters[61]—would therefore be in order to extend the validity of the metapopulation model to the full length of the dataset. Interestingly, a signature of this interplay is encoded in the daily incidence data and our GNN model appears to be able to capture it to some extent.

## Discussion

We introduced a data-driven approach that learns effective mechanisms governing the propagation of diverse dynamics on complex networks. We proposed a reliable training protocol, and we validated the projections of our GNN architecture on simple, complex, interacting contagion and metapopulation dynamics using synthetic networks. Interestingly, we found that many standard GNN architectures do not handle correctly the problem of learning contagion dynamics from time series. Also, we found that our approach performs better when trained on data whose underlying network structure is heterogeneous, which could prove useful in real-world applications of our method given the ubiquitousness of scale-free networks[62].

By recovering the bifurcation diagram of various dynamics, we illustrated how our approach can leverage time series from an

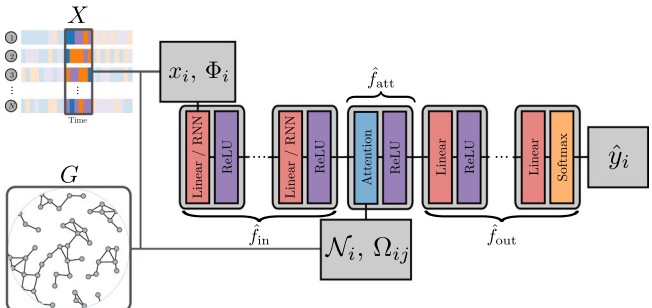

**Fig. 6 Visualization of the GNN architecture.** The blocks of different colors represent mathematical operations. The red blocks correspond to trainable affine transformation parametrized by weights and biases. The purple blocks represent activation functions between each layer. The core of the model is the attention module[41], which is represented in blue. The orange block at the end is an activation function that transforms the output into the proper outcomes.

unknown dynamical process to gain insights about its properties —e.g. the existence of a phase transition and its order. We have also shown how to use this framework on real datasets, which in turn could then be used to help build better effective models. In a way, we see this approach as the equivalent of a numerical Petri dish—offering a new way to experiment and gain insights about an unknown dynamics—that is complementary to traditional mechanistic modeling to design better intervention procedures, containment countermeasures and to perform model selection.

Although we focused the presentation of our method on contagion dynamics, its potential applicability reaches many other realms of complex systems modeling where intricate mechanisms are at play. We believe this work establishes solid foundations for the use of deep learning in the design of realistic effective models of complex systems.

Gathering detailed epidemiological datasets is a complex and labor-intensive process, meaning that datasets suitable for our approach are currently the exception rather than the norm. The current COVID-19 pandemic has, however, shown how an adequate international reaction to an emerging infectious pathogen critically depends on the free flow of information. New initiatives like Golbal health[63] are good examples of how the international epidemiological community is coming together to share data more openly and to make available comprehensive datasets to all researchers. Thanks to such initiatives, it is likely that future pandemics will see larger amount of data available to the scientific community in real time. It is therefore crucial for the community to start developing tools, such as the one presented here, to leverage these datasets so that we are ready for the next pandemic.

## Methods

**Graph neural network and training details.** In this section, we briefly present our GNN architecture, the training settings, the synthetic data generation procedure and the hyperparmeters used in our experiments.

*Architecture.* We use the GNN architecture shown in Fig. 6 and detailed in Table 1. First, we transform the state $x_i$ of every node with a shared multilayer perception (MLP), denoted $\hat{f}_{in} : \mathcal{S} \rightarrow \mathbb{R}^d$ where $d$ is the resulting number of node features, such that

$$\xi_i = \hat{f}_{in}(x_i). \tag{8}$$

We concatenate the node attributes $\Phi_i$ to $x_i$, when these attributes are available, in which case $\hat{f}_{in} : \mathcal{S} \times \mathbb{R}^Q \rightarrow \mathbb{R}^d$. At this point, $\xi_i$ is a vector of features representing the state (and attributes) of node $v_i$. Then, we aggregate the features of the first neighbors using a modified attention mechanism $\hat{f}_{att}$, inspired by ref.[41] (see Section "Attention mechanism"),

$$\nu_i = \hat{f}_{att}(\xi_i, \xi_{\mathcal{N}_i}), \tag{9}$$

where we recall that $\mathcal{N}_i = \{v_j | e_{ij} \in \mathcal{E}\}$ is the set of nodes connected to node $v_i$. We also include the edge attributes $\Omega_{ij}$ into the attention mechanism, when they are available. To do so, we transform the edge attributes $\Omega_{ij}$ into abstract edge features, such that $\psi_{ij} = \hat{f}_{edge}(\Omega_{ij})$ where $\hat{f}_{edge} : \mathbb{R}^P \rightarrow \mathbb{R}^{d_{edge}}$ is also a MLP, before they are used in the aggregation. Finally, we compute the outcome $\hat{y}_i$ of each node $v_i$ with another MLP $\hat{f}_{out} : \mathbb{R}^d \rightarrow \mathcal{R}$ such that

$$\hat{y}_i = \hat{f}_{out}(\nu_i) . \tag{10}$$

*Attention mechanism.* We use an attention mechanism inspired by the graph attention network architecture (GAT)[41]. The attention mechanism consists of three trainable functions $\mathcal{A} : \mathbb{R}^d \rightarrow \mathbb{R}$, $\mathcal{B} : \mathbb{R}^d \rightarrow \mathbb{R}$ and $\mathcal{C} : \mathbb{R}^{d_{edge}} \rightarrow \mathbb{R}$, that combine the feature vectors $\xi_i$, $\xi_j$ and $\psi_{ij}$ of a connected pair of nodes $v_i$ and $v_j$, where we recall that $d$ and $d_{edge}$ are the number of node and edge features, respectively. Then, the attention coefficient $a_{ij}$ is computed as follows

$$a_{ij} = \sigma \Big[ \mathcal{A}(\xi_i) + \mathcal{B}(\xi_j) + \mathcal{C}(\psi_{ij}) \Big] \tag{11}$$

where $\sigma(x) = [1 + e^{-x}]^{-1}$ is the logistic function. Notice that, by using this logistic function, the value of the attention coefficients is constrained to the open interval (0, 1), where $a_{ij} = 0$ implies that the feature $\xi_j$ does not change the value of $v_i$, and $a_{ij} = 1$ implies that it maximally changes the value of $v_i$. In principle, $a_{ij}$ quantifies the influence of the state of node $v_j$ over the outcome of node $v_i$. In reality, the representation learned by the GNN can be non-sparse, meaning that the neighbor features $\xi_{\mathcal{N}_i}$ can be combined in such a way that their noncontributing parts are canceled out without having $a_{ij}$ being necessarily zero. This can result in the failure of this interpretation of this attention coefficients (see the Supplementary Information for further details). Nevertheless, the attention coefficients can be used to assess how connected nodes interact together.

We compute the aggregated feature vectors $\nu_i$ of node $v_i$ as

$$\nu_i = \hat{f}_{att}(\xi_i, \xi_{\mathcal{N}_i}) = \xi_i + \sum_{v_j \in \mathcal{N}_i} a_{ij} \xi_j. \tag{12}$$

It is important to stress that, at this point, $\nu_i$ contains some information about $v_i$ and all of its neighbors in a pairwise manner. In all our experiments, we fix $\mathcal{A}$, $\mathcal{B}$, and $\mathcal{C}$ to be affine transformations with trainable weight matrix and bias vector. Also, we use multiple attention modules in parallel to increase the expressive power of the GNN architecture, as suggested by ref.[41].

The attention mechanism described by Eq. (11) is slightly different from the vanilla version of ref.[41]. Similarly to other well-known GNN architectures[33,60,64], the aggregation scheme of the vanilla GAT is designed as an average of the feature vectors of the neighbors—where, by definition, $\sum_{v_j \in \mathcal{N}_i} a_{ij} = 1$ for all $v_i$—rather than as a general weighted sum like for Eq. (12). This is often reasonable in the context of structure learning, where the node features represent some coordinates in a metric space where connected nodes are likely to be close[33]. Yet, in the general case, this type of constraint was shown to lessen dramatically the expressive power of the GNN architecture[31]. We also reached the same conclusion while using average-like GNN architectures (see Sec. III F of the Supplementary Information). By contrast, the aggregation scheme described by Eq. (12) allows our architecture to represent various dynamic processes on networks accurately.

*Training settings.* In all experiments on synthetic data, we use the cross entropy loss as the local loss function,

$$L(y_i, \hat{y}_i) = -\sum_m y_{i,m} \log \hat{y}_{i,m}, \tag{13}$$

where $y_{i,m}$ corresponds to the $m$-th element of the outcome vector of node $v_i$, which either is a transition probability for the stochastic contagion dynamics or a fraction of people for the metapopulation dynamics. For the simple, complex and interacting contagion dynamics, we used the observed outcomes, i.e. using $y_i \rightarrow \bar{y}_i$ in Eq. (13), corresponding to the stochastic state of node $v_i$ at the next time step, as the target in the loss function. While we noticed a diminished performance when using the observed outcomes as opposed to the true transition probabilities (see Sec. III E of the Supplementary Information), this setting is more realistic and shows what happens when the targets are noisy. The effect of noise can be tempered by increasing the size of the dataset (see the Supplementary Information). For the metapopulation dynamics, since this model is deterministic, we used the true targets without adding noise.

*Performance measures.* We use the Pearson correlation coefficient $r$ as a global performance measure defined on a set of targets $Y$ and predictions $\hat{Y}$ as

$$r = \frac{\mathbb{E}[(Y - \mathbb{E}[Y])(\hat{Y} - \mathbb{E}[\hat{Y}])]}{\sqrt{\mathbb{E}[(Y - \mathbb{E}[Y])^2] \, \mathbb{E}[(\hat{Y} - \mathbb{E}[\hat{Y}])^2]}} \tag{14}$$

where $\mathbb{E}[W]$ denotes the expectation of $W$. Also, because the maximum correlation occurs at $r = 1$, we also define $1 - r$ as the global error on the set of target-prediction pairs.

**Table 1 Layer by layer description of the GNN models for each dynamics.**

| Dynamics | Simple | Complex | Interacting | Metapopulation | COVID-19 |
|---|---|---|---|---|---|
| Input layers | Linear(1, 32) | Linear(1, 32) | Linear(1, 32) | Linear(4, 32)* | RNN(2, 4; L)* |
| | ReLU | ReLU | ReLU | ReLU | ReLU |
| | Linear(32, 32) | Linear(32, 32) | Linear(32, 32) | Linear(32, 32) | RNN(4, 8; L) |
| | ReLU | ReLU | ReLU | ReLU | ReLU |
| | | | Linear(32, 32) | Linear(32, 32) | RNN(8, 16; L)** |
| | | | ReLU | ReLU | ReLU |
| | | | | Linear(32, 32) | Linear(16, 32) |
| | | | | ReLU | ReLU |
| Number of attention layers | 2 | 2 | 4 | 8$^\dagger$ | 5$^{\dagger\dagger}$ |
| Output layers | Linear(32, 32) | Linear(32, 32) | Linear(32, 32) | Linear(32, 32) | Linear(32, 16) |
| | ReLU | ReLU | ReLU | ReLU | ReLU |
| | Linear(32, 2) | Linear(32, 2) | Linear(32, 32) | Linear(32, 32) | Linear(16, 8) |
| | Softmax | Softmax | ReLU | ReLU | ReLU |
| | | | Linear(32, 4) | Linear(32, 32) | Linear(8, 4) |
| | | | Softmax | ReLU | ReLU |
| | | | | Linear(32, 3) | Linear(4, 1) |
| | | | | Softmax | |
| Number of parameters | 6 698 | 6 698 | 11 188 | 99 883 | 7 190 |

For each sequence, the operations are applied from top to bottom. The operations represented by Linear(m, n) correspond to linear (or affine) transformations of the form $f(\mathbf{x}) = \mathbf{W}\mathbf{x} + \mathbf{b}$, where $\mathbf{x} \in \mathbb{R}^m$ is the input, $\mathbf{W} \in \mathbb{R}^{n \times m}$ and $\mathbf{b} \in \mathbb{R}^n$ are trainable parameters. The operation RNN(m, n; L) corresponds to an Elman recurrent neural network module[68] with m input features and n output features applied on sequences of length $L$[68]. The operations ReLU and Softmax are activation functions given by ReLU(x) = max{x, 0} and Softmax$(\mathbf{x}) = \frac{\exp(\mathbf{x})}{\sum_i \exp(x_i)}$.
\* Here, the dimension of the input is increased by one, because we aggregated to the state of the nodes $x_i$ their rescaled and centered population size $N_i$.
\*\* Here, only the features of the last element of the sequence—that corresponding to the state $X_t$—are kept to proceed further into the architecture.
$^\dagger$ Because the networks are weighted for the metapopulation dynamics, we initially transform the edge weights into abstract feature representations using a sequence of layers, i.e. (Linear(1, 4), ReLU, Linear(4, 4)) applied from left to right, before using them in the attention modules. These layers are trained alongside all the other layers.
$^{\dagger\dagger}$ The network is also weighted in this case, hence we used the same set up as for the metapopulation GNN model to transform the edge weights. Also, note that the five attention modules are each associated to a different layer in the multiplex network.

**Synthetic data generation.** We generate data from each dynamics using the following algorithm:

1. Sample a network $G$ from a given generative model (e.g. the Erdős-Rényi $G(N, M)$ or the Barabási-Albert network models).
2. Initialize the state of the system $X(0) = (x_i(0))_{i=1..N}$. For the simple, complex and interacting contagion dynamics, sample uniformly the number of nodes in each state. For the metapopulation dynamics, sample the population size for each node from a Poisson distribution of average $10^4$ and then sample the number of infected people within each node from a binomial distribution of parameter $10^{-5}$. For instance, a network of $|\mathcal{V}| = 10^3$ nodes will be initialized with a total of 100 infected people, on average, distributed among the nodes.
3. At time $t$, compute the observed outcome—$Y_t$ for the metapopulation dynamics, and $\tilde{Y}_t$ for the three stochastic dynamics. Then, record the states $X_t$ and $Y_t$ (or $\tilde{Y}_t$).
4. Repeat step 3 until $(t \mod t_s) = 0$, where $t_s$ is a resampling time. At this moment, apply step 2 to reinitialize the states $X_t$ and repeat step 3.
5. Stop when $t = T$, where $T$ is the targeted number of samples.

The resampling step parametrized by $t_s$ indirectly controls the diversity of the training dataset. We allow $t_s$ to be small for the contagion dynamics ($t_s = 2$) and larger for the metapopulation dynamics ($t_s = 100$) to emphasize on the performance of the GNN rather than the quality of the training dataset, while acknowledging that different values of $t_s$ could lead to poor training (see Sec. III D of the Supplementary Information).

We trained the simple, complex and interacting contagion GNN models on networks of size $|\mathcal{V}| = 10^3$ nodes and on time series of length $T = 10^4$. To generate the networks, we either used Erdős-Rényi (ER) random networks $G(N, M)$ or Barabási-Albert (BA) random networks. In both cases, the parameters of the generative network models are chosen such that the average degree is fixed to $\langle k \rangle = 4$.

To train our models on the metapopulation dynamics, we generated 10 networks of $|\mathcal{V}| = 100$ nodes and generated for each of them time series of $t_s = 100$ time steps. This number of time steps roughly corresponds to the moment where the epidemic dies out. Similarly to the previous models, we used the ER and the BA models to generate the networks, where the parameters were chosen such that $\langle k \rangle = 4$. However, because this dynamics is not stochastic, we varied the average degree of the networks to increase the variability in the time series. This was done by randomly removing a fraction $p = 1 - \ln(1 - \mu + e\mu)$ of their edges, where $\mu$ was sampled for each network uniformly between 0 and 1. In this scenario, the networks were directed and weighted, with each edge weight $e_{ij}$ being uniformly distributed between 0 and 1.

**Hyperparameters.** The optimization of the parameters was performed using the rectified Adam algorithm[65], which is hyperparameterized by $b_1 = 0.9$ and $b_2 = 0.999$, as suggested in ref. [65].

To build a validation dataset, we selected a fraction of the node states randomly for each time step. More specifically, we randomly chose node $v_i$ at time $t$ proportionally to its importance weight $w_i(t)$. For all experiments on synthetic dynamics, we randomly selected 10 nodes per time step to be part of the validation set, on average. For all experiments, the learning rate $\epsilon$ was reduced by a factor 2 every 10 epochs with initial value $\epsilon_0 = 0.001$. A weight decay of $10^{-4}$ was used as well to help regularize the training. We trained all models for 30 epochs, and selected the GNN model with the lowest loss on validation datasets. We fixed the importance sampling bias exponents for the training to $\lambda = 0.5$ in the simple, complex and interacting contagion cases, and fixed it to $\lambda = 1$ in the metapopulation case.

**Importance weights.** In this section, we show how to implement the importance weights in the different cases. Other versions of the importance weights are also available in Sec. I of the Supplementary Information.

*Discrete state stochastic dynamics.* When $\mathcal{S}$ is a finite countable set, the importance weights can be computed exactly using Eq. (7),

$$w_i(t) \propto \left[\rho\left(k_i, x_i(t), x_{\mathcal{N}_i}(t)\right)\right]^{-\lambda} \tag{15}$$

where $\rho(k, x, x_{\mathcal{N}})$ is the probability to observe a node of degree $k$ in state $x$ with a neighborhood in state $x_{\mathcal{N}}$ in the complete dataset $D$. The inputs can be simplified from $(k, x, x_{\mathcal{N}})$ to $(k, x, \boldsymbol{\ell})$ without loss of generality, where $\boldsymbol{\ell}$ is a vector whose entries are the number of neighbors in each state. The distribution is then estimated from the complete dataset $D$ by computing the fraction of inputs that are in every configuration

$$\rho(k, x, ) = \frac{1}{|\mathcal{V}|T}\sum_{i=1}^{|\mathcal{V}|} I(k_i = k) \\ \times \sum_{t=1}^{T} I(x_i(t) = x)\ I(_i(t) = \ell) \tag{16}$$

where $I(\cdot)$ is the indicator function.

*Continuous state deterministic dynamics.* The case of continuous states—e.g. for metapopulation dynamics—is more challenging than its discrete counterpart, especially if the node and edge attributes, $\Phi_i$ and $\Omega_{ij}$, need to be accounted for. One of the challenges is that we cannot count the inputs like in the discrete case. As a result, the estimated distribution $\rho$ cannot be estimated directly using Eq. (16), and we use instead

$$w_i(t) = [P(k_i)\,\Sigma(\Phi_i, \Omega_i|k_i)\,\Pi(\bar{x}(t))]^{-\lambda} \tag{17}$$

where $P(k_i)$ is the fraction of nodes with degree $k_i$, $\Sigma(\Phi_i, \Omega_i|k_i)$ is the joint probability density function (pdf) conditioned on the degree $k_i$ for the node attributes

$\Phi_i$ and the sum of the edge attributes $\Omega_i \equiv \sum_{v_j \in \mathcal{N}_i} \Omega_{ij}$, and where $\Pi(\bar{x}(t))$ is the pdf for the average of node states at time $t\bar{x}(t) = \frac{1}{|V|}\sum_{v_i \in \mathcal{V}} x_i(t)$. The pdf are obtained using nonparametric Gaussian kernel density estimators (KDE)[66]. Provided that the density values of the KDE are unbounded above, we normalize the pdf such that the density of each sample used to construct the KDE sum to one. Further details on how we designed the importance weights are provided in Sec. I of the Supplementary Information.

**Dynamics.** In what follows, we describe in detail the contagion dynamics used for our experiments. We specify the node outcome function $f$ introduced in Eq. (3) and the parameters of the dynamics.

*Simple contagion.* We consider the simple contagion dynamics called the susceptible-infected-susceptible (SIS) dynamics for which $\mathcal{S} = \{S, I\} = \{0, 1\}$—we use these two representations of $\mathcal{S}$ interchangeably. Because this dynamics is stochastic, we let $\mathcal{R} = [0,1]^2$. We define the infection function $\alpha(\ell)$ as the probability that a susceptible node becomes infected given its number of infected neighbors $\ell$

$$\Pr(S \to I | \ell) = \alpha(\ell) = 1 - (1-\gamma)^\ell , \tag{18}$$

where $\gamma \in [0, 1]$ is the disease transmission probability. In other words, a node can be infected by any of its infected neighbors independently with probability $\gamma$. We also define the constant recovery probability as

$$\Pr(I \to S) = \beta . \tag{19}$$

The node outcome function for the SIS dynamics is therefore

$$f(x_i, x_{\mathcal{N}_i}) = \begin{cases} (1 - \alpha(\ell_i), \ \alpha(\ell_i)) & \text{if } x_i = 0, \\ (\beta, \ 1 - \beta) & \text{if } x_i = 1, \end{cases} \tag{20}$$

where

$$\ell_i = \sum_{v_j \in \mathcal{N}_i} \delta(x_j, 1) \tag{21}$$

is the number of infected neighbors of $v_i$ and $\delta(x, y)$ is the Kronecker delta. Note that for each case in Eq. (20), the outcome is a two-dimensional probability vector, where the first entry is the probability that node $v_i$ becomes/remains susceptible at the following time step, and the second entry is the probability that it becomes/remains infected. We used $(\gamma, \beta) = (0.04, 0.08)$ in all experiments involving this simple contagion dynamics.

*Complex contagion.* To lift the independent transmission assumption of the SIS dynamics, we consider a complex contagion dynamics for which the node outcome function has a similar form as Eq. (20), but where the infection function $\alpha(\ell)$ has the nonmonotonic form

$$\alpha(\ell) = \frac{1}{z(\eta)} \frac{\ell^3}{e^{\ell/\eta} - 1} \tag{22}$$

where $z(\eta)$ normalizes the infection function such that $\alpha(\ell^*) = 1$ at its global maximum $\ell^*$ and $\eta > 0$ is a parameter controlling the position of $\ell^*$. This function is inspired by the Planck distribution for the black-body radiation, although it was chosen for its general shape rather than for any physical meaning whatsoever. We used $(\eta, \beta) = (8, 0.06)$ in all experiments involving this complex contagion dynamics.

*Interacting contagion.* We define the interacting contagion as two SIS dynamics that are interacting and denote it as the SIS-SIS dynamics. In this case, we have $\mathcal{S} = \{S_1 S_2, I_1 S_2, S_1 I_2, I_1 I_2\} = \{0, 1, 2, 3\}$. Simiarly to the SIS dynamics, we have $\mathcal{R} = [0, 1]^4$ and we define the infection probability functions

$$\alpha_g(\ell_g) = 1 - (1 - \gamma_g)^{\ell_g} \ \text{if} \ x = 0 \tag{23a}$$

$$\alpha_g^*(\ell_g) = 1 - (1 - \zeta\gamma_g)^{\ell_g} \ \text{if} \ x = 1, 2 , \tag{23b}$$

where $\zeta \geq 0$ is a coupling constant and $\ell_g$ is the number of neighbors infected by disease $g$, and also define the recovery probabilities $\beta_g$ for each disease ($g = 1, 2$). The case where $\zeta > 1$ corresponds to the situation in which the diseases are synergistic (i.e. being infected by one increases the probability of getting infected by the other), whereas competition is introduced if $\zeta < 1$ (being already infected by one decreases the probability of getting infected by the other). The case $\zeta = 1$ falls back on two independent SIS dynamics that evolve simultaneously on the network. The outcome function is composed of 16 entries that are expressed as follows

$$f(x_i, x_{\mathcal{N}_i})$$
$$= \begin{cases} \left([1 - \alpha_1(\ell_{i,1})] [1 - \alpha_2(\ell_{i,2})], \ \alpha_1(\ell_{i,1}) [1 - \alpha_2(\ell_{i,2})], \ [1 - \alpha_1(\ell_{i,1})] \alpha_2(\ell_{i,2}), \ \alpha_1(\ell_{i,1})\alpha_2(\ell_{i,2})\right) & \text{if } x_i = 0, \\ \left(\beta_1[1 - \alpha_2^*(\ell_{i,2})], \ [1 - \beta_1] [1 - \alpha_2^*(\ell_{i,2})], \ \beta_1\alpha_2^*(\ell_{i,2}), \ [1 - \beta_1]\alpha_2^*(\ell_{i,2})\right) & \text{if } x_i = 1, \\ \left([1 - \alpha_1^*(\ell_{i,1})] \beta_2, \ \alpha_1^*(\ell_{i,1})\beta_2, \ [1 - \alpha_1^*(\ell_{i,1})] [1 - \beta_2], \ \alpha_1^*(\ell_{i,1}) [1 - \beta_2]\right) & \text{if } x_i = 2, \\ (\beta_1\beta_2, \ [1 - \beta_1]\beta_2, \ \beta_1[1 - \beta_2], \ [1 - \beta_1][1 - \beta_2]) & \text{if } x_i = 3. \end{cases}$$
$$\tag{24}$$

where we define $\ell_{i,g}$ as the number of neighbors of $v_i$ that are infected by disease $g$. We used $(\gamma_1, \gamma_2, \beta_1, \beta_2, \zeta) = (0.01, 0.012, 0.19, 0.22, 50)$ in all experiments involving this interacting contagion dynamics.

*Metapopulation.* The metapopulation dynamics considered is a deterministic version of the susceptible-infection-recovered (SIR) metapopulation model[46–49]. We consider that the nodes are populated by a fixed number of people $N_i$, which can be in three states—susceptible (S), infected (I) or recovered (R). We therefore track the number of people in every state at each time. Furthermore, we let the network $G$ be weighted, with the weights describing the mobility flow of people between regions. In this case, $\Omega_{ij} \in \mathbb{R}$ is the average number of people that are traveling from node $v_j$ to node $v_i$. Finally, because we assume that the population size is on average steady, we let $\Phi_i = N_i$ be a node attribute and work with the fraction of people in every epidemiological state. More precisely, we define the state of node $v_j$ by $x_j = (s_j, i_j, r_j)$, where $s_j$, $i_j$ and $r_j$ are the fractions of susceptible, infected and recovered people, respectively. From these definitions, we define the node outcome function of this dynamics as

$$f(x_j, x_{\mathcal{N}_j}, G) = \begin{pmatrix} s_j - s_j\tilde{\alpha}_j \\ i_j - \frac{i_j}{\tau_r} + s_j\tilde{\alpha}_j \\ r_j + \frac{i_j}{\tau_r} \end{pmatrix} \tag{25}$$

where

$$\tilde{\alpha}_j = \alpha(i_j, N_j) + \sum_{v_l \in \mathcal{N}_j} \frac{k_j \Omega_{jl}\alpha(i_l, N_l)}{\sum_{v_n \in \mathcal{N}_j} \Omega_{jn}} , \tag{26}$$

and $k_j$ is the degree of node $v_j$. The function $\alpha(i, N)$ corresponds to the infection rate, per day, at which an individual is infected by someone visiting from a neighboring region with $iN$ infected people in it, and is equal to

$$\alpha(i, N) = 1 - \left(1 - \frac{R_0}{\tau_r N}\right)^{iN} \approx 1 - e^{-\frac{R_0}{\tau_r}i} . \tag{27}$$

where $R_0$ corresponds to the reproduction number and, $\tau_r$ is the average recovery time in days. In all experiments with this metapopulation dynamics, we used $(R_0, \tau_r) = (8.31, 7.5)$.

## COVID-19 outbreak in Spain

*Dataset.* The dataset is composed of the daily incidence of the 52 Spanish provinces (including Ceuta and Melilla) monitored for 450 days between January 1st 2020 and March 27th 2021[53]. The dataset is augmented with the origin-destination (OD) network of individual mobility[54]. This mobility network is multiplex, directed and weighted, where the weight of each edge $e_{ij}^\nu$ represents the mobility flow from province $v_j$ and to province $v_j$ using transportation $\nu$. The metadata associated to each node is the population of province $v_i$[67], noted $\Phi_i = N_i$. The metadata associated to each edge, $\Omega_{ij}^\nu$, corresponds to the average number of people that moved from $v_j$ to $v_i$ using $\nu$ as the main means of transportation.

*Models.* The GNN model used in Fig. 5 is very similar to the metapopulation GNN model—with node and edge attributes—with the exception that different attention modules are used to model the different OD edge types (plane, car, coach, train, and boat, see Table 1). To combine the features of each layer of the multiplex network, we average pooled the output features of the attention modules. We also generalize our model to take in input a sequence of $L$ states of the system, that is

$$\hat{Y}_t = \hat{\mathcal{M}}(X_{t:t-L+1}, G; \boldsymbol{\Theta}) \tag{28}$$

where $X_{t:t-L+1} = (X_t, X_{t-1}, \cdots, X_{t-L+1})$ and $L$ is a lag. At the local level, it reads

$$\hat{y}_i(t) = \hat{f}\big(x_i(t : t - L + 1), \\ \Phi_i, x_{\mathcal{N}_i}(t : t - L + 1), \Phi_{\mathcal{N}_i}, \Omega_{i\mathcal{N}_i}; \boldsymbol{\Theta}\big) \tag{29}$$

where $x_i(t : t - L + 1)$ corresponds to the $L$ previous state of node $i$ from time $t$ to time $t - L + 1$. As we now feed sequences of node states to the GNN, we use Elman recurrent neural networks[68] to transform these sequences of states before aggregating them instead of linear layers, as shown in Fig. 6 and Table 1. Additionally, because the outputs of the models are not probability vectors, like for the dynamics of Sec. "Dynamics", but real numbers, we use the mean square error (MSE) loss to train the model:

$$L(y_i, \hat{y}_i) = (y_i - \hat{y}_i)^2 . \tag{30}$$

We use five different baseline models to compare with the performance of our GNN: Three additional neural network architectures, a vector autoregressive model (VAR)[56] and an equivalent metapopulation model driven by a simple contagion mechanism. The first neural network architecture, denoted the KP-GNN model, was used in ref. [37] to forecast the evolution COVID-19 in the US using a similar strategy as ours with respect to the mobility network. As described in ref. [37], we used a single-layered MLP with 64 hidden units to transform the input, and then we used two graph convolutional networks (GCN) in series, each with 32 hidden units, to perform the feature aggregation. Finally, we computed the output of the model using another single-layered MLP with 32 hidden units. The layers of this

model are separated by ReLU activation functions and are sampled with a dropout rate of 0.5. Because this model is not directly adapted to multiplex networks, we merged all layers together into a single network and summed the weights of the edges. Then, as prescribed in ref. [37], we thresholded the merged network by keeping at most 32 neighbors with the highest edge weight for each node. We did not use our importance sampling procedure to train the KP-GNN model—letting $\lambda = 0$—to remain as close as possible to the original model.

The other two neural network architectures are very similar to the GNN model we presented in Table 1: The only different component is their aggregation mechanism. The IND model, where the nodes are assumed to be mutually independent, does not aggregate the features of the neighbors. It therefore acts like a univariate model, where the time series of each node are processed like different elements of a minibatch. In the FC model, the nodes interact via a single-layered MLP connecting all nodes together. The parameters of this MLP are learnable, which effectively allows the model to express any interaction patterns. Because the number of parameters of this MLP scales with $d|\mathcal{V}|^2$, where $d$ is the number of node features after the input layers, we introduce another layer of 8 hidden units to compress the input features before aggregating them.

The VAR model is a linear generative model adapted for multivariate time series forecasting

$$\hat{Y}_t = \hat{\mathcal{M}}(X_t, X_{t-1}, \cdots, X_{t-L+1}) = \sum_{l=0}^{L-1} A_l X_{t-l} + \boldsymbol{b} + \boldsymbol{\epsilon}_t \quad (31)$$

where $A_l \in \mathbb{R}^{|\mathcal{V}| \times |\mathcal{V}|}$ are weight matrices, $\boldsymbol{b} \in \mathbb{R}^{|\mathcal{V}|}$ is a trend vector and $\boldsymbol{\epsilon}_t$ is an error term with $\mathbb{E}[\boldsymbol{\epsilon}_t] = \boldsymbol{0}$ and $\mathbb{E}[\boldsymbol{\epsilon}_t \boldsymbol{\epsilon}_s] = \delta_{t,s} \Sigma$, with $\Sigma$ being a positive-semidefinite covariance matrix. While autoregressive models are often used to predict stock markets[69], they have also been used recently to forecast diverse COVID-19 outbreaks[57]. This model is fitted to the COVID-19 time series dataset also by minimizing the MSE.

The metapopulation model is essentially identical to the model presented in Sec. "Dynamics". However, because we track the incidence, i.e. the number of newly infectious cases $\chi_i(t)$ in each province $i$, instead of the complete state $(S_i, I_i, R_i)$ representing the number of individuals in each state, we allow the model to internally track the complete state based on the ground truth. At first, the whole population is susceptible, i.e. $S_i(1) = N_i$. Then, at each time step, we subtract the number of newly infectious cases in each node from $S_i(t)$, and add it to $I_i$. Finally, the model allows a fraction $\frac{1}{\tau_r}$ of its infected people to recover. The evolution equations of this model are as follows

$$S_i(t+1) = S_i(t) - \chi_i(t) , \quad (32a)$$

$$I_i(t+1) = I_i(t) + \chi_i(t) - \frac{1}{\tau_r} I_i(t) , \quad (32b)$$

$$R_i(t+1) = R_i(t) + \frac{1}{\tau_r} I_i(t) . \quad (32c)$$

Finally, we computed the incidence $\hat{\chi}_i(t)$ predicted by the metapopulation model using the current internal state as follows:

$$\hat{\chi}_i(t) = S_i \tilde{\alpha}_i , \quad (33)$$

where $\tilde{\alpha}_i$ is given by Eq. (26), using the mobility network $G$, Eq. (27) for $\alpha(i, N)$ and $i_j = \frac{I_j}{N_j}$. Since the mobility weights $\Omega_{ij}^\nu$ represent the average number of people traveling from province $\nu_j$ to province $\nu_i$, we assumed all layers to be equivalent and aggregated each layer into a single typeless network where $\Omega_{ij} = \sum_\nu \Omega_{ij}^\nu$. We fixed the parameters of the model to $R_0 = 2.5$ and $\tau_r = 7.5$, as these values were used in other contexts for modeling the propagation of COVID-19 in Spain[59].

*Training.* We trained the GNN and other neural networks for 200 epochs, while decreasing the learning rate by a factor of 2 every 20 epochs with an initial value of $10^{-3}$. For our GNN, the IND and the FC models, we fixed the importance sampling bias exponent to $\lambda = 0.5$ and, like the models trained on synthetic data, we used a weight decay of $10^{-4}$ (see Sec. "Training settings"). We fixed the lag of these models, including the VAR model, to $L = 5$. The KP-GNN model was trained using a weight decay of $10^{-5}$ following ref. [37], and we chose a lag of $L = 7$. For all models, we constructed the validation dataset by randomly selecting a partition of the nodes at each time step proportionally to their importance weights $w_i(t)$: 20% of the nodes are used for validation in this case. The test dataset was constructed by selecting the last 100 time steps of the time series of all nodes, which rough corresponds to the third wave of the outbreak in Spain.

## Data availability
The raw COVID-19, mobility and population datasets are publicly available respectively from[53,54], and[67]. The processed datasets shown in Fig. 4 and used in Fig. 5 are available on Zenodo (https://doi.org/10.5281/zenodo.5015063)[70].

## Code availability
The software developed for this project is available on Zenodo (https://doi.org/10.5281/zenodo.4974521)[71].

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

## Acknowledgements
The authors are grateful to Emily N. Cyr and Guillaume St-Onge for their comments and to Vincent Thibeault, Xavier Roy-Pomerleau, François Thibault, Patrick Desrosiers, Louis J. Dubé, Simon Hardy and Laurent Hébert-Dufresne for fruitful discussions. This work was supported by the Sentinelle Nord initiative from the Fonds d'excellence en recherche Apogée Canada (C.M., E.L., A.A.), the Conseil de recherches en sciences naturelles et en génie du Canada (C.M., A.A.) and the Fonds de recherche du Québec-Nature et technologie (E.L.).

## Author contributions
C.M., E.L., and A.A. designed the study. C.M. and E.L. designed the models and methods. C.M. performed the numerical analysis. C.M. and A.A. wrote the paper.

## Competing interests
The authors declare no competing interests.
