## [Peer Review File · Nature Communications]

Reviewers' Comments:

Reviewer #1:

Remarks to the Author:

This paper proposes a complementary approach based on Graph Neural Networks to automatically learn the stochastic dynamics from time series data on complex networks. By predicting the local transition probabilities of each node on various states, the authors provide a data-driven approach to learn the effective mechanisms governing the propagation process of contagion. The accuracy of the proposed method is demonstrated through a lot of experiments. On generated networks from classical patterns such as Barabasi-Albert random network, the proposed method successfully fits the ground truth. On real-world temporal networks, the proposed method recovers the bifurcation diagram of different contagion dynamics based on epidemic models well. The experiment settings are rigorous and the results are convincing.

However, in my opinion, the major contribution of this paper is not a novel one. GNN is a hot topic in data mining and machine learning communities, and there are hundreds of papers each year which apply GNN to learn various networks. Some of them also deal with dynamic networks and achieve good results, such as the papers in [1]. I read the details of the architecture and the attention mechanism of the proposed model, but do not find significant novel design that can make it stand out from the crowd. I suggest that the authors might highlight the key technical difference of the proposed method and why former models cannot solve the same problem.

The authors claim that the proposed method can uncover the underlying mechanism and gain insights about the properties. However, from my point of view, it is more like fitting history data rather than uncovering the underlying mechanism. The proposed method fits the probabilities of transition pretty well, but it is not clear why these probabilities are these values or what distributions they are, for which it is still far from revealing the mechanism. Although the results are accurate, the proposed method cannot reveal what the mechanism is, for which it faces similar problems to most machine learning methods, such as interpretability and stability. I suggest that the authors should show how they uncover the mechanism and what insights they gain from the proposed method. And the mechanism uncovered by the proposed method should be a complicated one from real-world datasets, because simple mechanisms like the epidemic model can be easily handled by mechanistic models.

As for the potential influence of this paper, since using deep learning methods in network science and propagation process are relatively common research ideas in recent years, I do not think that this paper is an enlightening work.

[1]Skarding, J., Gabrys, B., & Musial, K. (2020). Foundations and modelling of dynamic networks using Dynamic Graph Neural Networks: A survey. arXiv preprint arXiv:2005.07496.

Reviewer #2:

Remarks to the Author:

In their manuscript "Deep learning of stochastic contagion dynamics on complex networks" the authors develop a Deep Learning framework to estimate the transition probabilities for different classes of spreading processes. The authors then demonstrate the goodness of the predictions against several types of static and temporal networks.

The approach is surely interesting and in line with other works appearing in the literature in this period. The use of machine learning and in particular of deep learning to study dynamical processes on graphs will be an important area of research in the near future, thus the work is surely timely. On the technical point of view, I could not find evident flaws in both the definition of the model and in the analyses. However, what is missing, and I believe it is fundamental for a Nat. Comms. paper, is to demonstrate the relevance and applicability of the approach. In terms of methodology the work represents an important step however, from the manuscript, it is impossible to understand its relevance for real problems: e.g. estimating the parameters of a new disease or distinguish between different types of spreading from empirical data. Since deep learning (and

machine learning methods in general) do not provide mechanistic explanations, it is fundamental that they will be able to solve "real world" problems, otherwise they will just remain an exercise in style.

General:

- As I already said, from the main text and the SI it is hard to understand the possible implications of the work for real problems. This kind of discussion is totally missing but I believe that if the authors will be able to provide even few examples of how their framework could be useful in epidemiology or the study of other spreading phenomena (e.g. complex contagion in social sciences), it will add a great value to the work.

Major remarks:

- Along the same line, one question is: will it be possible to train the model with real epidemiological data? For example to estimate the parameters of an emerging disease. Epidemiological data are usually extremely limited and noisy (especially at the contact network level that is the focus of the work) while I see from the SI that the training datasets used are in the order of 10^2 - 10^4 data points. What is the limit in terms of minimum size, noise, etc. of the training dataset to have meaningful results?

- Could the framework be used for model selection? E.g. to distinguish between simple and complex contagion or complex and interacting diseases (the problem posed in ref. 9 of the main text). From the manuscript it seems that the authors use specific LTPs for different types of contagion (eqs. 5 for simple and eq. 6 for complex): i.e. they assume the full knowledge of the contagion process that generated the data. What would happen if I use a model trained for complex contagion to analyze data generated from a simple one? And, based on the results, would it be possible to decide which is the underlying process?

Minor remarks:

- One of the assumptions made by the authors is that the underlying process is stationary. However, for many models and also in real data (both epidemiological or from computational social sciences) this assumption does not hold. This limits the applicability of the framework and should be discussed in the text. Moreover, how stringent is this limitation in practice? Since the model learns the LTPs only from the state of the first neighbors and does not consider dynamical correlations, analyzing data from an SIR dynamics should not be a problem, right?

- The part of the text introducing the stars graphs of different sizes while describing Fig.2 is quite confusing. I had to read it three times to understand why the authors are using the k-stars for the comparison and if they have been used in the training or in the testing. Would it not be easier to use other random networks? E.g. with different maximum degrees or sizes?

- From the main text it is hard to understand why the authors use the approximated master equation framework to get the phase diagram of the three processes instead of standard MonteCarlo simulations (reading the SI things become just a bit clearer). Thus, my suggestion is also to clarify this part.

Reviewer #3:

Remarks to the Author:

Epidemics on networks have recently been used in multiple fields eg. spread of disease or fake news through contact or social networks. But how the underlying dynamics of spreading works in nature is not known to us, hence multiple statistical or mechanistic models have been proposed for specific situations and inference schemes for fitting them to the observed dataset.

The two main contributions of the present paper which would be of interest to the community is,

- Consider models of epidemics on networks whose Markovian nature is modelled using a Graphical neural network (+attention) rather than an interpretable mechanism.
- Choose the best among the above models by minimizing an approximate cross-entropy loss.
- Illustrates that the learned model can show similar behaviours in limit to the true underlying models.

Although I should point that the lack of interpretability of the learnt mechanism may not give us the same amount of scientific intuition/justification we can gain from simpler and more intuitive mechanistic models, which may significantly restrict the adaptation of the models/techniques presented here. In simple words, a domain scientist does not get any scientific knowledge out of the learned model here.

Only justification of using a 'learned' model like the one described here would be a better prediction performance. Unfortunately the present paper does not comment on the prediction performance of the strategy proposed for epidemics compared to the traditionally used epidemics models.

Further, one of the main interests in the epidemics community is the optimal intervention mechanism - which is also not very clear here, how could we learn that for an 'non-interpretable' model like this?

Saying all of the above, I would say the paper has novelty but may be not appropriate for the present venue and the paper can be further improved by adding results regarding the prediction performance or how intervention strategy can interact with these types of models.

Response to Referee 1

Comment 1.1

This paper proposes a complementary approach based on Graph Neural Networks to automatically learn the stochastic dynamics from time series data on complex networks. By predicting the local transition probabilities of each node on various states, the authors provide a data-driven approach to learn the effective mechanisms governing the propagation process of contagion. The accuracy of the proposed method is demonstrated through a lot of experiments. On generated networks from classical patterns such as Barabasi-Albert random network, the proposed method successfully fits the ground truth. On real-world temporal networks, the proposed method recovers the bifurcation diagram of different contagion dynamics based on epidemic models well. The experiment settings are rigorous and the results are convincing.

Reply 1.1

We acknowledge this generous presentation of our work and we thank Reviewer 1 for it.

Comment 1.2

However, in my opinion, the major contribution of this paper is not a novel one. GNN is a hot topic in data mining and machine learning communities, and there are hundreds of papers each year which apply GNN to learn various networks. Some of them also deal with dynamic networks and achieve good results, such as the papers in [1].

Reply 1.2

We disagree with Reviewer 1 with respect to the novelty of our work and contributions for a couple of reasons. We believe that, while our GNN architecture itself is not one of our major contributions but a contribution nonetheless, the application that we make of it and our training procedures are novel. We detail these points below.

First, we use graph neural networks (GNN) for applications, we believe, are novel. As pointed out by Reviewer 1, in the literature, GNN are typically used to learn network structure, that is learning some node embedding which codify in a lower-dimensional space their position in a network [4–7]. In our case, we are trying to accomplish something entirely different. Instead, we want to learn a dynamical model which has an underlying networked structure that we consider arbitrary. That being said, we understand that the GNN effectively computes some node embedding similarly to other GNN models, but, rather than being related to their position in the network, these embeddings encode their states with respect to the dynamical system.

Second, in order for our trained models to be accurate across a large range of network structures and dynamical regimes, we found that our importance sampling procedure was very beneficial. The important sampling procedure surely constitutes our second contribution, which, to the best of our knowledge, is also novel.

Finally, by investigating the bifurcation diagrams, we also showed that our approach can be used to design various applications involving the alteration of the underlying network. The design of intervention protocols, or containment measures, and forecasting are just a few examples of these applications.

Action taken 1.2

We include a new paragraph in the introduction discussing in detail the conceptual difference between structure learning and dynamics learning. Also, at the end of the introduction, we enumerate our three main contributions.

Comment 1.3

I read the details of the architecture and the attention mechanism of the proposed model, but do not find significant

novel design that can make it stand out from the crowd. I suggest that the authors might highlight the key technical difference of the proposed method and why former models cannot solve the same problem.

Reply 1.3

We agree with Reviewer 1 that our architecture specifically is not entirely novel. In fact, we do not claim it to be and we believe our main contributions, as pointed out in Reply 1.2, do not concern our architecture directly. That being said, we found that not all GNN architectures can accomplish this successfully (see Figs. 6 and 7 of the Supplementary Material), which justifies the modifications we included in our graph attention network model.

Action taken 1.3

We included a comparison analysis between our GNN architecture and other standard GNN approaches for the problem of learning the susceptible-infections-susceptible (SIS) dynamics on Barabási-Albert networks in the Supplementary Material. We also comment in the Supplementary Material about other GNN architectures and why they are unsuccessful [8].

Comment 1.4

The authors claim that the proposed method can uncover the underlying mechanism and gain insights about the properties. However, from my point of view, it is more like fitting history data rather than uncovering the underlying mechanism. The proposed method fits the probabilities of transition pretty well, but it is not clear why these probabilities are these values or what distributions they are, for which it is still far from revealing the mechanism. Although the results are accurate, the proposed method cannot reveal what the mechanism is, for which it faces similar problems to most machine learning methods, such as interpretability and stability. I suggest that the authors should show how they uncover the mechanism and what insights they gain from the proposed method. And the mechanism uncovered by the proposed method should be a complicated one from real-world datasets, because simple mechanisms like the epidemic model can be easily handled by mechanistic models.

Reply 1.4

We agree with Reviewer 1 that the "uncovering underlying mechanisms" claim might be confusing and even misleading. It is true that, from a trained GNN, it is difficult to extract an interpretation of its parameters and thus of the mechanisms it has learned. However, we believe that our method can do more than fitting the history data. Indeed, our method relies on the relationship between the time series data and the network structure that we assume generated it. Hence, once trained, we can use the model to investigate dynamical properties of the system by changing the underlying network structure. We give an example of this in the main paper with Fig. 3, of which we uncover from the learned models their bifurcation diagrams and their critical thresholds by using the model on networks of varying average degree.

Action taken 1.4

In order to prevent confusion, we removed from the main text the claim that our approach can uncover the underlying mechanism from data. Instead, we present our approach in the introduction and in the conclusion as a way to build effective models (*numerical Petri dish*).

Comment 1.5

As for the potential influence of this paper, since using deep learning methods in network science and propagation process are relatively common research ideas in recent years, I do not think that this paper is an enlightening work.

Reply 1.5

In very recent years, many works regarding learning dynamical systems from data have been published [9–13], even in prestigious journals such as in the *Nature* family [14]. While many of them focus on investigating this problem on

small or regularly structured systems, the idea of including general and possibly large underlying network structures in the approach is only emerging [15–17]. Thus, as pointed out by Reviewer 2, even though it touches on commonly studied fields of research such as deep learning, network science and propagation processes, our work is still very relevant to this young emerging field.

Action taken 1.5

As mentioned above, we included a new paragraph in the introduction that locates appropriately our contribution into this line of works and discussed the relevance to include an underlying network into learning dynamical systems.

Response to Referee 2

Comment 2.1

In their manuscript "Deep learning of stochastic contagion dynamics on complex networks" the authors develop a Deep Learning framework to estimate the transition probabilities for different classes of spreading processes. The authors then demonstrate the goodness of the predictions against several types of static and temporal networks.

The approach is surely interesting and in line with other works appearing in the literature in this period. The use of machine learning and in particular of deep learning to study dynamical processes on graphs will be an important area of research in the near future, thus the work is surely timely. On the technical point of view, I could not find evident flaws in both the definition of the model and in the analyses.

Reply 2.1

We thank Reviewer 2 to highlight the timeliness of our work.

Comment 2.2

However, what is missing, and I believe it is fundamental for a Nat. Comms. paper, is to demonstrate the relevance and applicability of the approach. In terms of methodology the work represents an important step however, from the manuscript, it is impossible to understand its relevance for real problems: e.g. estimating the parameters of a new disease or distinguish between different types of spreading from empirical data. Since deep learning (and machine learning methods in general) do not provide mechanistic explanations, it is fundamental that they will be able to solve "real world" problems, otherwise they will just remain an exercise in style.

Reply 2.2

We could not agree more with Reviewer 2 that the applicability to real systems of our approach is essential for it to have any value whatsoever. In fact, Reviewer 2 will find below, in Reply 2.3, a response that addresses directly and specifically this issue. In line with suggestions coming from Reviewer 3, we also include in Replies 3.3 and 3.4 some applications of our approach, which further substantiates its applicability to real systems.

Comment 2.3

Along the same line, one question is: will it be possible to train the model with real epidemiological data? For example, to estimate the parameters of an emerging disease. Epidemiological data are usually extremely limited and noisy (especially at the contact network level that is the focus of the work) while I see from the SI that the training datasets used are in the order of $10^2 - 10^4$ data points. What is the limit in terms of minimum size, noise, etc. of the training dataset to have meaningful results?

Reply 2.3

Reviewer 2 is absolutely right, and we thank them for pointing this out. Real epidemiological datasets, in the form of time series of the epidemic states of real people on real contact networks, are quite rare and difficult to work with for reasons listed by Reviewer 2. Even though we believe that our approach could absolutely be used on this kind of datasets, it would surely necessitate a large amount of data to properly train a model, as it is the case for most deep learning models. For example, in our experiments, we achieved high performances only if the sample size of the training datasets were large enough—datasets with about 1000 time steps on networks of size 1000 yielded models with accurate results. We found that, in the case of stochastic dynamics, more data are necessary if the correct bifurcation diagrams are to be recovered accurately by the GNN models.

FIG. 1. **Main results from the metapopulation dynamics experiment:** (a) comparison of the targets and predictions for GNNs trained on Erdős-Rényi networks, (b) comparison of the targets and predictions for GNNs trained on Barabási-Albert networks, (c) Pearson error between the targets and the predictions as a function of the number of neighbors, (d) reconstruction of the bifurcation diagram of the metapopulation dynamics on Poisson networks with varying average degree. All these figures can be found in the main text, hence we refer to it for details.

That being said, our approach is easily applicable to other types of dynamical systems. To remain in the context of epidemiology, we generalized it to metapopulation dynamics on weighted and multiplex networks [1, 18]. This class of dynamical systems is frequently used to forecast epidemic outbreaks and investigate containment measures in real-world scenarios [2, 19].

On Fig. 1, we show the main results from the metapopulation dynamics experiment on weighted networks. Similarly to the previous experiments with stochastic contagion dynamics, the predictions of the GNN are very accurate with respect to the ground truth dynamics, even though the amount of data needed to train effective models is quite small (1000 time steps with networks of size 100). It can also predict accurately the bifurcation diagram on Poisson networks.

Action taken 2.3

First, we include in the results section a new experiment involving a metapopulation dynamics on weighted networks. Second, we demonstrate the feasibility of using real data to train our models by training a GNN on the dataset of the COVID-19 outbreak in Spain, using the multiplex mobility network. To make room for the latter result, we removed the Fig. 4 of the main text involving the application of our trained models on real temporal networks.

Comment 2.4

Could the framework be used for model selection? E.g. to distinguish between simple and complex contagion or complex and interacting diseases (the problem posed in ref. 9 of the main text). From the manuscript it seems that the authors use specific LTPs for different types of contagion (eqs. 5 for simple and eq. 6 for complex): i.e. they assume the full knowledge of the contagion process that generated the data. What would happen if I use a model trained for complex contagion to analyze data generated from a simple one? And, based on the results, would it be possible to decide which is the underlying process?

Reply 2.4

We think our framework could be used for model selection, we thank Reviewer 2 for mentioning it. To be more precise, suppose a dataset X that we assume was generated on a network G . Furthermore, assume that a model M is a candidate for modeling X on G . The idea here would be to train a GNN model $\hat{M}(\Theta)$ such that the parameters

Θ are learned from X and G , and then compare the predictions of $\hat{M}(\Theta)$ with the predictions from M . In the case where $\hat{M}(\Theta)$ is correctly trained, their predictions should be similarly if M is actually a good candidate, and dissimilar otherwise.

To give a brief example of model selection, we compared the predictions of a standard mass-action metapopulation model for the evolution of the COVID-19 outbreak in Spain, and compared it to the predictions of a trained GNN model (see Fig. 4b of the main paper). The results show that the mass-action model overestimates by far the daily number of newly infected cases, while our GNN predicts it quite accurately. This suggests that the mass-action model, implicitly driven by a simple contagion mechanism, is not appropriate to model the evolution of COVID-19, which in turn suggests that COVID-19 is probably closer to a complex contagion than a simple one.

We stress that this simple comparison is by no means principled, and that further investigation would be needed to substantiate the true feasibility of model selection with our approach. Indeed, our goal is simply to quickly illustrate how some intuitions can be gained from our approach with respect to the propagation of COVID-19.

Action taken 2.4

We mention in the opening of the paper the possibility to perform model selection as a future application.

Comment 2.5

One of the assumptions made by the authors is that the underlying process is stationary. However, for many models and also in real data (both epidemiological or from computational social sciences) this assumption does not hold. This limits the applicability of the framework and should be discussed in the text. Moreover, how stringent is this limitation in practice? Since the model learns the LTPs only from the state of the first neighbors and does not consider dynamical correlations, analyzing data from a SIR dynamics should not be a problem, right?

Reply 2.5

We thank Reviewer 2 for pointing this out. It seems there was a confusion resulting from the nomenclature “stationary process”. By stationary we did not mean that the process needs to reach a stationary distribution or steady state. Instead, we meant that the underlying process is time invariant. Another valid terminology coming from the study of Markov chain would be *time-homogeneous*.

Action taken 2.5

We changed “stationary” for “time invariant” in the main text, where the model’s assumptions are discussed.

Comment 2.6

The part of the text introducing the stars graphs of different sizes while describing Fig.2 is quite confusing. I had to read it three times to understand why the authors are using the k -stars for the comparison and if they have been used in the training or in the testing. Would it not be easier to use other random networks? E.g. with different maximum degrees or sizes?

Reply 2.6

We agree with Reviewer 2 that the star graph analysis is confusing at first glance. The idea was to test the models, after they were trained, on all possible inputs and determine specifically on which kind of inputs the models struggled. In this case, the star graph of $k + 1$ nodes is instrumental to generate any input of node of degree k . Unfortunately, this procedure is hardly generalizable non-discrete state dynamics, where the node states are continuous and/or unbounded, or when real-valued metadata are involved.

We realize that the star graph analysis was more confusing than helpful, but we believe that it still has some value to

precisely assess our model performance.

Action taken 2.6

We remove the star graph analysis completely, and replaced it in the text by a more standard scatter plot representation, from which we can also compute the Pearson coefficient r as a performance measure for each model.

Comment 2.7

From the main text it is hard to understand why the authors use the approximated master equations framework to get the phase diagram of the three processes instead of standard Monte Carlo simulations (reading the SI things become just a bit clearer). Thus, my suggestion is also to clarify this part.

Reply 2.7

The idea was to show that our GNN models can be analyzed through conventional methods, such as approximate master equations, to estimate their dynamical properties (fixed points, stability, critical thresholds, etc.). Whereas the individual parameters of the GNN are hardly interpretable, this demonstrates that the model as the whole has some interpretability to it. As we show in the main text, these frameworks can also be used alongside with Monte Carlo simulations (see Fig. 3 (d–f) and Fig. 3 (j–l) of the old manuscript). However, similarly to the star graph analysis, the applicability of approximated master equations remains limited to discrete state dynamics with a small number of states.

We realize now that the use of approximated master equation in this context was confusing—no explanation was left in the main text—and did not convey the message it was supposed to. We thank Reviewer 2 for pointing this out.

Action taken 2.7

We remove the approximated master equation results from the paper altogether and leave only the results from Monte Carlo simulations.

Response to Referee 3

Comment 3.1

Epidemics on networks have recently been used in multiple fields e.g. spread of disease or fake news through contact or social networks. But how the underlying dynamics of spreading works in nature is not known to us, hence multiple statistical or mechanistic models have been proposed for specific situations and inference schemes for fitting them to the observed dataset.

The two main contributions of the present paper which would be of interest to the community is,

- Consider models of epidemics on networks whose Markovian nature is modeled using a graphical neural network (+attention) rather than an interpretable mechanism.*
- Choose the best among the above models by minimizing an approximate cross-entropy loss.*
- Illustrates that the learned model can show similar behaviours in limit to the true underlying models.*

Reply 3.1

We thank Reviewer 3 for their accurate depiction of our work and to have highlighted these three points of interest for the community.

Comment 3.2

Although I should point that the lack of interpretability of the learnt mechanism may not give us the same amount of scientific intuition/justification we can gain from simpler and more intuitive mechanistic models, which may significantly restrict the adaptation of the models/techniques presented here. In simple words, a domain scientist does not get any scientific knowledge out of the learned model here.

Reply 3.2

We agree with Reviewer 3 that the effective mechanisms—i.e. the parameters of the GNN—learned from data are hard to interpret, as it is the case in most if not all deep learning approaches. It is not our intent to examine the learned weights directly of a GNN.

We believe that this does not necessarily mean that our approach cannot provide no scientific intuition or knowledge altogether. First, we showed that a model can be used to study the bifurcation diagram of the underlying process accurately once it is properly trained, from which tremendous insight can be gained: Is there a phase transition? What is the order of this phase transition? What are the thresholds? Second, we showed that, by comparing the predictions of our GNN model and the predictions another mechanistic model, insights can be gained about the complex nature of the COVID-19 contagion dynamics. This suggests that we can investigate accurately the relationship between the states of the nodes and the structure of the network, and more precisely the effects of the network structural properties on the dynamical outcomes.

Furthermore, even though mechanistic models are extremely useful to gain some intuition about real phenomena, they often need to be very complicated in order to be used alongside real data. As more mechanisms are added, they often lose interpretability, and it is at this point that effective mechanisms become more appealing [20]. We believe our approach to be more like a way of building effective mechanisms directly from real data which can be investigated as we please—a top-bottom type of approach—, rather than a way to gain insight at the microscopic level—a bottom-up

type of approach.

We realize that the interpretability of our approach, which could play a big role in deciding whether to use it or not, was not discussed in detail in the main text. We thank Reviewer 3 for pointing this out.

Action taken 3.2

We added in the main text a comment on the interpretability of our approach and how it can be used to build effective mechanistic models.

Comment 3.3

Unfortunately the present paper does not comment on the prediction performance of the strategy proposed for epidemics compared to the traditionally used epidemic models.

Reply 3.3

From the analysis that we presented, it was hard to include a comparison between our approach and other epidemiological approaches, because we did not have access to real datasets to test both types of hypotheses. It is true that it would benefit greatly from such demonstration.

As we mentioned in Reply 2.4 to Reviewer 2, we believe that our approach can be used for model selection. We illustrate this by comparing the predictions a GNN trained on the dataset of the COVID-19 outbreak in Spain, with the predictions of a standardly used mass-action metapopulation model whose parameters have been chosen to suit the specifics of COVID-19 [2]. As discussed above, our analysis suggests that simple contagion models are inappropriate to model the propagation of this disease over long period of time where containment measures have been implemented.

We wish to emphasize that this metapopulation model nor our GNN model are state-of-arts in this specific context. The intent of our work remains a proof of concept, rather than a dedicated study of the COVID-19 outbreak in Spain.

Comment 3.4

Further, one of the main interests in the epidemics community is the optimal intervention mechanism - which is also not very clear here, how could we learn that for an 'non-interpretable' model like this?

Reply 3.4

As we mentioned in Reply 3.2, even though the parameters of our models are not interpretable, they can still be used in a top-bottom kind of way.

Action taken 3.4

Along the same lines as Reply 3.3, we include a comment regarding the possibility of designing intervention procedures using our approach.

Comment 3.5

Saying all of the above, I would say the paper has novelty but may not be appropriate for the present venue and the paper can be further improved by adding results regarding the prediction performance or how intervention strategy can interact with these types of models.

Reply 3.5

We thank Reviewer 3 for acknowledging the novelty of our work and hope that the substantial modifications we applied to the paper satisfied their requirements.

-
- [1] V. Colizza, R. Pastor-Satorras, and A. Vespignani, “Reaction–diffusion processes and metapopulation models in heterogeneous networks,” *Nat. Phys.* **3**, 276–282 (2007).
 - [2] A. Aleta and Y. Moreno, “Evaluation of the potential incidence of COVID-19 and effectiveness of containment measures in Spain: A data-driven approach,” *BMC Med.* **18**, 157 (2020).
 - [3] E. Dong, H. Du, and L. Gardner, “An interactive web-based dashboard to track COVID-19 in real time,” (2020).
 - [4] W. L. Hamilton, R. Ying, and J. Leskovec, “Representation Learning on Graphs: Methods and Applications,” (2017), arXiv:1709.05584.
 - [5] J. Zhou, Gé Cui, Z. Zhang, C. Yang, Z. Liu, L. Wang, C. Li, and M. Sun, “Graph Neural Networks: A Review of Methods and Applications,” (2018), arXiv:1812.08434.
 - [6] Z. Zhang, P. Cui, and W. Zhu, “Deep Learning on Graphs: A Survey,” (2018), arXiv:1812.04202.
 - [7] I. Chami, S. Abu-El-Haija, B. Perozzi, C. Ré, and K. Murphy, “Machine Learning on Graphs: A Model and Comprehensive Taxonomy,” (2020), arXiv:2005.03675.
 - [8] K. Xu, W. Hu, J. Leskovec, and S. Jegelka, “How Powerful are Graph Neural Networks?” (2018), arXiv:1810.00826.
 - [9] S. L. Brunton, J. L. Proctor, and J. N. Kutz, “Discovering governing equations from data by sparse identification of nonlinear dynamical systems,” *Proc. Natl. Acad. Sci. USA* **113**, 3932–3937 (2016).
 - [10] J. N. Kutz, “Deep learning in fluid dynamics,” *J. Fluid Mech.* **814**, 1–4 (2017).
 - [11] J. Pathak, Z. Lu, B. R. Hunt, M. Girvan, and E. Ott, “Using machine learning to replicate chaotic attractors and calculate Lyapunov exponents from data,” *Chaos* **27**, 121102 (2017).
 - [12] J. Pathak, B. Hunt, M. Girvan, Z. Lu, and E. Ott, “Model-Free Prediction of Large Spatiotemporally Chaotic Systems from Data: A Reservoir Computing Approach,” *Phys. Rev. Lett.* **120**, 024102 (2018).
 - [13] B. M. de Silva, D. M. Higdon, S. L. Brunton, and J. N. Kutz, “Discovery of Physics From Data: Universal Laws and Discrepancies,” *Front. Artif. Intell.* **3**, 25 (2020).
 - [14] B. Lusch, J. N. Kutz, and S. L. Brunton, “Deep learning for universal linear embeddings of nonlinear dynamics,” *Nat. Commun.* **9**, 1–10 (2018).
 - [15] F. A. Rodrigues, T. Peron, C. Connaughton, J. Kurths, and Y. Moreno, “A machine learning approach to predicting dynamical observables from network structure,” (2019), arXiv:1910.00544.
 - [16] A. Salova, J. Emenheiser, A. Rupe, J. P. Crutchfield, and R. M. D’Souza, “Koopman operator and its approximations for systems with symmetries,” *Chaos* **29**, 093128 (2019).
 - [17] C. Shah, N. Dehmamy, N. Perra, M. Chinazzi, A.-L. Barabási, A. Vespignani, and R. Yu, “Finding Patient Zero: Learning Contagion Source with Graph Neural Networks,” (2020), arXiv:2006.11913.
 - [18] D. Balcan, B. Gonçalves, H. Hu, J. J. Ramasco, V. Colizza, and A. Vespignani, “Modeling the spatial spread of infectious diseases: The global epidemic and mobility computational model,” *J. Comput. Sci.* **1**, 132–145 (2010).
 - [19] L. Hébert-Dufresne, A. Allard, J.-G. Young, and L. J. Dubé, “Global efficiency of local immunization on complex networks,” *Sci. Rep.* **3**, 2171 (2013).
 - [20] L. Hébert-Dufresne, S. V. Scarpino, and J.-G. Young, “Interacting contagions are indistinguishable from social reinforcement,” *arXiv* (2019), arXiv:1906.01147.

Deep learning of contagion dynamics on complex networks

Charles Murphy, Edward Laurence, and Antoine Allard

*Département de Physique, de Génie Physique, et d'Optique,
Université Laval, Québec (Québec), Canada G1V 0A6 and
Centre interdisciplinaire en modélisation mathématique,
Université Laval, Québec (Québec), Canada G1V 0A6*

(Dated: January 14, 2021)

Forecasting the evolution of contagion dynamics is still an open problem to which mechanistic models only offer a partial answer. To remain mathematically or computationally tractable, these models must rely on simplifying assumptions, thereby limiting the quantitative accuracy of their predictions and the complexity of the dynamics they can model. Here, we propose a complementary approach based on deep learning where the effective local mechanisms governing a dynamic are learned from time series data. Our graph neural network architecture makes very few assumptions about the dynamics, and we demonstrate its accuracy using different contagion dynamics of increasing complexity. By allowing simulations on arbitrary network structures, our approach makes it possible to explore the properties of the learned dynamics beyond the training data. Finally, we illustrate the applicability of our approach using real data of the COVID-19 outbreak in Spain from January to October 2020. Our results demonstrate how deep learning offers a new and complementary perspective to build effective models of contagion dynamics on networks.

I. INTRODUCTION

Our capacity to prevent or contain outbreaks of infectious diseases is directly linked to our ability to accurately model contagion dynamics. Since the seminal work of Kermack and McKendrick almost a century ago [1], a variety of models incorporating ever more sophisticated contagion mechanisms has been proposed [2–5]. These mechanistic models have provided invaluable insights about how infectious diseases spread, and have thereby contributed to the design of better public health policies. However, several challenges remain unresolved, which call for contributions from new modeling approaches [6–8].

For instance, many complex contagion processes involve the nontrivial interaction of several pathogens [9–12], and some social contagion phenomena, like the spread of misinformation, require to go beyond pairwise interactions between individuals [13–15]. Also, while qualitatively informative, the forecasts of most mechanistic models lack quantitative accuracy [16]. Indeed, most models are constructed from a handful of mechanisms which can hardly reproduce the intricacies of real complex contagion dynamics. One approach to these challenges is to complexify the models by adding more detailed and sophisticated mechanisms. However, mechanistic models become rapidly intractable as new mechanisms are added. Moreover, models with higher complexity require the specification of a large number of parameters whose values can be difficult to infer from limited data.

There has been a recent gain of interest towards using machine learning to address the issue of the often limiting complexity of mechanistic models [12, 17–23]. This new kind of approach aims at training predictive models directly from observational time series data. These data-driven models are then used for various tasks such as making accurate predictions [19, 21], gaining useful intuitions about complex phenomena [12] and discovering new patterns from which better mechanisms can be designed [17, 18]. While these approaches were originally designed for regularly structured

data, this new paradigm is now being applied to epidemics spreading on networked systems [24, 25], and more generally to dynamical systems [26–28]. Meanwhile, the machine learning community has dedicated a considerable amount of attention on deep learning on graphs, structure learning and graph neural networks (GNN) [29–31]. Recent works showed great promise for GNN in the context of community detection [32], link prediction [33], network inference [34], as well as in the context of discovering new materials and drugs [35, 36]. These advances suggest that GNN could be prime candidates for building effective data-driven dynamical models on networks.

In this paper, we show how GNN, usually used for structure learning, can also be used to model contagion dynamics on complex networks. Our contribution is threefold. First, we design a training procedure and an appropriate GNN architecture capable of representing a wide range of dynamics, with very few assumptions. Second, we demonstrate the validity of our approach using various contagion dynamics on networks of different natures, with increasing complexity as well as on real epidemiological data. Finally, we show how our approach can provide predictions for previously unseen network structures, therefore allowing the exploration of the properties of the learned dynamics beyond the training data. Our work generalizes the idea of constructing dynamical models from regularly structured data to arbitrary network structures, and suggests that our approach could be accurately extended to many other classes of dynamical processes.

II. RESULTS

In our approach, we assume that an unknown dynamical process, denoted \$\mathcal{M}\$, takes place on a known network structure—or ensemble of networks—, denoted \$G = (\mathcal{V}, \mathcal{E}; \Phi, \Omega)\$, where \$\mathcal{V} = {v_1, \dots, v_N}\$ is the node set and \$\mathcal{E} = {e_{ij} | v_j \text{ is connected to } v_i \wedge (v_i, v_j) \in \mathcal{V}^2}\$ is the edge set. We also assume that the network(s) can have some

metadata, taking the form of node and edge attributes denoted $\Phi_i = (\phi_1(v_i), \dots, \phi_Q(v_i))$ for node v_i and $\Omega_{ij} = (\omega_1(e_{ij}), \dots, \omega_P(e_{ij}))$ for edge e_{ij} , respectively, where $\phi_q : \mathcal{V} \rightarrow \mathbb{R}$ and $\omega_p : \mathcal{E} \rightarrow \mathbb{R}$. These metadata can take various forms like node characteristics or edge weights. We also denote the node and edge attribute matrices $\Phi = (\Phi_i | v_i \in \mathcal{V})$ and $\Omega = (\Omega_{ij} | e_{ij} \in \mathcal{E})$, respectively.

Next, we assume that \mathcal{M} has generated a time series D on G . This time series takes the form of a pair of consecutive snapshots $D = (\mathbf{X}, \mathbf{Y})$ with $\mathbf{X} = (X_1, \dots, X_T)$ and $\mathbf{Y} = (Y_1, \dots, Y_T)$, where $X_t \in \mathcal{S}^{|\mathcal{V}|}$ is the state of the nodes at time t , $Y_t \in \mathcal{R}^{|\mathcal{V}|}$ is the outcome of \mathcal{M} defined as

$$\mathbf{Y}_t = \mathcal{M}(\mathbf{X}_t, \mathbf{G}), \quad (1)$$

\mathcal{S} is the set of possible node states, and \mathcal{R} is the set of possible node outcomes. This way of defining D allows us to formally concatenate multiple realizations of the dynamics in a single dataset. Additionally, the elements $x_i(t) \equiv (X_t)_i$ and $y_i(t) \equiv (Y_t)_i$ correspond to the state of node v_i at time t and its outcome, respectively. Typically, we consider that the outcome $y_i(t)$ is simply the state of node v_i after transitioning from state $x_i(t)$. In this case, we have $\mathcal{S} = \mathcal{R}$ and $x_i(t + \Delta t) = y_i(t)$ where Δt is the length of the time steps. However, if \mathcal{S} is a discrete set—i.e. finite and countable— $y_i(t)$ is a transition probability vector conditioned on $x_i(t)$ from which the following state, $x_i(t + \Delta t)$, will be sampled. The element $(y_i(t))_m$ corresponds to the probability that node v_i evolves to state $m \in \mathcal{S}$ given that it was previously in state $x_i(t)$ —i.e. $\mathcal{R} = [0, 1]^{|\mathcal{S}|}$. Since, in the case \mathcal{M} is a stochastic dynamics, we do not typically have access to the transition probabilities $y_i(t)$ directly but to the observed outcome state—e.g. $x_i(t + \Delta t)$ in the event where \mathbf{X} is temporally ordered—we define the observed outcome $\tilde{y}_i(t)$ as

$$(\tilde{\mathbf{y}}_i(\mathbf{t}))_{\mathbf{m}} = \delta(\mathbf{x}_i(\mathbf{t} + \Delta \mathbf{t}), \mathbf{m}), \quad \forall \mathbf{m} \in \mathcal{S} \quad (2)$$

where $\delta(x, y)$ is the Kronecker delta. Finally, we assume that \mathcal{M} acts on X_t locally and identically at all times, according to the structure of G . In other words, knowing the state x_i as well as the states of all the neighbors of v_i , the outcome y_i is computed using a time independent function f identical for all nodes:

$$\mathbf{y}_i \equiv \mathbf{f}(\mathbf{x}_i, \Phi_i, \mathbf{x}_{\mathcal{N}_i}, \Phi_{\mathcal{N}_i}, \Omega_{i\mathcal{N}_i}), \quad (3)$$

where $\mathbf{x}_{\mathcal{N}_i} = (x_j | v_j \in \mathcal{N}_i)$ denotes the states of the neighbors, $\mathcal{N}_i := \{v_j | e_{ij} \in \mathcal{E}\}$ is the set of the neighbors, $\Phi_{\mathcal{N}_i} := \{\Phi_j | v_j \in \mathcal{N}_i\}$ and $\Omega_{i\mathcal{N}_i} := \{\Omega_{ij} | v_j \in \mathcal{N}_i\}$. As a result, we impose a notion of locality where the underlying dynamics is time invariant and invariant under the permutation of the node labels in G , under the assumption that the node and edge attributes are left invariant.

Our objective is to build a model $\hat{\mathcal{M}}$, parametrized by a GNN with a set of tunable parameters Θ , trained on the observed dataset D to mimic \mathcal{M} given G , that is

$$\hat{\mathcal{M}}(\mathbf{X}'_t, \mathbf{G}'; \Theta) \approx \mathcal{M}(\mathbf{X}'_t, \mathbf{G}'), \quad (4)$$

for all states \mathbf{X}'_t and all networks \mathbf{G}' . The architecture of $\hat{\mathcal{M}}$, detailed in Sec. V A, is designed to act locally similarly to \mathcal{M} .

FIG. 1. (color online) **Predictions of GNN trained on a Barabási-Albert random network (BA) [38] for the (a) simple and (a) complex contagion dynamics.** The solid and dashed lines correspond to the transition probabilities of the dynamics used to generate the training data (labeled GT for “ground truth”), and predicted by the GNN, respectively. Markers correspond to the maximum likelihood estimation (MLE) of the transition probabilities computed from the dataset D . The colors indicate the type of transition: infection ($S \rightarrow I$) in blue and recovery ($S \rightarrow R$) in red. The standard deviations, as a result of averaging the outcomes given ℓ , are shown using a colored area around the lines (typically narrower than the width of the lines) and using vertical bars for the markers.

In this case, the locality is imposed by a modified attention mechanism inspired by Ref. [37]. The advantage of imposing locality allows our architecture to be *inductive*: If the GNN is trained on a wide range of local structures—i.e. nodes with different neighborhood sizes (or degrees) and states—it can then be used on any other networks within that range. This suggests that the topology of G will have a strong impact on the quality of the trained models, an intuition that is confirmed below. Similarly to Eq. (3), we can write each individual node outcome computed by the GNN using a function \hat{f} such that

$$\hat{\mathbf{y}}_i \equiv \hat{\mathbf{f}}(\mathbf{x}_i, \Phi_i, \mathbf{x}_{\mathcal{N}_i}, \Phi_{\mathcal{N}_i}, \Omega_{i\mathcal{N}_i}; \Theta) \quad (5)$$

where \hat{y}_i is the outcome of node v_i predicted by $\hat{\mathcal{M}}$.

The objective described by Eq. (4) must be encoded into a global loss function, denoted $\mathcal{L}(\Theta)$.

Like the outcome functions, $\mathcal{L}(\Theta)$ can be decomposed locally, where the local losses of each node $L(y_i, \hat{y}_i)$ are arithmetically averaged over all possible node inputs $(x_i, \Phi_i, \mathbf{x}_{\mathcal{N}_i}, \Phi_{\mathcal{N}_i}, \Omega_{i\mathcal{N}_i})$, where

y_i and \hat{y}_i are given by Eqs (3) and (5), respectively. By using an arithmetic mean to the evaluation of $\mathcal{L}(\Theta)$, we assume that the node inputs are distributed uniformly. Consequently, the model should be trained equally well on all of them. This is important because in practice we only have access to a finite number of inputs in D and G , for which the node input distribution is typically far from being uniform. Hence, in order to train effective models, we recalibrate the inputs using the following global loss

$$\mathcal{L}(\Theta) = \sum_{t \in \mathcal{T}'} \sum_{\mathbf{v}_i \in \mathcal{V}'(t)} \frac{\mathbf{w}_i(\mathbf{t})}{Z'} \mathbf{L}(\mathbf{y}_i(\mathbf{t}), \hat{\mathbf{y}}_i(\mathbf{t})) \quad (6)$$

where $w_i(t)$ is a weight assigned to node v_i at time t , and $Z' = \sum_{t \in \mathcal{T}'} \sum_{v_i \in \mathcal{V}'(t)} w_i(t)$ is a normalization factor. Here, the training node set $\mathcal{V}'(t) \subseteq \mathcal{V}$ and the training time set $\mathcal{T}' \subseteq [1, T]$ allow us to partition the training dataset for validation and testing when required.

The choice of weights needs to reflect the importance of each node at each time. Because we wish to lower the influence of overrepresented inputs and increase that of rare inputs, a sound choice of weights is

$$\mathbf{w}_i(\mathbf{t}) \propto \rho\left(\mathbf{k}_i, \mathbf{x}_i, \Phi_i, \mathbf{x}_{\mathcal{N}_i}, \Phi_{\mathcal{N}_i}, \Omega_{i\mathcal{N}_i}\right)^{-\lambda} \quad (7)$$

where k_i is the degree of node v_i in G , $0 \leq \lambda \leq 1$ is an hyperparameter. Equation (7) is an ideal choice, because it corresponds to a principled importance sampling approximation of Eq. (6) [39], which is relaxed via the exponent λ . We obtain a pure importance sampling scheme when $\lambda = 1$. Note that the weights can rarely be exactly computed using Eq.

(7), because the distribution ρ is typically computationally intensive to obtain from data, especially for continuous \mathcal{S} with metadata. We illustrate various ways to evaluate the weights in Sec. V B and in Supplementary Material.

We now illustrate the accuracy of our approach by applying it to four types of synthetic dynamics of various natures (see Sec. V C for details on the dynamics). We first consider a *simple contagion* dynamics: The discrete-time susceptible-infected-susceptible (SIS) dynamics. In this dynamics, nodes are either susceptible (S) or infected (I)

by some disease, i.e. $\mathcal{S} = \{S, I\} = \{0, 1\}$, and transition between each state stochastically according to an infection probability function $\alpha(\ell)$, where ℓ is the number of infected neighbors of a node, and a constant recovery probability β . A notable feature of simple contagion dynamics is that susceptible nodes get infected by the disease through their infected neighbors independently. This reflects the assumption that disease transmission behaves identically whether a person has a large number of infected neighbors or not.

Second, to relax this assumption we consider a *complex contagion* dynamics with a nonmonotonic infection function $\alpha(\ell)$ where the aforementioned transmission events are no longer independent [14]. This contagion dynamics has an interesting interpretation in the context of the propagation of a social behavior, where the local popularity of a behavior (large ℓ)

hinders its adoption. The independent transmission assumption can also be lifted when multiple diseases are interacting [10]. Thus, we also consider an asymmetric *interacting contagion* dynamics with two diseases. In this case,

$\mathcal{S} = \{S_1 S_2, I_1 S_2, S_1 I_2, I_1 I_2\} = \{0, 1, 2, 3\}$ where $U_1 V_2$ corresponds to a state where a node is in state U with respect to the first disease and in state V with respect to the second disease. The interaction between the diseases occurs through a coupling that is active only when a node is infected by at least one disease, otherwise it behaves identically to the simple contagion dynamics. This coupling might increase or decrease the virulence of the other disease.

Whereas the previously presented dynamics capture various features of contagion phenomena, real datasets containing this level of detail about the interactions among individuals are rare [40–42]. A class of dynamics for which dataset are easier to find is that of mass-action *metapopulation* dynamics [43–46], where the status of the individuals are gathered by geographical regions. These dynamics typically evolve on the weighted networks of the individuals' mobility between regions and the state of a region consists in the number of people that are in each individual health state. As our final case, we investigate a type of deterministic metapopulation dynamics where the population size is constant and where people can either be susceptible (S), infected (I) or recovered from the disease (R). As a result, we define the state of the node as three-dimensional vectors specifying the fraction of people in each state—i.e. $\mathcal{S} = \mathcal{R} = [0, 1]^3$.

Figure 1 shows the GNN predictions for the infection and recovery probabilities of the simple and complex contagion dynamics as a function of the number of infected neighbors ℓ . We then compare them with their ground truths, i.e. Eq. (20) using Eqs. (18)–(22) for the infection functions. We also show the maximum likelihood estimators (MLE) of the transition probabilities computed from the fraction of nodes in state x and with ℓ infected neighbors that transitioned to state y in the complete dataset D . The MLE, which are typically used in this kind of inference problem [47], stands as a reference to benchmark the performance of our approach.

We find that the GNN learns remarkably well the transition probabilities of the simple and complex contagion dynamics. In fact, the predictions of the GNN seem to be systematically smoother than the ones provided by the MLE. This is because the MLE is computed for each individual pair (x, ℓ) from disjoint subsets of the training dataset. This implies that a large number of samples of each pair (x, ℓ) is needed for the MLE to be accurate; a condition rarely met in realistic settings, especially for high degree nodes. This also means that the MLE cannot be used directly to interpolate beyond the pairs (x, ℓ) present in the training dataset, in sharp contrast with the GNN which, by definition, can interpolate within the dataset D . Furthermore, all of its parameters are hierarchically involved during training, meaning that the GNN benefits from any sample to improve all of its predictions, which are smoother and more consistent.

It is worth mentioning that the GNN is not specifically designed nor trained to compute the transition probabilities as a function of a single variable, namely the number of infected ℓ . In reality, the GNN computes its outcome from the complete multivariate state of the neighbors of a node. The interacting contagion and the metapopulation dynamics, unlike the simple and complex contagions, are examples of such multi-

FIG. 2. (color online) **Comparison between the targets and the predictions of GNN trained on Erdős-Rényi networks (ER, top row) and on Barabási-Albert networks [38] (BA, middle row) for the (a, e, i) simple, (b, f, j) complex, (c, g, k) interacting and (d, h, l) metapopulation dynamics.** Each point shown on the panels (a–h) corresponds to a different pair $(y_i(t), \hat{y}_i(t))$ in the complete dataset D . We also indicate the Pearson coefficient r on each panel to measure the correlation between the predictions and the targets and use it as a global performance measure. The panels (i–l) show the errors $(1 - r)$ as a function of the number of neighbors for GNN trained on ER and BA networks, and those of the corresponding MLE. These errors are obtained from the Pearson coefficients computed from subsets of the prediction-target pairs where all nodes have degree k .

variate cases. Their outcome is thus harder to visualize in a representation similar to Fig. 1. Figures 2(a–h) addresses this issue by comparing each of the GNN predictions $\hat{y}_i(t)$ with its corresponding target $y_i(t)$ in the dataset D . We quantify the global performance of the models in different scenarios, for the different dynamics and underlying network structures, using the Pearson correlation coefficient r between the predictions and targets (see Sec. V A). We also compute the error, defined from the Pearson coefficient as $1 - r$, for each degree class k , i.e. between the predictions and targets of only the nodes of degree k . This allows us to quantify the GNN performance for every local structure.

Figures 2(i–k) confirm that the GNN provides more accurate predictions than the MLE in general and across all degrees. This is especially true in the case of the interacting contagion, where the accuracy of the MLE seems to deteriorate rapidly for large degree nodes. This is a consequence of how scarce the inputs are for this dynamics compared to both the simple and complex contagion dynamics for training datasets of the same size, and of how fast the size of the set of possible inputs scales, thereby quickly rendering MLE completely ineffective for small training datasets. The GNN, on the other hand, is less affected by the scarcity of the data, since any

sample improves its global performance, as discussed above.

Figure 2 also exposes the crucial role of the network G on which the dynamics evolves in the global performance of the GNN. Namely, the heterogeneous degree distributions of Barabási-Albert networks (BA)—or any scale-free network for that matter—offer, without surprise, a wider range of degrees than those of homogeneous Erdős-Rényi networks (ER). We can take advantage of this heterogeneity to train GNN models that generalize well across a larger range of local structures, as seen on Fig. 2(i–l) (see also Supplementary Material). Although, for low degrees, the predictions on BA networks are not systematically always better than those on ER networks as shown in the interacting and metapopulation cases. This nonetheless suggests a wide applicability of our approach for real complex systems, whose underlying network structures recurrently exhibit a heterogeneous degree distribution [48].

We test the trained GNN on unseen network structures by recovering the bifurcation diagrams of the contagion and metapopulation dynamics. In the infinite-size limit $|\mathcal{V}| \rightarrow \infty$, these dynamics have two possible long-term outcomes: the absorbing state where the diseases quickly die out, and the endemic/epidemic state in which a macroscopic fraction of

FIG. 3. **Bifurcation diagrams of the (a) simple, (b) complex, (c) interacting and (d) metapopulation dynamics on Poisson networks [38] composed of $|\mathcal{V}| = 2000$ nodes with different average degrees $\langle k \rangle$.** The prevalence is defined as the average fraction of nodes that are asymptotically infected by at least one disease and the outbreak size corresponds to the average fraction of nodes that have recovered. Those are computed from numerical simulations using the “ground truth” (GT) dynamics (circles in blue) and the GNN trained on Barabási-Albert networks (triangles in orange). The error bars correspond to the standard deviations of these numerical simulations. The trained GNN used are the same ones as those used for Fig. 2. As a reference, we also indicate with dashed lines the value(s) of average degree $\langle k \rangle$ corresponding to the network(s) on which the GNN were trained. On panel (d), more than one value of $\langle k \rangle$ appear as multiple networks with different average degrees were used to train the GNN.

nodes remains (endemic) or has been infected over time (epidemic) [4, 10, 49]. These possible long-term outcomes exchange stability during a phase transition which is continuous for the simple contagion and metapopulation dynamics, and discontinuous for the complex and interacting contagion dynamics. The position of the phase transition depends on the parameters of the dynamics as well as on the topology of the network. Note that for the interacting contagion dynamics, the stability of absorbing and endemic states do not change at the same point, giving rise to a bistable regime where both states are stable.

Figure 3 shows the different bifurcation diagrams obtained by performing numerical simulations with the trained GNN models [using Eq. (5)] while varying the average degree of networks on which the GNN has not been trained. Quantitatively, the predictions are also strikingly accurate—essentially perfect for the simple and complex contagion dynamics—which is remarkable given that the bifurcation diagrams were obtained on networks the GNN had never seen before. On the one hand, it illustrates how insights can be gained about the underlying process concerning the existence of phase transitions and their order, among other things. On the other hand, it suggests how the GNN can be used for diverse applications, such as predicting the dynamics under various network structures (e.g. designing intervention strategies that affect the way individuals interact and are therefore connected).

Finally, to further substantiate the applicability of our approach, we illustrate the performance of a GNN model trained on a dataset of the COVID-19 outbreak in Spain from March to October 2020 (see Sec. VD for further details). As seen in Fig. 4(c–k), the performance results indicate that the model predicts correctly the number of susceptible and recovered people. Additionally, the GNN predictions for the number of infected people are highly correlated with the ground truth, even though they are far from perfectly accurate, which may be due to the low prevalence of the disease in the dataset. In

Fig. 4(b), we also compare the predictions of the GNN with

a metapopulation model with a simple contagion infection mechanism (see Sec. VC4). The parameters of the metapopulation model, which have been fixed to fit the specifics of the COVID-19 disease (see Sec. VD), have also been used in a previous work to evaluate the efficacy of the containment measures in Spain [50]. Interestingly, the GNN seems to predict correctly the daily outbreak size, whereas the metapopulation model overestimate by far the outbreak size. This could be due to the social distancing and confinement measures in place during that period that fortunately diminished dramatically the observed number of cases. In fact, the interaction between the epidemic and the social dynamics (e.g. confinements) generated a dynamical process that, unlike the metapopulation model, is probably closer to a complex contagion phenomena than to a simple one [12]. In turn, our results suggest that simple contagion models are not appropriate to accurately predict the evolution of the COVID-19 pandemic over a long period of time, and that complex contagion models could provide better effective models.

III. DISCUSSION

We introduced a data-driven approach that learns effective mechanisms governing the propagation of diverse dynamics on complex networks. We proposed a reliable training protocol, and we validated the projections of our GNN architecture on simple, complex, interacting contagion and metapopulation dynamics using synthetic networks. Interestingly, we found that our approach performs better when trained on data whose underlying network structure is heterogeneous, which could prove useful in real-world applications of our method given the ubiquitousness of scale-free networks [51].

By recovering the bifurcation diagram of various dynamics, we illustrated how our approach can leverage time series from an unknown dynamical process to gain insights about its properties—e.g. the existence of a phase transition and its or-

FIG. 4. (color online) **GNN Training of the Spain COVID-19 outbreak dataset on the multiplex weighted network of people mobility between Spain 52 provinces:** (a) Spain mobility multiplex network, (b) daily predictions of the number of infected people, (c–k) detailed performance scatter plots for the training (c–f–i), validation (d–g–j) and testing (e–h–k) datasets. In all panels, the ground truth (GT) values correspond to the values as they appear in the dataset. On panel (a), the thickness of the edges are proportional to the average number of people transitioning between all connected node pairs, and the size of the nodes are proportional to the size of the population N_i within the corresponding province. On panel (b), we show the evolution of the GT outbreak size (solid line) and compare it with the outbreak size predicted by the GNN (dashed line) and an equivalent metapopulation model (dash-dotted line). The predictions, rather than being trajectories, are computed using the GT states of the previous time step, which explains the seemingly inconsistent decrease of the outbreak size over time predicted by the metapopulation model. This is due to a systematic overestimation of the metapopulation model with respect to the GT. We use different scales for the GT/GNN predictions (blue axis on the left) and metapopulation model ones (red axis on the right) to improve the visualization. On panels (c–k), we show the accuracy as obtained from comparing the target and prediction outcomes. We use the number of people in the different states to represent these outcomes, that we compute from the product of the population sizes N_i with the GT and the GNN outcomes— $y_i(t)$ and $\hat{y}_i(t)$, respectively, which we recall correspond to probability vectors of the fraction of nodes in each state. We also separated the target-prediction pairs in columns with respect to the states susceptible (c–e), infected (f–h) and recovered (i–k).

der. We have also shown how to use this framework on real datasets, which in turn could then be used to help build better effective models. In a way, we see this approach as the equivalent of a numerical Petri dish—offering a new way to experiment and gain insights about an unknown dynamics—that is complementary to traditional mechanistic modeling to design better intervention procedures, containment countermeasures and to perform model selection.

Although we focused the presentation of our method on contagion dynamics, its potential applicability reaches many other realms of complex systems modeling where intricate mechanisms are at play. We believe this work establishes solid foundations for the use of deep learning in the design of realistic effective models of complex systems.

IV. ACKNOWLEDGEMENTS AND SUPPORT

The authors are grateful to Emily M. Cyr and Guillaume St-Onge for their comments and to Vincent Thibeault, Xavier Roy-Pomerleau, François Thibault, Patrick Desrosiers, Louis J. Dubé, Simon Hardy and Laurent Hébert-Dufresne for fruitful discussions. They also want to thank the anonymous reviewers for their insightful comments which led to substantial improvements. This work was supported by the Sentinelle Nord initiative from the Canada First Research Excellence Fund (CM, EL, AA), the Conseil de recherches en sciences naturelles et en génie du Canada (CM, AA) and the Fonds de recherche du Québec-Nature et technologie (EL).

V. MATERIAL AND METHODS

A. Graph neural network and training details

In this section, we briefly present our GNN architecture, the training settings, the synthetic data generation procedure and the hyperparameters used in our experiments.

1. Architecture

We use the GNN architecture shown in Fig. 5 and detailed in Tab. I. First, we transform the state x_i of every node with a shared multilayer perceptron (MLP), denoted $\hat{f}_{\text{in}} : \mathcal{S} \rightarrow \mathbb{R}^d$ where d is the resulting number of node features, such that

$$\xi_{\mathbf{i}} = \hat{\mathbf{f}}_{\text{in}}(\mathbf{x}_{\mathbf{i}}). \quad (8)$$

We concatenate the node attributes Φ_i to x_i , when these attributes are available, in which case $\hat{f}_{\text{in}} : \mathcal{S} \times \mathbb{R}^Q \rightarrow \mathbb{R}^d$. At this point, ξ_i is a vector of features representing the state (and attributes) of node v_i . Then, we aggregate the features of the first neighbors using a modified attention mechanism \hat{f}_{att} , inspired by Ref. [37] (see Sec. V A 2),

$$\nu_{\mathbf{i}} = \hat{\mathbf{f}}_{\text{att}}(\xi_{\mathbf{i}}, \xi_{\mathcal{N}_{\mathbf{i}}}), \quad (9)$$

where we recall that $\mathcal{N}_i = \{v_j | e_{ij} \in \mathcal{E}\}$ is the set of nodes connected to node v_i . We also include the edge attributes Ω_{ij} into the attention mechanism, when they are available. To do so, we transform the edge attributes Ω_{ij} into abstract edge features, such that $\psi_{ij} = \hat{f}_{\text{edge}}(\Omega_{ij})$ where $\hat{f}_{\text{edge}} : \mathbb{R}^P \rightarrow \mathbb{R}^{d_{\text{edge}}}$ is also a MLP, before they are used in the aggregation. Finally, we compute the outcome \hat{y}_i of each node v_i with another MLP $\hat{f}_{\text{out}} : \mathbb{R}^d \rightarrow \mathcal{R}$ such that

$$\hat{\mathbf{y}}_{\mathbf{i}} = \hat{\mathbf{f}}_{\text{out}}(\nu_{\mathbf{i}}). \quad (10)$$

2. Attention Mechanism

We use an attention mechanism inspired by the graph attention network architecture (GAT) [37]. The attention mechanism consists of three trainable functions $\mathcal{A} : \mathbb{R}^d \rightarrow \mathbb{R}$, $\mathcal{B} : \mathbb{R}^d \rightarrow \mathbb{R}$ and $\mathcal{C} : \mathbb{R}^{d_{\text{edge}}} \rightarrow \mathbb{R}$, that combine the feature vectors ξ_i , ξ_j and ψ_{ij} of a connected pair of nodes v_i and v_j , where we recall that d and d_{edge} are the number of node and edge features, respectively. Then, the attention coefficient a_{ij} is computed as follows:

$$\mathbf{a}_{\mathbf{ij}} = \sigma \left[\mathcal{A}(\xi_{\mathbf{i}}) + \mathcal{B}(\xi_{\mathbf{j}}) + \mathcal{C}(\psi_{\mathbf{ij}}) \right] \quad (11)$$

where $\sigma(x) = [1 + e^{-x}]^{-1}$ is the logistic function. Notice that, by using this logistic function, the value of the attention coefficients is constrained to the open interval $(0, 1)$, where $a_{ij} = 0$ implies that the state of v_j has no effect on the outcome of v_i and $a_{ij} = 1$ implies the effect is maximal. As a

result, a_{ij} quantifies the influence of the state of node v_j over the outcome of node v_i , which also adds to the interpretability of our model. We compute the aggregated feature vectors ν_i of node v_i such that

$$\nu_{\mathbf{i}} = \hat{\mathbf{f}}_{\text{att}}(\xi_{\mathbf{i}}, \xi_{\mathcal{N}_{\mathbf{i}}}) = \xi_{\mathbf{i}} + \sum_{v_{\mathbf{j}} \in \mathcal{N}_{\mathbf{i}}} \mathbf{a}_{\mathbf{ij}} \xi_{\mathbf{j}}. \quad (12)$$

It is important to stress that, at this point, ν_i contains some information about v_i and all of its neighbors in a pairwise manner. In all our experiments, we fix \mathcal{A} , \mathcal{B} , and \mathcal{C} to be affine transformations with trainable weight matrix and bias vector. Also, we use multiple attention modules in parallel to increase the expressive power of the GNN architecture, as suggested by Ref. [37].

The attention mechanism described by Eq. (11) is slightly different from the vanilla version of Ref. [37]. Similarly to other well-known GNN architectures [33, 52, 53], the aggregation scheme of the vanilla GAT is designed as an average of the feature vectors of the neighbors—where, by definition, $\sum_{v_{\mathbf{j}} \in \mathcal{N}_{\mathbf{i}}} a_{ij} = 1$ for all v_i —rather than as a general weighted sum like for Eq. (12). This is often reasonable in the context of structure learning, where the node features represent some coordinates in a metric space where connected nodes are likely to be close [33]. Yet, in the general case, this type of constraint was shown to lessen dramatically the expressive power of the GNN architecture [31]. We also reached the same conclusion while using average-like GNN architectures (see the Supplementary Material). By contrast, the aggregation scheme described by Eq. (12) allows our architecture to represent various dynamic processes correctly on networks.

3. Training settings

In all experiments, we use the cross entropy loss as the local loss function,

$$\mathbf{L}(\mathbf{y}_{\mathbf{i}}, \hat{\mathbf{y}}_{\mathbf{i}}) = - \sum_{\mathbf{m}} \mathbf{y}_{\mathbf{i}, \mathbf{m}} \log \hat{\mathbf{y}}_{\mathbf{i}, \mathbf{m}}, \quad (13)$$

where $y_{i,m}$ corresponds to the m -th element of the outcome vector of node v_i , which either is a transition probability for the stochastic contagion dynamics or a fraction of people for the metapopulation dynamics. For the simple, complex and interacting contagion dynamics, we used the observed outcomes \tilde{y}_i , corresponding to the stochastic state of node v_i at the next time step, as the target in the loss function. While we noticed a diminished performance when using the observed outcomes as opposed to the true transition probabilities (see Supplementary Material), this setting is more realistic and shows what happens when the targets are noisy. The effect of noise can be tempered by increasing the size of the dataset (see the Supplementary Material). For the metapopulation dynamics, since this model is deterministic, we used the true targets without adding noise.

4. Performance measures

FIG. 5. (color online) **Visualization of the GNN architecture.** The blocks of different colors represent mathematical operations. The red blocks correspond to trainable affine transformation parametrized by weights and biases. The purple blocks represent activation functions between each layer. The core of the model is the attention module [37], which is represented in blue. The orange block at the end is an exponential Softmax activation that transforms the output into properly normalized outcomes.

Dynamics	Simple	Complex	Interacting	Metapopulation
Input layers			Linear(1, 32)	Linear(4, 32)*
	Linear(1, 32)	Linear(1, 32)	ReLU	ReLU
	ReLU	ReLU	Linear(32, 32)	Linear(32, 32)
	Linear(32, 32)	Linear(32, 32)	ReLU	ReLU
	ReLU	ReLU	Linear(32, 32)	Linear(32, 32)
Number of attention layers	2	2	4	8**
Output layers			Linear(32, 32)	Linear(32, 32)
	Linear(32, 32)	Linear(32, 32)	ReLU	ReLU
	ReLU	ReLU	Linear(32, 32)	Linear(32, 32)
	Linear(32, 2)	Linear(32, 2)	ReLU	ReLU
	Softmax	Softmax	Linear(32, 4)	Linear(32, 32)
Number of parameters	6 698	6 698	11 188	99 883

TABLE I. **Layer by layer description of the GNN models for each dynamics.** For each sequence, the operations are applied from top to bottom. The operations represented by $\text{Linear}(m, n)$ correspond to linear (or affine) transformations of the form $f(\mathbf{x}) = \mathbf{W}\mathbf{x} + \mathbf{b}$, where $\mathbf{x} \in \mathbb{R}^m$ is the input, $\mathbf{W} \in \mathbb{R}^{n \times m}$ and $\mathbf{b} \in \mathbb{R}^n$ are trainable parameters. The operations ReLU and Softmax are activation functions given by $\text{ReLU}(x) = \max\{x, 0\}$ and $\text{Softmax}(\mathbf{x}) = \frac{\exp(\mathbf{x})}{\sum_i \exp(x_i)}$. (*) Here, the dimension of the input is 4, because we concatenated with three-dimensional vector state with the rescaled and centered population size N_i . (**) Because the networks are weighted for the metapopulation dynamics, we initially transform the edge weights into abstract feature representations using a sequence of layers, i.e. $(\text{Linear}(1, 32), \text{ReLU}, \text{Linear}(1, 32))$ applied from left to right, before using them in the attention modules. These layers are trained alongside all the other layers.

We use the Pearson correlation coefficient r as a global performance measure defined on a set of targets Y and predictions \hat{Y} as

$$r = \frac{\mathbb{E}[(\mathbf{Y} - \mathbb{E}[\mathbf{Y}])(\hat{\mathbf{Y}} - \mathbb{E}[\hat{\mathbf{Y}}])]}{\sqrt{\mathbb{E}[(\mathbf{Y} - \mathbb{E}[\mathbf{Y}])^2] \mathbb{E}[(\hat{\mathbf{Y}} - \mathbb{E}[\hat{\mathbf{Y}}])^2]}} \quad (14)$$

where $\mathbb{E}[W]$ denotes the expectation of W . Also, because the

maximum correlation occurs at $r = 1$, we also define $1 - r$ as the global error on the set of target-prediction pairs.

5. Synthetic data generation

We generate data from each dynamics using the following algorithm:

1. Sample a graph G from a given generative model (e.g. the Erdős-Rényi $G(N, M)$ or the Barabási-Albert network models).
2. Initialize the state of the system $X(0) = (x_i(0))_{i=1..N}$. For the simple, complex and interacting contagion dynamics, sample uniformly the number of nodes in each state. For the metapopulation dynamics, sample the population size for each node from a Poisson distribution of average 10^4 and then sample the number of infected people within each node from a binomial distribution of parameter 10^{-5} . For instance, a network of $|\mathcal{V}| = 10^3$ nodes will be initialized with a total of 100 infected people, on average, distributed among the nodes.
3. At time t , compute the observed outcome— Y_t for the metapopulation dynamics, and \tilde{Y}_t for the three stochastic dynamics. Then, record the states X_t and Y_t .
4. Repeat step 3 until $(t \bmod t_s) = 0$, where t_s is a resampling time. At this moment, apply step 2 to reinitialize the states X_t and repeat step 3.
5. Stop when $t = T$, where T is the targeted number of samples.

The resampling step parametrized by t_s indirectly controls the diversity of the training dataset. We allow t_s to be small for the contagion dynamics ($t_s = 2$) and larger for the metapopulation dynamics ($t_s = 100$) to emphasize on the performance of the GNN rather than the quality of the training dataset, while acknowledging that different values of t_s could lead to poor training (see Supplementary Material).

We trained the simple, complex and interacting contagion GNN models on networks of size $|\mathcal{V}| = 10^3$ nodes and on time series of length $T = 10^4$. To generate the networks, we either used Erdős-Rényi (ER) random networks $G(N, M)$ or Barabási-Albert (BA) random networks. In both cases, the parameter of the generative network models are chosen such that the average degree is fixed to $\langle k \rangle = 4$.

To train our models on the metapopulation dynamics, we generated 10 networks of $|\mathcal{V}| = 100$ nodes and generated for each of them time series of $t_s = 100$ time steps. This number of time steps roughly corresponds to the moment where the epidemic dies out. Similarly to the previous experiments, we used the ER and the BA models to generate the networks, where the parameters were chosen such that $\langle k \rangle = 4$. However, because this dynamics is not stochastic, we varied the average degree of the networks to increase the variability in the time series. This was done by randomly removing a fraction $p = 1 - \ln(1 - \mu + e\mu)$ of their edges, where μ was sampled for each network uniformly between 0 and 1. In this scenario, the networks were directed and weighted, with each edge weight e_{ij} being uniformly distributed between 0 and 1.

6. Hyperparameters

The optimization of the parameters was performed using the rectified Adam algorithm [54], which is hyperparameterized by $b_1 = 0.9$ and $b_2 = 0.999$, as suggested in Ref. [54].

To build a validation dataset, we selected a fraction of the node states randomly for each time step. More specifically, we chose node v_i at time t proportionally to its importance weight $w_i(t)$. For all experiments on synthetic dynamics, we randomly selected 10 nodes to be part of the validation set, on average. For all experiments, the learning rate ϵ was reduced by a factor 2 every 10 epochs with initial value $\epsilon_0 = 0.001$. A weight decay of 10^{-4} was used as well to help regularize the training. We trained all models for 30 epochs, and selected the GNN model with the lowest loss on validation datasets. We fixed the importance sampling bias exponents for the training to $\lambda = 0.5$ in the simple, complex and interacting contagion cases, and fixed it to $\lambda = 1$ in the metapopulation case.

B. Importance weights

In this section, we show how to implement the importance weights in the different cases. Other versions of the importance weights are also available in the Supplementary Material.

1. Discrete state stochastic dynamics

When \mathcal{S} is a finite countable set, the importance weights can be computed exactly using Eq. (7),

$$\mathbf{w}_i(\mathbf{t}) \propto \left[\rho(\mathbf{k}_i, \mathbf{x}_i(\mathbf{t}), \mathbf{x}_{\mathcal{N}_i}(\mathbf{t})) \right]^{-\lambda} \quad (15)$$

where $\rho(k, x, x_{\mathcal{N}})$ is the probability to observe a node of degree k in state x with a neighborhood in state $x_{\mathcal{N}}$ in the complete dataset D . When \mathcal{S} is a discrete and countable set, inputs can be simplified from $(k, x, x_{\mathcal{N}})$ to (k, x, ℓ) without loss of generality, where ℓ is a vector whose entries are the number of neighbors in each state. The distribution is then estimated from the complete dataset D by computing the fraction of inputs that are in every configuration

$$\rho(\mathbf{k}, \mathbf{x}, \ell) = \frac{1}{|\mathcal{V}|^{\mathbf{T}}} \sum_{\mathbf{i}=1}^{|\mathcal{V}|} \mathbf{I}(\mathbf{k}_i = \mathbf{k}) \times \sum_{\mathbf{t}=1}^{\mathbf{T}} \mathbf{I}(\mathbf{x}_i(\mathbf{t}) = \mathbf{x}) \mathbf{I}(\ell_i(\mathbf{t}) = \ell) \quad (16)$$

where $I(\cdot)$ is the indicator function.

2. Continuous state deterministic dynamics

The case of continuous states—e.g. for metapopulation dynamics—is more challenging than its discrete counterpart, especially if the node and edge attributes, Φ_i and Ω_{ij} , need

to be accounted for. One of the challenges is that we cannot count the inputs like in the previous case. As a result, the estimated distribution ρ cannot be estimated directly using Eq. (16), and we use instead

$$\mathbf{w}_i(\mathbf{t}) = [\mathbf{P}(\mathbf{k}_i) \Sigma(\Phi_i, \Omega_i | \mathbf{k}_i) \Pi(\bar{\mathbf{x}}(\mathbf{t}))]^{-\lambda} \quad (17)$$

where $P(k_i)$ is the fraction of nodes with degree k_i , $\Sigma(\Phi_i, \Omega_i | k_i)$ is the joint probability density function (pdf) conditioned on the degree k_i for the node attributes Φ_i and the sum of the edge attributes $\Omega_i \equiv \sum_{v_j \in \mathcal{N}_i} \Omega_{ij}$, and where $\Pi(\bar{\mathbf{x}}(t))$ is the pdf for the average of node states at time t $\bar{\mathbf{x}}(t) = \frac{1}{|\mathcal{V}|} \sum_{v_i \in \mathcal{V}} x_i(t)$. The pdf are obtained using Gaussian kernel density estimators (KDE) [55]. Provided that the density values of the KDE are unbounded above, we normalize the pdf such that the density of each sample used to construct the KDE sum to one. Further details on how we designed the importance weights are provided in the Supplementary Material.

C. Dynamics

In what follows, we describe in details the contagion dynamics used for our experiments. We specify the node outcome function f introduced in Eq. (3) and the parameters of the dynamics.

1. Simple contagion

We consider the simple contagion dynamics called the susceptible-infected-susceptible (SIS) dynamics for which $\mathcal{S} = \{S, I\} = \{0, 1\}$ —we use these two representations of \mathcal{S} interchangeably. Because this dynamics is stochastic, we let $\mathcal{R} = [0, 1]^2$. We define the infection function $\alpha(\ell)$ as the probability that a susceptible node becomes infected given its number of infected neighbors ℓ

$$\Pr(\mathbf{S} \rightarrow \mathbf{I} | \ell) = \alpha(\ell) = \mathbf{1} - (\mathbf{1} - \gamma)^\ell, \quad (18)$$

where $\gamma \in [0, 1]$ is the disease transmission probability. In other words, a node can be infected by any of its infected neighbors independently with probability γ . We also define the constant recovery probability as

$$\Pr(\mathbf{I} \rightarrow \mathbf{S}) = \beta. \quad (19)$$

The node outcome function for the SIS dynamics is therefore

$$\mathbf{f}(\mathbf{x}_i, \mathbf{x}_{\mathcal{N}_i}) = \begin{cases} (1 - \alpha(\ell_i), \alpha(\ell_i)) & \text{if } x_i = 0, \\ (\beta, 1 - \beta) & \text{if } x_i = 1, \end{cases} \quad (20)$$

where

$$\ell_i = \sum_{v_j \in \mathcal{N}_i} \delta(\mathbf{x}_j, \mathbf{1}) \quad (21)$$

is the number of infected neighbors of v_i and $\delta(x, y)$ is the Kronecker delta. Note that for each case in Eq. (20), the outcome is a two-dimensional probability vector, where the first entry is the probability that node v_i becomes/remains susceptible at the following time step, and the second entry is the probability that it becomes/remains infected. We used $(\gamma, \beta) = (0.04, 0.08)$ in all experiments involving this simple contagion dynamics.

2. Complex contagion

To lift the independent transmission assumption of the SIS dynamics, we consider a complex contagion dynamics for which the node outcome function has a similar form as Eq. (20), but where the infection function $\alpha(\ell)$ has the non-monotonic form

$$\alpha(\ell) = \frac{\mathbf{1}}{\mathbf{z}(\eta)} \frac{\ell^3}{e^{\ell/\eta} - \mathbf{1}} \quad (22)$$

where $z(\eta)$ normalizes the infection function such that $\alpha(\ell^*) = 1$ at its global maximum ℓ^* and $\eta > 0$ is a parameter controlling the position of ℓ^* . This function is inspired by the Planck distribution for the black-body radiation, although it was chosen for its general shape rather than for any physical meaning whatsoever. We used $(\eta, \beta) = (8, 0.06)$ in all experiments involving this complex contagion dynamics.

3. Interacting contagion

We define the interacting contagion as two SIS dynamics that are interacting and denote it as the SIS-SIS dynamics. In this case, we have $\mathcal{S} = \{S_1 S_2, I_1 S_2, S_1 I_2, I_1 I_2\} = \{0, 1, 2, 3\}$. Similarly to the SIS dynamics, we have $\mathcal{R} = [0, 1]^4$ and we define the infection probability functions

$$\alpha_{\mathbf{g}}(\ell_{\mathbf{g}}) = \mathbf{1} - (\mathbf{1} - \gamma_{\mathbf{g}})^{\ell_{\mathbf{g}}} \quad \text{if } \mathbf{x} = \mathbf{0} \quad (23a)$$

$$\alpha_{\mathbf{g}}^*(\ell_{\mathbf{g}}) = \mathbf{1} - (\mathbf{1} - \zeta \gamma_{\mathbf{g}})^{\ell_{\mathbf{g}}} \quad \text{if } \mathbf{x} = \mathbf{1}, \mathbf{2}, \quad (23b)$$

where $\zeta \geq 0$ is a coupling constant and $\ell_{\mathbf{g}}$ is the number of neighbors infected by disease \mathbf{g} , and also define the recovery probabilities $\beta_{\mathbf{g}}$ for each disease ($\mathbf{g} = 1, 2$). The case where $\zeta > 1$ corresponds to the situation in which the diseases are synergistic (i.e. being infected by one increases the probability of getting infected by the other), whereas competition is introduced if $\zeta < 1$ (being already infected by one decreases the probability of getting infected by the other). The case $\zeta = 1$ falls back on two independent SIS dynamics that evolve simultaneously on the network. The outcome function is composed of 16 entries that are expressed as follows

$$\mathbf{f}(\mathbf{x}_i, \mathbf{x}_{\mathcal{N}_i}) = \begin{cases} \left([1 - \alpha_1(\ell_{i,1})][1 - \alpha_2(\ell_{i,2})], \alpha_1(\ell_{i,1})[1 - \alpha_2(\ell_{i,2})], [1 - \alpha_1(\ell_{i,1})]\alpha_2(\ell_{i,2}), \alpha_1(\ell_{i,1})\alpha_2(\ell_{i,2}) \right) & \text{if } x_i = 0, \\ \left(\beta_1[1 - \alpha_2^*(\ell_{i,2})], [1 - \beta_1][1 - \alpha_2^*(\ell_{i,2})], \beta_1\alpha_2^*(\ell_{i,2}), [1 - \beta_1]\alpha_2^*(\ell_{i,2}) \right) & \text{if } x_i = 1, \\ \left([1 - \alpha_1^*(\ell_{i,1})]\beta_2, \alpha_1^*(\ell_{i,1})\beta_2, [1 - \alpha_1^*(\ell_{i,1})][1 - \beta_2], \alpha_1^*(\ell_{i,1})[1 - \beta_2] \right) & \text{if } x_i = 2, \\ \left(\beta_1\beta_2, [1 - \beta_1]\beta_2, \beta_1[1 - \beta_2], [1 - \beta_1][1 - \beta_2] \right) & \text{if } x_i = 3. \end{cases} \quad (24)$$

where we define $\ell_{i,g}$ as the number of neighbors of v_i that are infected by disease g . We used $(\gamma_1, \gamma_2, \beta_1, \beta_2, \zeta) = (0.01, 0.012, 0.19, 0.22, 50)$ in all experiments involving this interacting contagion dynamics.

4. Metapopulation

The metapopulation dynamics considered is a deterministic version of the susceptible-infection-recovered (SIR) metapopulation model [43–46]. We consider that the nodes are populated by a fixed number of people N_i , which can be in three states—susceptible (S), infected (I) or recovered (R). We therefore track the number of people in every state at each time. Furthermore, we let the network G be weighted, with the weights describing the mobility flow of people between regions. In this case, $\Omega_{ij} \in \mathbb{R}$ is the average number of people that are traveling from node v_j to node v_i . Finally, because we assume that the population size is on average steady, we let $\Phi_i = N_i$ be a node attribute and work with the fraction of people in every epidemiological state. More precisely, we define the state of node v_j by $x_j = (s_j, i_j, r_j)$, where s_j, i_j and r_j are the fractions of susceptible, infected and recovered people, respectively. From these definitions, we define the node outcome function of this dynamics as

$$\mathbf{f}(\mathbf{x}_j, \mathbf{x}_{\mathcal{N}_j}, \mathbf{G}) = \begin{pmatrix} s_j - s_j\tilde{\alpha}_j \\ i_j - \frac{i_j}{\tau_r} + s_j\tilde{\alpha}_j \\ r_j + \frac{i_j}{\tau_r} \end{pmatrix} \quad (25)$$

where

$$\tilde{\alpha}_j = \alpha(\mathbf{i}_j, \mathbf{N}_j) + \sum_{\mathbf{v}_i \in \mathcal{N}_j} \frac{k_j \Omega_{ji} \alpha(\mathbf{i}_i, \mathbf{N}_i)}{\sum_{\mathbf{v}_n \in \mathcal{N}_j} \Omega_{jn}}, \quad (26)$$

and k_j is the degree of node v_j . The function $\alpha(i, N)$ corresponds to the infection rate, per day, at which an individual is infected by someone visiting from a neighboring region with iN infected people in it, and is equal to

$$\alpha(\mathbf{i}, \mathbf{N}) = \mathbf{1} - \left(\mathbf{1} - \frac{\mathbf{R}_0}{\tau_r \mathbf{N}} \right)^{i\mathbf{N}} \approx \mathbf{1} - e^{-\frac{\mathbf{R}_0}{\tau_r} \mathbf{i}}. \quad (27)$$

where R_0 corresponds to the reproduction number and, τ_r is the average recovery time in days. In all experiments with this metapopulation dynamics, we used $(R_0, \tau_r) = (8.31, 7.5)$.

D. COVID-19 outbreak in Spain

1. Dataset

We use the time series of the COVID-19 outbreak in Spain. The dataset is originally composed of the daily incidence number of the 52 Spanish provinces monitored for 262 days between February 1st 2020 and October 20th 2020 [56], and coupled with the origin destination (OD) network of individual mobility [57]. The network is multiplex, directed and weighted, where the weight of each edge e_{ij}^ν represents mobility flow from province v_j and to province v_j using transportation ν . The metadata associated to each node is the population of province v_i [58], noted $\Phi_i = N_i$. The metadata associated to each edge, Ω_{ij}^ν , corresponds to the average number of people that moved from v_j to v_i using ν as the main means of transportation.

Because the raw dataset only includes the incidence number, that is the number of new infected cases per day, we artificially include a recovery mechanism where the average recovery time $\tau_r = 7.5$ days, which was used in previous studies involving COVID-19 in Spain [50].

2. Models

We consider a GNN model that is essentially identical to the one used for the metapopulation dynamics (see Tab. I), with the exception that we used different attention modules for each type of edges. Then, to combine the features of each multiplex layer, we simply average pooled the features together.

For the metapopulation model, we fixed $R_0 = 2.5$ and $\tau_r = 7.5$. These values were used in Ref. [50] to evaluate the efficacy of the containment measures in Spain.

3. Training

To train the model on the COVID-19 dataset, we use the same hyperparameters values that we used for the metapopulation experiment, which are listed in Sec. V A 3. The only differences were the value of the exponent bias, which we fixed to $\lambda = 0.75$. Similarly an earlier discussion in Sec. V A, we constructed the validation dataset by randomly selecting a partition of the nodes at each time step proportionally to their

importance weights $w_i(t)$. Analogously, the test dataset was constructed by uniformly selecting the partitions of nodes at

each time step. We selected on average 10% of the complete dataset to build the validation set and used 30% to build the testing set, thus leaving 60% of the samples for training set.

-
- [1] W. O. Kermack and A. G. McKendrick, "A Contribution to the Mathematical Theory of Epidemics," *Proc. R. Soc. A* **115**, 700–721 (1927).
- [2] H. W. Hethcote, "The Mathematics of Infectious Diseases," *SIAM Rev.* **42**, 599–653 (2000).
- [3] C. I. Siettos and L. Russo, "Mathematical modeling of infectious disease dynamics," *Virulence* **4**, 295–306 (2013).
- [4] I. Z. Kiss, J. C. Miller, and P. L. Simon, *Mathematics of Epidemics on Networks* (Springer, 2017) p. 598.
- [5] F. Brauer, C. Castillo-Chavez, and Z. Feng, *Mathematical Models in Epidemiology* (Springer, 2019).
- [6] N. C. Grassly and C. Fraser, "Mathematical models of infectious disease transmission," *Nat. Rev. Microbiol.* **6**, 477–487 (2008).
- [7] A. Pastore y Piontti, N. Perra, L. Rossi, N. Samay, and A. Vespignani, *Charting the Next Pandemic: Modeling Infectious Disease Spreading in the Data Science Age* (Springer, 2019).
- [8] C. Viboud and A. Vespignani, "The future of influenza forecasts," *Proc. Natl. Acad. Sci. U.S.A.* **116**, 2802–2804 (2019).
- [9] D. M. Morens, J. K. Taubenberger, and A. S. Fauci, "Predominant Role of Bacterial Pneumonia as a Cause of Death in Pandemic Influenza: Implications for Pandemic Influenza Preparedness," *J. Infect. Dis.* **198**, 962–970 (2008).
- [10] J. Sanz, C.-Y. Xia, S. Meloni, and Y. Moreno, "Dynamics of Interacting Diseases," *Phys. Rev. X* **4**, 041005 (2014).
- [11] S. Nickbakhsh, C. Mair, L. Matthews, R. Reeve, P. C. D. Johnson, F. Thornburn, B. von Wissmann, A. Reynolds, J. McMenamin, R. N. Gunson, and P. R. Murcia, "Virus–virus interactions impact the population dynamics of influenza and the common cold," *Proc. Natl. Acad. Sci. U.S.A.* **116**, 27142–27150 (2019).
- [12] L. Hébert-Dufresne, S. V. Scarpino, and J.-G. Young, "Macroscopic patterns of interacting contagions are indistinguishable from social reinforcement," *Nat. Phys.* **16**, 426–431 (2020).
- [13] D. Centola, "The Spread of Behavior in an Online Social Network Experiment," *Science* **329**, 1194–1197 (2010).
- [14] S. Lehmann and Y.-Y. Ahn, eds., *Complex Spreading Phenomena in Social Systems*, Computational Social Sciences (Springer, 2018).
- [15] I. Iacopini, G. Petri, A. Barrat, and V. Latora, "Simplicial models of social contagion," *Nat. Commun.* **10**, 2485 (2019).
- [16] M. Biggerstaff, M. Johansson, D. Alper, L. C. Brooks, P. Chakraborty, D. C. Farrow, S. Hyun, S. Kandula, C. McGowan, N. Ramakrishnan, R. Rosenfeld, J. Shaman, R. Tibshirani, R. J. Tibshirani, A. Vespignani, W. Yang, Q. Zhang, and C. Reed, "Results from the second year of a collaborative effort to forecast influenza seasons in the United States," *Epidemics* **24**, 26–33 (2018).
- [17] S. L. Brunton, J. L. Proctor, and J. N. Kutz, "Discovering governing equations from data by sparse identification of nonlinear dynamical systems," *Proc. Natl. Acad. Sci. USA* **113**, 3932–3937 (2016).
- [18] J. N. Kutz, "Deep learning in fluid dynamics," *J. Fluid Mech.* **814**, 1–4 (2017).
- [19] J. Pathak, Z. Lu, B. R. Hunt, M. Girvan, and E. Ott, "Using machine learning to replicate chaotic attractors and calculate Lyapunov exponents from data," *Chaos* **27**, 121102 (2017).
- [20] B. Lusch, J. N. Kutz, and S. L. Brunton, "Deep learning for universal linear embeddings of nonlinear dynamics," *Nat. Commun.* **9**, 1–10 (2018).
- [21] J. Pathak, B. Hunt, M. Girvan, Z. Lu, and E. Ott, "Model-Free Prediction of Large Spatiotemporally Chaotic Systems from Data: A Reservoir Computing Approach," *Phys. Rev. Lett.* **120**, 024102 (2018).
- [22] B. M. de Silva, D. M. Higdon, S. L. Brunton, and J. N. Kutz, "Discovery of Physics From Data: Universal Laws and Discrepancies," *Front. Artif. Intell.* **3**, 25 (2020).
- [23] X. Chen, T. Weng, H. Yang, C. Gu, J. Zhang, and M. Small, "Mapping topological characteristics of dynamical systems into neural networks: A reservoir computing approach," *Phys. Rev. E* **102**, 033314 (2020).
- [24] R. Dutta, A. Mira, and J.-P. Onnela, "Bayesian inference of spreading processes on networks," *Proc. R. Soc. A* **474**, 20180129 (2018).
- [25] C. Shah, N. Dehmamy, N. Perra, M. Chinazzi, A.-L. Barabási, A. Vespignani, and R. Yu, "Finding Patient Zero: Learning Contagion Source with Graph Neural Networks," (2020), [arXiv:2006.11913](https://arxiv.org/abs/2006.11913).
- [26] F. A. Rodrigues, T. Peron, C. Connaughton, J. Kurths, and Y. Moreno, "A machine learning approach to predicting dynamical observables from network structure," (2019), [arXiv:1910.00544](https://arxiv.org/abs/1910.00544).
- [27] A. Salova, J. Emenheiser, A. Rupe, J. P. Crutchfield, and R. M. D'Souza, "Koopman operator and its approximations for systems with symmetries," *Chaos* **29**, 093128 (2019).
- [28] E. Laurence, C. Murphy, G. St-Onge, X. Roy-Pomerleau, and V. Thibeault, "Detecting structural perturbations from time series with deep learning," (2020), [arXiv:2006.05232](https://arxiv.org/abs/2006.05232).
- [29] Z. Zhang, P. Cui, and W. Zhu, "Deep Learning on Graphs: A Survey," (2018), [arXiv:1812.04202](https://arxiv.org/abs/1812.04202).
- [30] J. Zhou, Gé Cui, Z. Zhang, C. Yang, Z. Liu, L. Wang, C. Li, and M. Sun, "Graph Neural Networks: A Review of Methods and Applications," (2018), [arXiv:1812.08434](https://arxiv.org/abs/1812.08434).
- [31] K. Xu, W. Hu, J. Leskovec, and S. Jegelka, "How Powerful are Graph Neural Networks?" (2018), [arXiv:1810.00826](https://arxiv.org/abs/1810.00826).
- [32] B. Perozzi, R. Al-Rfou, and S. Skiena, "DeepWalk: Online Learning of Social Representations," *Proc. ACM SIGKDD Int. Conf. Knowl. Discov. Data Min.*, 701–710 (2014), [arXiv:1403.6652](https://arxiv.org/abs/1403.6652).
- [33] W. L. Hamilton, R. Ying, and J. Leskovec, "Representation Learning on Graphs: Methods and Applications," (2017), [arXiv:1709.05584](https://arxiv.org/abs/1709.05584).
- [34] Z. Zhang, Y. Zhao, J. Liu, S. Wang, R. Tao, R. Xin, and J. Zhang, "A general deep learning framework for network reconstruction and dynamics learning," *Appl. Netw. Sci.* **4**, 110 (2019).
- [35] A. Fout, J. Byrd, B. Shariat, and A. Ben-Hur, "Protein Interface Prediction using Graph Convolutional Networks," in *Adv. Neural Inf. Process. Syst.* **30** (2017) pp. 6530–6539.
- [36] M. Zitnik, M. Agrawal, and J. Leskovec, "Modeling polypharmacy side effects with graph convolutional networks," in *Bioin-*

- formatics*, Vol. 34 (2018) pp. i457–i466.
- [37] P. Veličković, G. Cucurull, A. Casanova, A. Romero, P. Liò, and Y. Bengio, “Graph attention networks,” in *International Conference on Learning Representations* (2018).
- [38] A.-L. Barabási, *Network Science* (Cambridge University Press, 2016) p. 474.
- [39] R. Y. Rubinstein and D. P. Kroese, *Simulation and the Monte Carlo Method*, 3rd ed. (Wiley, 2016) p. 414.
- [40] L. Isella, J. Stehlé, A. Barrat, C. Cattuto, J.-F. Pinton, and W. Van den Broeck, “What’s in a crowd? Analysis of face-to-face behavioral networks,” *J. Theor. Biol.* **271**, 166–180 (2011).
- [41] R. Mastrandrea, J. Fournet, and A. Barrat, “Contact Patterns in a High School: A Comparison between Data Collected Using Wearable Sensors, Contact Diaries and Friendship Surveys,” *PLOS ONE* **10**, e0136497 (2015).
- [42] M. Génois and A. Barrat, “Can co-location be used as a proxy for face-to-face contacts?” *EPJ Data Sci.* **7**, 11 (2018).
- [43] V. Colizza, R. Pastor-Satorras, and A. Vespignani, “Reaction–diffusion processes and metapopulation models in heterogeneous networks,” *Nat. Phys.* **3**, 276–282 (2007).
- [44] D. Balcan, B. Gonçalves, H. Hu, J. J. Ramasco, V. Colizza, and A. Vespignani, “Modeling the spatial spread of infectious diseases: The global epidemic and mobility computational model,” *J. Comput. Sci.* **1**, 132–145 (2010).
- [45] M. Ajelli, Q. Zhang, K. Sun, S. Merler, L. Fumanelli, G. Chowell, L. Simonsen, C. Viboud, and A. Vespignani, “The RAPIDD Ebola forecasting challenge: Model description and synthetic data generation,” *Epidemics* **22**, 3–12 (2018).
- [46] D. Soriano-Paños, L. Lotero, A. Arenas, and J. Gómez-Gardeñes, “Spreading Processes in Multiplex Metapopulations Containing Different Mobility Networks,” *Phys. Rev. X* **8**, 031039 (2018).
- [47] P. Eichelsbacher and A. Ganesh, “Bayesian Inference for Markov Chains,” *J. Appl. Probab.* **39**, 91–99 (2002).
- [48] I. Voitalov, P. van der Hoorn, R. van der Hofstad, and D. Krioukov, “Scale-free networks well done,” *Phys. Rev. Research* **1**, 033034 (2019).
- [49] R. Pastor-Satorras, C. Castellano, P. Van Mieghem, and A. Vespignani, “Epidemic processes in complex networks,” *Rev. Mod. Phys.* **87**, 925–979 (2015).
- [50] A. Aleta and Y. Moreno, “Evaluation of the potential incidence of COVID-19 and effectiveness of containment measures in Spain: A data-driven approach,” *BMC Med.* **18**, 157 (2020).
- [51] I. Voitalov, P. van der Hoorn, R. van der Hofstad, and D. Krioukov, “Scale-free networks well done,” *Phys. Rev. Research* **1**, 033034 (2019).
- [52] T. N. Kipf and M. Welling, “Semi-Supervised Classification with Graph Convolutional Networks,” (2016), [arXiv:1609.02907](https://arxiv.org/abs/1609.02907).
- [53] C. Morris, M. Ritzert, M. Fey, W. L. Hamilton, J. E. Lenssen, G. Rattan, and M. Grohe, “Weisfeiler and Leman Go Neural: Higher-order Graph Neural Networks,” (2018), [arXiv:1810.02244](https://arxiv.org/abs/1810.02244).
- [54] L. Liu, H. Jiang, P. He, W. Chen, X. Liu, J. Gao, and J. Han, “On the Variance of the Adaptive Learning Rate and Beyond,” (2019), [arXiv:1908.03265](https://arxiv.org/abs/1908.03265).
- [55] W. J. Conover, *Practical nonparametric statistics* (John Wiley & Sons, 1998) p. 350.
- [56] “COVID-19 en España,” <https://cneccovid.isciii.es> (2020), [Accessed: 20-October-2020].
- [57] “Observatorio del Transporte y la Logística en España,” <https://observatoriotransporte.mitma.gob.es/estudio-experimental> (2018), [Accessed: 11-August-2020].
- [58] “Instituto Nacional de Estadística,” <https://www.ine.es> (2020), [Accessed: 11-August-2020].

Reviewers' Comments:

Reviewer #1:

Remarks to the Author:

The main contributions of this paper have not changed in the revision, for which my summary of the paper is the same as the last review. This paper proposes a complementary approach based on Graph Neural Networks to automatically learn the stochastic dynamics from time series data on complex networks. By predicting the local transition probabilities of each node on various states, the authors provide a data-driven approach to learn the effective mechanisms governing the propagation process of contagion. The accuracy of the proposed method is demonstrated through a lot of experiments. On generated networks from classical patterns such as Barabasi-Albert random network, the proposed method successfully fits the ground truth. On real-world temporal networks, the proposed method recovers the bifurcation diagram of different contagion dynamics based on epidemic models well. The experiment settings are rigorous and the results are convincing.

In the revision, the authors spent substantial efforts and the improvement of the paper is impressive. Compared to the paper before the revision, an experiment using real data of the COVID-19 outbreak in Spain has been added, which demonstrate the feasibility of the proposed model and the potential application value. The main contributions are also more clearly and accurately stated. Most of my concerns in the last review have received satisfactory responses in the revision and rebuttal.

However, I still have one major concern about the novelty of this paper. I understand that GNN are usually used to learn network structure rather than dynamical systems, which is the main novelty of this paper. Figs. 6, 7 and the corresponding result part in the Supplementary Material show that classical GNN models that focus on network structure fail to learn the SIS dynamics on Barabasi-Albert networks. Nevertheless, there are also GNN papers that deal with dynamic networks and achieve good results, such as the papers mentioned in the survey [1]. I wonder whether the models mentioned in [1] may also handle contagion dynamics under few assumptions. I suggest that the authors might also prove the superiority of the proposed framework over the methods in [1] through the results of experiments, like Figs. 6 and 7 did. If the methods in [1] are not applicable for these experiments, I recommend that the authors should explain the reasons or design new experiments that can compare these methods.

Another suggestion from me is about the baselines of the experiments. For example in the new experiment about COVID-19, the authors compare the GNN with MetaPop., a mechanistic model. It is not surprising that MetaPop. is far worse than GNN, due to the existence of the factors or assumptions that are unknown to the mechanistic model, such as confinement measures. GNN fits the empirical curve directly in a top-bottom way and is naturally more accurate. But most machine learning methods also learn the empirical data directly and they may be better baselines in this experiment to prove the effectiveness of GNN. I understand that the authors may intend to prove the superiority of GNN over mechanistic models, but the results compared with other machine learning methods are also important.

[1]Skarding, J., Gabrys, B., & Musial, K. (2020). Foundations and modelling of dynamic networks using Dynamic Graph Neural Networks: A survey. arXiv preprint arXiv:2005.07496.

Reviewer #2:

None

Reviewer #3:

Remarks to the Author:

I appreciate the works and the efforts of the authors in improving their manuscript. Still saying so, I am afraid that they have not been able to answer my three main concerns: interpretability, assessment of their models prediction performance and applicability to real data set.

The authors say about the unavailability of dataset, but I would like to point the availability of the real COVID dataset to them for almost last 10 months. That is the best any epidemiological dataset can be. Further, I point out that the difficulty may lie in the dearth of data in the field of epidemiology on which this model need to be trained, whereas this model can only be trained when you have a lot of data. Somehow, after this resubmission I get the feeling authors know this fact and not able to appreciate that.

Response to Referee 1

Comment 1.1

The main contributions of this paper have not changed in the revision, for which my summary of the paper is the same as the last review. This paper proposes a complementary approach based on Graph Neural Networks to automatically learn the stochastic dynamics from time series data on complex networks. By predicting the local transition probabilities of each node on various states, the authors provide a data-driven approach to learn the effective mechanisms governing the propagation process of contagion. The accuracy of the proposed method is demonstrated through a lot of experiments. On generated networks from classical patterns such as Barabasi-Albert random network, the proposed method successfully fits the ground truth. On real-world temporal networks, the proposed method recovers the bifurcation diagram of different contagion dynamics based on epidemic models well. The experiment settings are rigorous and the results are convincing.

Reply 1.1

Once again, we thank the Reviewer for this accurate presentation of our work.

Comment 1.2

In the revision, the authors spent substantial efforts and the improvement of the paper is impressive. Compared to the paper before the revision, an experiment using real data of the COVID-19 outbreak in Spain has been added, which demonstrate the feasibility of the proposed model and the potential application value. The main contributions are also more clearly and accurately stated. Most of my concerns in the last review have received satisfactory responses in the revision and rebuttal.

Reply 1.2

We appreciate that the Reviewer acknowledges the new contributions and the amount of effort we have put in this new version of the paper. We also thank them for recognizing that our rebuttal was satisfactory in regards to most of their concerns.

Comment 1.3

However, I still have one major concern about the novelty of this paper. I understand that GNN are usually used to learn network structure rather than dynamical systems, which is the main novelty of this paper. Figs. 6, 7 and the corresponding result part in the Supplementary Material show that classical GNN models that focus on network structure fail to learn the SIS dynamics on Barabasi-Albert networks.

Reply 1.3

We thank the Reviewer for sharing their remaining concern in such a constructive manner. We also appreciate that the Reviewer now acknowledges one of our main contributions, that of using GNN to learn networked dynamical systems. Of course, we believe this conceptual leap is worth publishing, but we believe that our contribution goes well beyond this conceptual leap.

First, we have developed a general mathematical framework to work with these kinds of tools. This framework is very general and versatile, as we have been able to demonstrate through the variety of our experiments—from simple and complex stochastic contagion dynamics, to metapopulation dynamics on weighted networks and real-world time series forecasting on multiplex, directed and weighted networks. The versatility of our framework is in sharp contrast with other existing approaches [1, 2] which were designed for specific applications—and are therefore not seamlessly transferable to other dynamical processes—, in addition to suffering from most of the limitations of the GNNs used in

Supp. Fig. 6 and 7. Our framework is also applicable in contexts outside of epidemiology (e.g. linking fMRI time series with structural connectivity of the brain in neurosciences), which adds to the relevance of our contribution to the broad readership of Nature Communications.

Second, we have shown how to rigorously train and test our models. While the former is mostly addressed by our importance sampling procedure—of which the novelty has not been previously commented by the Reviewer—, the latter is addressed by our analysis of the extracted bifurcation diagrams. Together, these two techniques allow us to build from data accurate and consistent models for dynamics on networks. The depth of our analysis was possible in part because we worked with synthetic time series of which the underlying dynamics is known. This is a luxury that only a few machine learning works have, especially in the context of GNN learning where toy examples are hard to come by. This set-up allowed us to show that degree heterogeneity is advantageous in GNN learning, which to the best of our knowledge is also a novel contribution.

Third, we agree with the Reviewer that our GNN architecture is not particularly groundbreaking—after all, it is mostly inspired by the graph attention network (GAT) [3]. Nevertheless, we believe it is a contribution on its own. Namely, we showed in Figs. 6 and 7 of the Supplementary Materials that not all architectures can accurately carry out the task of dynamics learning, and we went further into generalizing our GNN architecture to weighted and multiplex networks. While we are aware that other weighted-multiplex GNN frameworks also exist [4, 5], they are typically oriented towards structure learning and often use classical GNN models at their core. For instance, Ref. [4] uses a GraphSAGE architecture. They are therefore expected to suffer from the same limitations as the models presented in Supp. Figs. 6 and 7.

Comment 1.4

Nevertheless, there are also CNN papers that deal with dynamic networks and achieve good results, such as the papers mentioned in the survey [6]. I wonder whether the models mentioned in [6] may also handle contagion dynamics under few assumptions. I suggest that the authors might also prove the superiority of the proposed framework over the methods in [6] through the results of experiments, like Figs. 6 and 7 did. If the methods in [6] are not applicable for these experiments, I recommend that the authors should explain the reasons or design new experiments that can compare these methods.

Reply 1.4

We thank the Reviewer for mentioning Ref. [6]. We realize that it was mentioned in their first report, and we apologize for having overlooked it. In short, the models presented in Ref. [6] are inappropriate for our task, i.e. that of learning dynamical systems on *static graphs*, because they are applied to time-varying (or temporal) graphs where nodes and edges appear and disappear over time. Their review covers a large variety of GNN learning problems indeed, from supervised to unsupervised structure learning tasks with discrete and continuous time-varying graph sequences, but these tools are hardly transferable to the challenge of learning the dynamical process occurring on a static network.

In this framework, the datasets are assumed to be composed of sequences of temporally ordered graphs ($\dots, G_{t-1}, G_t, G_{t+1}, \dots$) and the task is to learn embeddings of these graph sequences. Among the architectures presented in Ref. [6] that would potentially be applicable to our learning task, the *stacked dynamic CNN* (SDGNN)—presented in Fig. 1—uses an array of p graph convolutions to compute the node embeddings of a sequence of graphs $\rho = (G_1, \dots, G_p)$ —each graph convolution being applied to its respective graph. Then, it combines the temporal information of the node embeddings using parallel LSTM cells [7]. The output of the SDGNN hence represents temporal node embeddings of the graph sequence ρ . The other architectures presented in Ref. [6], such as the *dynamic graph autoencoders* (DGAE) and the *integrated dynamic CNN* (IDGNN), are either designed for other purposes (e.g. DGAE) or they operate in a very similar fashion as the SDGNN shown in Fig. 1 (e.g. IDGNN). While it could be possible to train a

FIG. 1. GNN architecture described in Reply 1.4. The node features x_t^i and the adjacency matrices A^t of 4 graph snapshots are fed to 4 different graph convolutional networks in parallel, then fed a sequence of LSTMs—one for each node. This figure was taken from Ref. [6], Fig. 7.

SDGNN (or an IDGNN) for learning dynamical systems on static graphs—which contain no temporal information, by definition—, it would be a suboptimal use of their architectural features.

That being said, we understand the need to compare the performance of our architecture with competing GNN architectures. Lately, the use of GNNs for forecasting certain kinds of time series has been considered by a few recent works, more specifically in the context of forecasting COVID-19 [1]. These models are used to forecast time series of the incidence by aggregating spatiotemporal information between the nodes using GNNs.

In Ref. [1], their GNN aggregator stacks two graph convolutional networks (GCN) in series. They use this architecture, jointly with an unweighted and thresholded mobility flow network, to forecast the COVID-19 incidence in US counties. Their approach seems to perform very well on this dataset, hence we believe it would yield an adequate comparison with our approach.

Supp. Figs. 6 and 7 show that this architecture, denoted by Kapoor-GNN, performs quite similarly to the other non-extensive aggregators—e.g. the GraphSAGE, GAT and GCN architectures. This suggests that increasing the depth of the GNN models does not seem to help these architectures to better learn the dynamics, at least in the context of synthetic contagion dynamics. Also, we used the architecture as a baseline for our COVID-19 experiment—the KP model—, among other baselines (see Comment 1.5). Like for synthetic dynamics, their model has a hard time learning an accurate effective model for the COVID-19 evolution in Spain. The fact that this architecture performed badly in both our synthetic and real-world experiments shows that the problem of dynamics learning is not handled straightforwardly by standard techniques. It shows that any GNN architecture must be carefully designed in order to be successful.

Action taken 1.4

As we mentioned in the reply, we added the Kapoor aggregation mechanism to our experiment of Supp. Figs. 6 and 7. This experiment suggests that, like many GNN architectures, the Kapoor-GNN is not adequate for learning stochastic contagion dynamics. We also included the original model of Ref. [1], denoted the KP model, as a baseline in our COVID-19 experiment (see Comment 1.5).

Comment 1.5

Another suggestion from me is about the baselines of the experiments. For example in the new experiment about COVID-19, the authors compare the GNN with MetaPop., a mechanistic model. It is not surprising that MetaPop. is far worse than GNN, due to the existence of the factors or assumptions that are unknown to the mechanistic model, such as confinement measures. GNN fits the empirical curve directly in a top-bottom way and is naturally more accurate. But most machine learning methods also learn the empirical data directly and they may be better baselines in this experiment to prove the effectiveness of GNN. I understand that the authors may intend to prove the superiority of GNN over mechanistic models, but the results compared with other machine learning methods are also important.

Reply 1.5

We agree with the Reviewer that including additional baselines to our COVID-19 experiment would benefit greatly our analysis, and we thank them for insisting on this. In our new COVID-19 experiment, we included four alternative data-driven baselines: Three neural network architectures—the IND, FC and KP-GNN models—, one of which (the KP-GNN model) has been used to predict COVID-19 in the US [1], and a vector autoregressive (VAR) model [8]. Whereas the KP-GNN model is a natural competing architecture to ours (see Reply 1.4), the other two architectures embody the assumptions that the nodes are either mutually independent from one another (the IND model), or arbitrary codependent (the FC model) regardless of the structure of the mobility network.

Our new analysis suggests two things. First, classical multivariate techniques, such as the FC and the VAR models, are more susceptible to overfitting than models like the GNN and IND models when the datasets are small. This is unsurprising because the FC and VAR models have no proxy on how the nodes interact, contrary to the GNN and IND models, thus they are likely to learn faulty interaction patterns. Second, the GNN seems to be robust to spurious networks: The network does not have to be 100% accurate or influential with respect to the time series for the GNN to learn a good effective model. This is reflected by the fact that the mobility network, which was recorded in 2018, was likely different from the true, evolving mobility network of 2020 because of lockdowns and containment measures. Interestingly, these variations in the mobility flow between provinces due to the pandemic did not prevent the GNN from performing well.

Action taken 1.5

We included the performance of each baseline in Fig. 5 (a-b) of the new version of the manuscript and extended the related discussion in the main text to provide more details about them. We also added the remaining technical details in the Material and Methods Sec. D2 of the manuscript.

Response to Referee 3

Comment 3.1

I appreciate the works and the efforts of the authors in improving their manuscript.

Reply 3.1

We thank the Reviewer for acknowledging the amount of effort we have put in this new version of the paper.

Comment 3.2

Still saying so, I am afraid that they have not been able to answer my three main concerns: interpretability, ...

Reply 3.2

The Reviewer criticizes the lack of interpretability of the learned parameters, which is the case for any deep learning approach, whether they are applied to the learning of an unknown dynamical process (as in our manuscript), to natural language processing or to image classification. Although some explainability techniques exist for GNN used in specific contexts [9], the lack of interpretability remains an important drawback of these methods as a whole. However the reason these methods are so powerful is that they offer something else that is complementary to "traditional" interpretable approaches. Indeed, deep learning models can provide invaluable insights about natural phenomena, as was shown in Refs. [10, 11] for instance. If the models are carefully designed, regardless of the fact that their parameters and hidden feature layers are by themselves hardly interpretable, they can be very powerful tools for studying complex dynamical systems, thus complementing "traditional" mathematical modeling.

We provide an illustration of this complementarity in the new version of the manuscript by showing how our method can be used to unveil the nature (e.g. continuous/discontinuous) and the position (e.g. epidemic threshold) of the phase transition from the data alone, without any assumption whatsoever about the dynamical process (other than very general assumptions like that it is Markovian). This is in sharp contrast with any mechanistic approach where the nature of the phase transition, for instance, will be a direct consequence of the assumptions made about the dynamical process.

Besides the additions made to the manuscript during the previous round of reviews, which the Reviewer did not mention in their report (e.g. suggestions of ways to address the inherent lack of interpretability of deep learning techniques, or examples of articles in which the lack of interpretability has been addressed), we added a new experiment in the Supplementary Material where the interpretability of our models is discussed. More specifically, we discuss the interpretability of the attention coefficients, standing at the very core of our models.

Let us recall that the attention mechanism computes attention coefficients a_{ij} which weigh the influence of a source node j over one of its targeted neighbors i as a function of their states x_j and x_i . When available, the attention coefficients also depend on the node attributes, D_j and D_i , and the edge attributes, ω_{ij} . As a result, the attention coefficients could therefore have a direct interpretation: The higher the value of a_{ij} , the stronger is the interaction between nodes v_i and v_j . The new section of the Supplementary Material shows that this intuition is only partially true: When nodes are in neighbor-invariant states-such that their transition probabilities are independent of the states of their neighbors-the attention coefficients are correctly close to zero as shown on Fig. 2. However, we also find that some coefficients which we would expect to be zero (e.g. S \rightarrow S pairs) are not so.

We believe these non-zero attention coefficients are due to a sparsity problem in the representation learned by the attention modules, which is analogous to what happens in underdetermined linear regression problems. Indeed, the

FIG. 2. **Attention coefficients as a function of the source-target states for (a-d-g) simple contagion, (b-e-h) complex contagion and (c-f-i) interacting contagion dynamics.** We show the attention coefficients of different models: (a–c) are models with one attention layer, (d–f) have two attention layers and (g–i) have four. The values for each different attention layer are shown by the increasingly lighter colored bars. For the source-target states, we indicate the role of nodes using the directionality of the arrows: for $X \rightarrow Y$, X is state of the source node and Y is the state of the target node. We also highlight the source-target states where we expect the transition probability of the target node to be dependent on the state of the neighbor (i.e. the source node). The hyperparameters of the presented models are given in Tab. 1 of the main paper and in Material and Methods, Sec. A6.

attention coefficients are not the only parameters influencing the predictions of our model, and it is largely probable that their non-zero contribution ends up canceling each other in the outer layers of the GNN. In other words, our GNN architecture has several “degrees of freedom” in the way it can learn a representation of the dynamics, and since the loss function used to guide the learning of the parameters does not penalize spurious non-zero attention coefficients (but rather penalizes bad predictions outputted by the GNN), the learned attention coefficients may not respect the intuition discussed above.

This sparsity problem could be solved by regularization techniques such as LASSO [12] which would include an additional regularization term in the loss that would penalize unnecessary non-zero attention coefficients. Doing so would likely contribute to the interpretability of our models. However, doing so would also require the redesigning of some aspects of our method, and we leave this to future works.

Action taken 3.2

We clarified the interpretability of our model where the attention mechanism is presented (Material and Methods, Sec. A2), and in the new Sec. IV of the Supplementary Material where the addition of the L-1 norm to the loss function is also discussed.

Comment 3.3

... assessment of their models prediction performance ...

Reply 3.3

We are somewhat puzzled by the Reviewer's comment given that the purpose of Figs. 1, 2, 3 and 4(c)-(k) (in the previous version of the manuscript) was precisely to address the performance of our approach. These figures also show how our GNN model fundamentally differs from classical mechanistic models, in that it can learn to behave like many dynamics on networks without specifying their internal mechanism. We would be grateful to the Reviewer if they could provide us with more explanations and perhaps suggests ways to improve our validation process.

With respect to comparing our model's performance with the traditionally used epidemic models, Fig. 4(c) (in the previous version of the manuscript) compared the predictions of our model with that of a metapopulation model, a class of models that is currently used to forecast the current COVID-19 pandemic [13–31], and whose predictions have been featured in journals like Nature, Science and The Lancet, as well as being part of the CDC's Ensemble Forecasts of COVID-19 [32]. Our results illustrated the importance of including time-dependent or prevalence-dependent parameters to account for the dynamical interplay between social contacts (e.g. the lessening or the heightening of restrictions) and the spread of the virus [33, 34]. Unsurprisingly, using estimates from January and February 2020 for the epidemiological parameters—before major restrictions had been imposed except in a handful of countries—in a metapopulation model (as in Ref. [13]) yields poor forecasts. Other mechanisms, such as the ones used in Ref. [33], should therefore be added to our metapopulation model to extend its validity to the full length of the dataset we considered. Interestingly, this interplay between R_t and the prevalence of the disease is in part encoded in the daily incidence data and our GNN model appears to be able to capture it to some extent.

Additionally, we now compare the performance of our models for predicting COVID-19 in Spain with a vector autoregressive model (VAR), which belongs to a class of data-driven linear models for multivariate time series modeling. Whereas VAR models are not exactly mechanistic in nature, in that their parameters are not directly associated with physical or tangible quantities, they are often said to be interpretable because of their linearity. Like we discussed in the main text, our analysis shows that the VAR model tends to overfit the data, as shown by a significant decrease in performance over the out-of-sample prediction. They are also non-inductive by nature, thus making them impossible to use for extracting phase transition diagrams from a single time series data.

Action taken 3.3

As it is mentioned above and in Reply 1.5, we added the VAR model as a baseline in the COVID-19 experiment. We discussed in the main text the performance of this model with respect to 5 other modeling approaches. We now also discuss the performance of the metapopulation models in greater details.

Comment 3.4

... and applicability to real data set. The authors say about the unavailability of dataset, but I would like to point the availability of the real COVID dataset to them for almost last 10 months. That is the best any epidemiological dataset can be.

Reply 3.4

We fully agree with the Reviewer that the COVID-19 pandemic generated a body of datasets of unprecedented details

and volume, and that this kind of information is very well suited for the training of deep learning models. This is why we are quite surprised that the Reviewer does not acknowledge the new Fig. 4 of the previous version of the manuscript which uses precisely the COVID-19 dataset from Spain. Does the Reviewer believe we could have used it in a different way?

Comment 3.5

Further, I point out that the difficulty may lie in the dearth of data in the field of epidemiology on which this model need to be trained, whereas this model can only be trained when you have a lot of data. Somehow, after this resubmission I get the feeling authors know this fact and not able to appreciate that.

Reply 3.5

We fully agree with the Reviewer that epidemiological data require a lot of resources to be gathered, and that their size and details seldom reach the levels required by deep learning techniques. That being said, it is not because such datasets are currently not typical to come by that they will not become more common in a near future. For instance, new initiatives like Global.health [35] are good examples on how the international epidemiological community is coming together to share data more openly and to make available comprehensive datasets to all researchers. Thanks to such initiatives, it is likely that future pandemics will see larger amount of data available to the scientific community. It is therefore critical for the community to develop tools to leverage these datasets so that we are ready when the next pandemic hits.

Action taken 3.5

We added a paragraph in the discussion about the role of data-driven approaches in the future pandemics.

-
- [1] A. Kapoor, X. Ben, L. Liu, B. Perozzi, M. Barnes, M. Blais, and S. O'Banion, "Examining covid-19 forecasting using spatio-temporal graph neural networks," (2020), [arXiv:2007.03113](https://arxiv.org/abs/2007.03113).
 - [2] J. Gao, R. Sharma, C. Qian, L. M Glass, J. Spaeder, J. Romberg, J. Sun, and C. Xiao, "STAN: spatio-temporal attention network for pandemic prediction using real-world evidence," *J. Am. Med. Inform. Assoc* **28**, 733–743 (2021).
 - [3] P. Veličković, G. Cucurull, A. Casanova, A. Romero, P. Liò, and Y. Bengio, "Graph attention networks," in *International Conference on Learning Representations* (2018).
 - [4] Y. Cen, X. Zou, J. Zhang, H. Yang, J. Zhou, and J. Tang, "Representation learning for attributed multiplex heterogeneous network," in *Proceedings of the 25th ACM SIGKDD International Conference on Knowledge Discovery: Data Mining* (2019) p. 1358–1368.
 - [5] R. Zhum, K. Zhao, H. Yang, W. Lin, C. Zhou, B. Ai, Y. Li, and J. Zhou, "Aligraph: A comprehensive graph neural network platform," (2019), [arXiv:1902.08730](https://arxiv.org/abs/1902.08730).
 - [6] J. Skarding, B. Gabrys, and Musial K., "Foundations and modelling of dynamic networks using dynamic graph neural networks: A survey," (2020), [arXiv:2005.07496](https://arxiv.org/abs/2005.07496).
 - [7] I. Goodfellow, Y. Bengio, and A. Courville, *Deep Learning* (MIT Press, 2016).
 - [8] W. W. S. Wei, *Multivariate time series analysis and applications* (John Wiley & Sons, 2018).
 - [9] H. Yuan, H. Yu, S. Gui, and S. Ji, "Explainability in Graph Neural Networks: A Taxonomic Survey," (2021), [arXiv:2012.15445](https://arxiv.org/abs/2012.15445).
 - [10] B. Lusch, J. N. Kutz, and S. L. Brunton, "Deep learning for universal linear embeddings of nonlinear dynamics," *Nat. Commun.* **9**, 1–10 (2018).
 - [11] B. M. de Silva, D. M. Higdon, S. L. Brunton, and J. N. Kutz, "Discovery of Physics from Data: Universal Laws and Discrepancies," *Front. Artif. Intell.* **3** (2019), [arXiv:1906.07906](https://arxiv.org/abs/1906.07906).
 - [12] R. Tibshirani, "Regression shrinkage and selection via the lasso," *J. R. Stat. Soc. B* **58**, 267–288 (1996).
 - [13] A. Aleta and Y. Moreno, "Evaluation of the potential incidence of COVID-19 and effectiveness of containment measures in Spain: A data-driven approach," *BMC Med.* **18**, 157 (2020).
 - [14] A. Arenas, W. Cota, J. Gómez-Gardeñes, S. Gómez, C. Granell, J. T. Matamalas, D. Soriano-Paños, and B. Steinegger, "Modeling the Spatiotemporal Epidemic Spreading of COVID-19 and the Impact of Mobility and Social Distancing Interventions," *Phys. Rev. X* **10**, 041055 (2020).
 - [15] S. Chang, E. Pierson, P. W. Koh, J. Gerardin, B. Redbird, D. Grusky, and J. Leskovec, "Mobility network models of covid-19 explain inequities and inform reopening," *Nature* **589**, 82–87 (2021).
 - [16] M. Galanti, S. Pei, T. K. Yamana, F. J. Angulo, A. Charos, F. Khan, K. M. Shea, D. L. Swerdlow, and J. Shaman, "Non-pharmaceutical interventions and inoculation rate shape SARS-CoV-2 vaccination campaign success," *medRxiv*, 2021.02.22.21252240 (2021).
 - [17] S. Pei and J. Shaman, "Initial Simulation of SARS-CoV2 Spread and Intervention Effects in the Continental US," *medRxiv*, 2020.03.21.20040303 (2020).
 - [18] W. Yang, J. Shaff, and J. Shaman, "Effectiveness of non-pharmaceutical interventions to contain COVID-19: A case study of the 2020 spring pandemic wave in New York City," *J. R. Soc. Interface* **18**, 20200822 (2021).
 - [19] W. Yang, S. Kandula, M. Huynh, S. K. Greene, G. Van Wye, W. Li, H. T. Chan, E. McGibbon, A. Yeung, D. Olson, A. Fine, and J. Shaman, "Estimating the infection-fatality risk of SARS-CoV-2 in New York City during the spring 2020 pandemic wave: A model-based analysis," *The Lancet Infectious Diseases* **21**, 203–212 (2021).
 - [20] W. Yang, S. Kandula, and J. Shaman, "Simulating the impact of different vaccination policies on the COVID-19 pandemic in New York City," *medRxiv*, 2021.01.21.21250228 (2021).
 - [21] A. Zebrowski, A. Rundle, S. Pei, T. Yaman, W. Yang, B. G. Carr, S. Sims, R. Doorley, N. Schluger, J. W. Quinn, J. Shaman, and C. C. Branas, "A Spatiotemporal Tool to Project Hospital Critical Care Capacity and Mortality From COVID-19 in US Counties," *Am. J. Public Health* **111**, 1113–1122 (2021).
 - [22] D. Calvetti, A. P. Hoover, J. Rose, and E. Somersalo, "Metapopulation Network Models for Understanding, Predicting, and Managing the Coronavirus Disease COVID-19," *Front. Phys.* **8**, 261 (2020).
 - [23] L. Danon, E. Brooks-Pollock, M. Bailey, and M. Keeling, "A spatial model of CoVID-19 transmission in England and Wales: Early spread and peak timing," *medRxiv*, 2020.02.12.20022566 (2020).
 - [24] J. C. Lemaitre, K. H. Grantz, J. Kaminsky, H. R. Meredith, S. A. Truelove, S. A. Lauer, L. T. Keegan, S. Shah, J. Wills, K. Kaminsky, J. Perez-Saez, J. Lessler, and E. C. Lee, "A scenario modeling pipeline for COVID-19 emergency planning," *Sci. Rep.* **11**, 7534 (2021).

- [25] R. Li, S. Pei, B. Chen, Y. Song, T. Zhang, W. Yang, and J. Shaman, “Substantial undocumented infection facilitates the rapid dissemination of novel coronavirus (SARS-CoV-2),” Science 368, 489–493 (2020).
- [26] S. Pei, S. Kandula, and J. Shaman, “Differential effects of intervention timing on COVID-19 spread in the United States,” Sci. Adv. 6, eabd6370 (2020).
- [27] P. S. Peixoto, D. Marcondes, C. Peixoto, and S. M. Oliva, “Modeling future spread of infections via mobile geolocation data and population dynamics. An application to COVID-19 in Brazil,” PLOS ONE 15, e0235732 (2020).
- [28] T. Yamana, S. Pei, S. Kandula, and J. Shaman, “Projection of COVID-19 Cases and Deaths in the US as Individual States Re-open May 4, 2020,” medRxiv, 2020.05.04.20090670 (2020).
- [29] B. Rader, S. V. Scarpino, A. Nande, A. L. Hill, B. Adlam, R. C. Reiner, D. M. Pigott, B. Gutierrez, A. E. Zarebski, M. Shrestha, J. S. Brownstein, M. C. Castro, C. Dye, H. Tian, O. G. Pybus, and M. U. G. Kraemer, “Crowding and the shape of COVID-19 epidemics,” Nat. Med. 26, 1829–1834 (2020).
- [30] E. Estrada, “COVID-19 and SARS-CoV-2. Modeling the present, looking at the future,” Phys. Rep. (2020), 10.1016/j.physrep.2020.07.005.
- [31] J. T. Wu, K. Leung, and G. M. Leung, “Nowcasting and forecasting the potential domestic and international spread of the 2019-nCoV outbreak originating in Wuhan, China: A modelling study,” Lancet (2020), 10.1016/S0140-6736(20)30260-9.
- [0] <https://www.cdc.gov/coronavirus/2019-ncov/science/forecasting/forecasting-us.html>.
- [32] Q. Wang, S. Xie, Y. Wang, and D. Zeng, “Survival-Convolution Models for Predicting COVID-19 Cases and Assessing Effects of Mitigation Strategies,” Front. Public Health 8, 325 (2020).
- [33] L. Hébert-Dufresne, S. V. Scarpino, and J.-G. Young, “Macroscopic patterns of interacting contagions are indistinguishable from social reinforcement,” Nat. Phys. 16, 426–431 (2020).
- [1] <https://global.health>.

Deep learning of contagion dynamics on complex networks

Charles Murphy, Edward Laurence, and Antoine Allard
*Département de Physique, de Génie Physique, et d'Optique,
Université Laval, Québec (Québec), Canada G1V 0A6 and
Centre interdisciplinaire en modélisation mathématique,
Université Laval, Québec (Québec), Canada G1V 0A6*
(Dated: May 25, 2021)

Forecasting the evolution of contagion dynamics is still an open problem to which mechanistic models only offer a partial answer. To remain mathematically or computationally tractable, these models must rely on simplifying assumptions, thereby limiting the quantitative accuracy of their predictions and the complexity of the dynamics they can model. Here, we propose a complementary approach based on deep learning where the effective local mechanisms governing a dynamic are learned from time series data. Our graph neural network architecture makes very few assumptions about the dynamics, and we demonstrate its accuracy using different contagion dynamics of increasing complexity. By allowing simulations on arbitrary network structures, our approach makes it possible to explore the properties of the learned dynamics beyond the training data. Finally, we illustrate the applicability of our approach using real data of the COVID-19 outbreak in Spain. Our results demonstrate how deep learning offers a new and complementary perspective to build effective models of contagion dynamics on networks.

I. INTRODUCTION

Our capacity to prevent or contain outbreaks of infectious diseases is directly linked to our ability to accurately model contagion dynamics. Since the seminal work of Kermack and McKendrick almost a century ago [1], a variety of models incorporating ever more sophisticated contagion mechanisms has been proposed [2–5]. These mechanistic models have provided invaluable insights about how infectious diseases spread, and have thereby contributed to the design of better public health policies. However, several challenges remain unresolved, which call for contributions from new modeling approaches [6–8].

For instance, many complex contagion processes involve the nontrivial interaction of several pathogens [9–12], and some social contagion phenomena, like the spread of misinformation, require to go beyond pairwise interactions between individuals [13–15]. Also, while qualitatively informative, the forecasts of most mechanistic models lack quantitative accuracy [16]. Indeed, most models are constructed from a handful of mechanisms which can hardly reproduce the intricacies of real complex contagion dynamics. One approach to these challenges is to complexify the models by adding more detailed and sophisticated mechanisms. However, mechanistic models become rapidly intractable as new mechanisms are added. Moreover, models with higher complexity require the specification of a large number of parameters whose values can be difficult to infer from limited data.

There has been a recent gain of interest towards using machine learning to address the issue of the **often-limiting** complexity of mechanistic models [12, 17–23]. This new kind of approach aims at training predictive models directly from observational time series data. These data-driven models are then used for various tasks such as making accurate predictions [19, 21], gaining useful intuitions about complex phenomena [12] and discovering new patterns from which better mechanisms can be designed [17, 18]. While these approaches were originally designed for regularly structured

data, this new paradigm is now being applied to epidemics spreading on networked systems [24, 25], and more generally to dynamical systems [26–28]. Meanwhile, the machine learning community has dedicated a considerable amount of attention on deep learning on graphs, structure learning and graph neural networks (GNN) [29–31]. Recent works showed great promise for GNN in the context of community detection [32], link prediction [33], network inference [34], as well as in the context of discovering new materials and drugs [35, 36]. These advances **and recent results** [37–39] suggest that GNN could be prime candidates for building effective data-driven dynamical models on networks.

In this paper, we show how GNN, usually used for structure learning, can also be used to model contagion dynamics on complex networks. Our contribution is threefold. First, we design a training procedure and an appropriate GNN architecture capable of representing a wide range of dynamics, with very few assumptions. Second, we demonstrate the validity of our approach using various contagion dynamics on networks of different natures **and of increasing complexity**, as well as on real epidemiological data. Finally, we show how our approach can provide predictions for previously unseen network structures, therefore allowing the exploration of the properties of the learned dynamics beyond the training data. Our work generalizes the idea of constructing dynamical models from regularly structured data to arbitrary network structures, and suggests that our approach could be accurately extended to many other classes of dynamical processes.

II. RESULTS

In our approach, we assume that an *unknown dynamical process*, denoted M , takes place on a *known network structure*—or ensemble of networks—, denoted $G =$

$E = \{e_{ij}/v_i \text{ is connected to } v_j \wedge (v_i, v_j) \in V^2 \text{ is the edge } (V, E; \mathbb{X})\}$, where $V = \{v_1, \dots, v_N\}$ is the node set and \mathbb{X} is the set of parameters. We also assume that the network(s) can have some

metadata, taking the form of node and edge attributes denoted $\mathbf{x}_i = (\mathbf{x}_1(v_i), \dots, \mathbf{x}_Q(v_i))$ for node v_i and $\mathbf{x}_{ij} = (l_1(e_{ij}), \dots, l_P(e_{ij}))$ for edge e_{ij} , respectively, where $\mathbf{x}_q : \mathbf{V} \rightarrow \mathbb{R}$ and $l_p : \mathbf{E} \rightarrow \mathbb{R}$. These metadata can take various forms like node characteristics or edge weights. We also denote the node and edge attribute matrices $\mathbf{X} = (\mathbf{x}_i | v_i \in \mathbf{V})$ and $\mathbf{L} = (\mathbf{x}_{ij} | e_{ij} \in \mathbf{E})$, respectively.

Next, we assume that \mathcal{M} has generated a time series \mathcal{D} on \mathcal{G} . This time series takes the form of a pair of consecutive snapshots $\mathcal{D} = (\mathbf{X}, \mathbf{Y})$ with $\mathbf{X} = \mathbf{x}_1, \dots, \mathbf{x}_T$ and $\mathbf{Y} = \mathbf{y}_1, \dots, \mathbf{y}_T$, where $\mathbf{x}_t \in \mathcal{S}^{|V|}$ is the state of the nodes at time t , $\mathbf{y}_t \in \mathcal{R}^{|V|}$ is the outcome of \mathcal{M} defined as

$$\mathbf{y}_t = \mathcal{M}(\mathbf{x}_t, \mathcal{G}), \quad (1)$$

\mathcal{S} is the set of possible node states, and \mathcal{R} is the set of possible node outcomes. This way of defining \mathcal{D} allows us to formally concatenate multiple realizations of the dynamics in a single dataset. Additionally, the elements $x_i(t) = (\mathbf{x}_t)_i$ and $y_i(t) = (\mathbf{y}_t)_i$ correspond to the state of node v_i at time t and its outcome, respectively. Typically, we consider that the outcome $y_i(t)$ is simply the state of node v_i after transitioning from state $x_i(t)$. In this case, we have $\mathcal{S} = \mathcal{R}$ and $x_i(t + \Delta t) = y_i(t)$ where Δt is the length of the time steps. However, if \mathcal{S} is a discrete set—i.e. finite and countable— $y_i(t)$ is a transition probability vector conditioned on $x_i(t)$ from which the following state, $x_i(t + \Delta t)$, will be sampled. The element $(y_i(t))_m$ corresponds to the probability that node v_i evolves to state $m \in \mathcal{S}$ given that it was previously in state $x_i(t)$ —i.e. $\mathcal{R} = [0, 1]^{|\mathcal{S}|}$. When \mathcal{M} is a stochastic dynamics, we do not typically have access to the transition probabilities $y_i(t)$ directly, but rather to the observed outcome state—e.g. $x_i(t + \Delta t)$ in the event where \mathcal{X} is temporally ordered—, we therefore define the observed outcome $y_i(t)$ as

$$(y_i(t))_m = \delta(x_i(t + \Delta t), m), \quad m \in \mathcal{S} \quad (2)$$

where $\delta(x, y)$ is the Kronecker delta. Finally, we assume that \mathcal{M} acts on \mathbf{x}_t locally and identically at all times, according to the structure of \mathcal{G} . In other words, knowing the state x_i as well as the states of all the neighbors of v_i , the outcome y_i is computed using a time independent function f identical for all nodes

$$y_i = f(x_i, \mathbf{x}_{N_i}, \mathbf{N}_i, \mathbf{L}_{N_i}), \quad (3)$$

where $\mathbf{x}_{N_i} = (\mathbf{x}_j | v_j \in N_i)$ denotes the states of the neighbors, $N_i := \{v_j | e_{ij} \in \mathbf{E}\}$ is the set of the neighbors, $\mathbf{N}_i := \{v_j | v_j \in N_i\}$ and $\mathbf{L}_{N_i} := (\mathbf{x}_{ij} | v_j \in N_i)$. As a result, we impose a notion of locality where the underlying dynamics is time invariant and invariant under the permutation of the node labels in \mathcal{G} , under the assumption that the node and edge attributes are left invariant.

Our objective is to build a model $\hat{\mathcal{M}}$, parametrized by a GNN with a set of tunable parameters $\boldsymbol{\theta}$ and trained on the observed dataset \mathcal{D} to mimic \mathcal{M} given \mathcal{G} , such that

$$\hat{\mathcal{M}}(\mathbf{x}_t^\theta, \mathcal{G}^\theta; \boldsymbol{\theta}) \hat{=} \mathcal{M}(\mathbf{x}_t^\theta, \mathcal{G}^\theta), \quad (4)$$

for all states \mathbf{x}_t^θ and all networks \mathcal{G}^θ . The architecture of $\hat{\mathcal{M}}$, detailed in Sec. V A, is designed to act locally similarly to \mathcal{M} .

FIG. 1. **Predictions of GNN trained on a Barabási-Albert random network (BA) [41] for the (a) simple and (a) complex contagion dynamics.** The solid and dashed lines correspond to the transition probabilities of the dynamics used to generate the training data (labeled GT for “ground truth”), and predicted by the GNN, respectively. Symbols correspond to the maximum likelihood estimation (MLE) of the transition probabilities computed from the dataset \mathcal{D} .

The colors indicate the type of transition: infection ($\mathcal{S} \rightarrow \mathcal{I}$) in blue and recovery ($\mathcal{S} \rightarrow \mathcal{R}$) in red. The standard deviations, as a result of averaging the outcomes given \mathcal{D} , are shown using a colored area around the lines (typically narrower than the width of the lines) and using vertical bars for the symbols.

In this case, the locality is imposed by a modified attention mechanism inspired by Ref. [40]. The advantage of imposing locality allows our architecture to be *inductive*: If the GNN is trained on a wide range of local structures—i.e. nodes with different neighborhood sizes (or degrees) and states—it can then be used on any other networks within that range. This suggests that the topology of \mathcal{G} will have a strong impact on the quality of the trained models, an intuition that is confirmed below. Similarly to Eq. (3), we can write each individual node outcome computed by the GNN using a function f such that

$$y_i = f(x_i, \mathbf{x}_{N_i}, \mathbf{N}_i, \mathbf{L}_{N_i}; \boldsymbol{\theta}) \quad (5)$$

where y_i is the outcome of node v_i predicted by $\hat{\mathcal{M}}$.

The objective described by Eq. (4) must be encoded into a global loss function, denoted $\mathcal{L}(\boldsymbol{\theta})$. Like the outcome functions, $\mathcal{L}(\boldsymbol{\theta})$ can be decomposed locally, where the local losses of each node $\mathcal{L}(y_i, \hat{y}_i)$ are arithmetically averaged over all possible node inputs $(x_i, \mathbf{x}_{N_i}, \mathbf{N}_i, \mathbf{L}_{N_i})$, where y_i and \hat{y}_i are given by Eqs. (3) and (5), respectively. By using an arithmetic mean to the evaluation of $\mathcal{L}(\boldsymbol{\theta})$, we assume that the node inputs are distributed uniformly. Consequently, the model should be trained equally well on all of them. This is important because in practice we only have access to a finite number of inputs in \mathcal{D} and \mathcal{G} , for which the node input distribution is typically far from being uniform. Hence, in order to train effective models, we recalibrate the inputs using the following global loss

$$\mathcal{L}(\boldsymbol{\theta}) = \frac{1}{Z^0} \sum_{t=0}^T \sum_{v_i \in \mathcal{V}} w_i(t) \mathcal{L}(y_i, \hat{y}_i)$$

where $w_i(t)$ is a weight assigned to node v_i at time t , and $Z^0 = \sum_{t=0}^T \sum_{v_i \in \mathcal{V}} V_0(t) w_i(t)$ is a normalization factor. Here,

FIG. 2. **Comparison between the targets and the predictions of GNN trained on Erdős-Rényi networks (ER, top row) and on Barabási-Albert networks [41] (BA, middle row) for the (a, e, i) simple, (b, f, j) complex, (c, g, k) interacting and (d, h, l) metapopulation dynamics.** Each point shown on the panels (a–h) corresponds to a different pair $(y_i(t), \hat{y}_i(t))$ in the complete dataset D . We also indicate the Pearson coefficient r on each panel to measure the correlation between the predictions and the targets and use it as a global performance measure. The panels (i–l) show the errors $(1 - r)$ as a function of the number of neighbors for GNN trained on ER and BA networks, and those of the corresponding MLE. These errors are obtained from the Pearson coefficients computed from subsets of the prediction-target pairs where all nodes have degree k .

the training node set $V^0(t) \subset V$ and the training time set $T^0 \subset [1, T]$ allow us to partition the training dataset for validation and testing when required.

The choice of weights needs to reflect the importance of each node at each time. Because we wish to lower the influence of overrepresented inputs and increase that of rare inputs, a sound choice of weights is

$$\downarrow w(v) \propto k^{-\alpha}, \quad \alpha \in [0, 1] \quad (7)$$

where k is the degree of node v in G , and $0 < \alpha < 1$ is a hyperparameter. Equation (7) is an ideal choice, because it corresponds to a principled importance sampling approximation of Eq. (6) [42], which is relaxed via the exponent α . We obtain a pure importance sampling scheme when $\alpha = 1$. Note that the weights can rarely be exactly computed using Eq. (7), because the distribution $\propto k^{-\alpha}$ is typically computationally intensive to obtain from data, especially for continuous S with metadata. We illustrate various ways to evaluate the weights in Sec. V B and in Supplementary Material.

We now illustrate the accuracy of our approach by applying it to four types of synthetic dynamics of various natures (see Sec. V C for details on the dynamics). We first consider

a *simple contagion* dynamics: The discrete-time susceptible-infected-susceptible (SIS) dynamics. In this dynamics, nodes are either susceptible (S) or infected (I) by some disease, i.e. $S = \{S, I\} = \{0, 1\}$, and transition between each state stochastically according to an infection probability function $\phi(\tilde{\nu})$, where $\tilde{\nu}$ is the number of infected neighbors of a node, and a constant recovery probability μ . A notable feature of simple contagion dynamics is that susceptible nodes get infected by the disease through their infected neighbors independently. This reflects the assumption that disease transmission behaves identically whether a person has a large number of infected neighbors or not.

Second, we relax this assumption by considering a *complex contagion* dynamics with a nonmonotonic infection function $\phi(\tilde{\nu})$ where the aforementioned transmission events are no longer independent [14]. This contagion dynamics has an interesting interpretation in the context of the propagation of a social behavior, where the local popularity of a behavior (large $\tilde{\nu}$) hinders its adoption. The independent transmission assumption can also be lifted when multiple diseases are interacting [10]. Thus, we also consider an asymmetric *interacting contagion* dynamics with two diseases. In this case, $S = \{S_1 S_2, I_1 S_2, S_1 I_2, I_1 I_2\} = \{0, 1, 2, 3\}$ where $u \nu$ cor-

FIG. 3. **Bifurcation diagrams of the (a) simple, (b) complex, (c) interacting and (d) metapopulation dynamics on Poisson networks [41] composed of $|V| = 2000$ nodes with different average degrees $\langle k \rangle$.** The prevalence is defined as the average fraction of nodes that are asymptotically infected by at least one disease and the outbreak size corresponds to the average fraction of nodes that have recovered. These quantities are obtained from numerical simulations using the “ground truth” (GT) dynamics (blue circles) and the GNN trained on Barabási-Albert networks (orange triangles). The error bars correspond to the standard deviations of these numerical simulations. The trained GNN used are the same ones as those used for Fig. 2. As a reference, we also indicate with dashed lines the value(s) of average degree $\langle k \rangle$ corresponding to the network(s) on which the GNN were trained. On panel (d), more than one value of $\langle k \rangle$ appear as multiple networks with different average degrees were used to train the GNN.

responds to a state where a node is in state U with respect to the first disease and in state V with respect to the second disease. The interaction between the diseases happens via a coupling that is active only when a node is infected by at least one disease, otherwise it behaves identically to the simple contagion dynamics. This coupling may increase or decrease the virulence of the other disease.

Whereas the previously presented dynamics capture various features of contagion phenomena, real datasets containing this level of detail about the interactions among individuals are rare [43–45]. A class of dynamics for which dataset are easier to find is that of mass-action *metapopulation* dynamics [46–49], where the status of the individuals are gathered by geographical regions. These dynamics typically evolve on the weighted networks of the individuals’ mobility between regions and the state of a region consists in the number of people that are in each individual health state. As a fourth case study, we consider a type of deterministic metapopulation dynamics where the population size is constant and where people can either be susceptible (S), infected (I) or recovered from the disease (R). As a result, we define the state of the node as three-dimensional vectors specifying the fraction of people in each state—i.e. $\mathbf{S} = \mathbf{R} = [0, 1]^3$.

Figure 1 shows the GNN predictions for the infection and recovery probabilities of the simple and complex contagion dynamics as a function of the number of infected neighbors ξ . We then compare them with their ground truths, i.e. Eq. (20) using Eqs. (18)–(22) for the infection functions. We also show the maximum likelihood estimators (MLE) of the transition probabilities computed from the fraction of nodes in state x and with ξ infected neighbors that transitioned to state y in the complete dataset D. The MLE, which are typically used in this kind of inference problem [52], stands as a reference to benchmark the performance of our approach.

We find that the GNN learns remarkably well the transition probabilities of the simple and complex contagion dynamics. In fact, the predictions of the GNN seem to be systematically

smoother than the ones provided by the MLE. This is because the MLE is computed for each individual pair (x, ξ) from disjoint subsets of the training dataset. This implies that a large number of samples of each pair (x, ξ) is needed for the MLE to be accurate; a condition rarely met in realistic settings, especially for high degree nodes. This also means that the MLE cannot be used directly to interpolate beyond the pairs (x, ξ) present in the training dataset, in sharp contrast with the GNN which, by definition, can interpolate within the dataset D. Furthermore, all of its parameters are hierarchically involved during training, meaning that the GNN benefits from any sample to improve all of its predictions, which are then smoother and more consistent.

It is worth mentioning that the GNN is not specifically designed nor trained to compute the transition probabilities as a function of a single variable, namely the number of infected ξ . In reality, the GNN computes its outcome from the complete multivariate state of the neighbors of a node. The interacting contagion and the metapopulation dynamics, unlike the simple and complex contagions, are examples of such multivariate cases. Their outcome is thus harder to visualize in a representation similar to Fig. 1. Figures 2(a–h) address this issue by comparing each of the GNN predictions $\hat{y}_i(t)$ with its corresponding target $y_i(t)$ in the dataset D. We quantify the global performance of the models in different scenarios, for the different dynamics and underlying network structures, using the Pearson correlation coefficient r between the predictions and targets (see Sec. V A). We also compute the error, defined from the Pearson coefficient as $1 - r$ for each degree class k (i.e. between the predictions and targets of only the nodes of degree k). This allows us to quantify the GNN performance for every local structure.

Figures 2(i–k) confirm that the GNN provides more accurate predictions than the MLE in general and across all degrees. This is especially true in the case of the interacting contagion, where the accuracy of the MLE seems to deteriorate rapidly for large degree nodes. This is a consequence of how

FIG. 4. **Spain COVID-19 dataset.** (a) Spain mobility multiplex network [50]. The thickness of the edges is proportional to the average number of people transitioning between all connected node pairs. The size of the nodes is proportional to the population M living in the province. (b) Time series of the incidence for the 52 provinces of Spain between January 2020 and March 2021 [51]. Each province is identified by its corresponding ISO code. Each incidence time series has been rescaled by its maximum value for the purpose of visualization. The shaded area indicates the training and validation datasets (in-sample) from January 1st 2020 to December 1st 2021. The remaining of the dataset is used for testing.

scarce the inputs are for this dynamics compared to both the simple and complex contagion dynamics for training datasets of the same size, and of how fast the size of the set of possible inputs scales, thereby quickly rendering MLE completely ineffective for small training datasets. The GNN, on the other hand, is less affected by the scarcity of the data, since any sample improves its global performance, as discussed above.

Figure 2 also exposes the crucial role of the network G on which the dynamics evolves in the global performance of the GNN. Namely, the heterogeneous degree distributions of Barabási-Albert networks (BA)—or any **heterogeneous degree distribution**—offer a wider range of degrees than those of homogeneous Erdős-Re’nyi networks (ER). We can take advantage of this heterogeneity to train GNN models that generalize well across a larger range of local structures, as seen in Fig. 2(i–l) (see also Supplementary Material). **However**, the predictions on BA networks are not systematically always better for low degrees than those on ER networks, as seen in the interacting and metapopulation cases. This nonetheless suggests a wide applicability of our approach for real complex systems, whose underlying network structures recurrently exhibit a heterogeneous degree distribution [53].

We now test the trained GNN on unseen network structures by recovering the bifurcation diagrams of the four dynamics. In the infinite-size limit $|V| \rightarrow \infty$, these dynamics have two possible long-term outcomes: the absorbing state where the diseases quickly die out, and the endemic/epidemic state in which a macroscopic fraction of nodes remains (endemic) or has been infected over time (epidemic) [4, 10, 54]. These possible long-term outcomes exchange stability during a phase transition which is continuous for the simple contagion and metapopulation dynamics, and discontinuous for the complex and interacting contagion dynamics. The position of the phase transition depends on the parameters of the dynamics as well as on the topology of the network. Note that for the interacting contagion dynamics, the stability of absorbing and endemic states do not change at the same point, giving rise to a bistable regime where both states are stable.

Figure 3 shows the different bifurcation diagrams obtained by performing numerical simulations with the trained GNN models [using Eq. (5)] while varying the average degree of networks, on which the GNN has not been trained. Quantitatively, the predictions are **again** strikingly accurate—essentially perfect for the simple and complex contagion dynamics—which is remarkable given that the bifurcation diagrams were obtained on networks the GNN had never seen before. **These results illustrate** how insights can be gained about the underlying process concerning the existence of phase transitions and their order, among other things. **They also suggest** how the GNN can be used for diverse applications, such as predicting the dynamics under various network structures (e.g. designing intervention strategies that affect the way individuals interact and are therefore connected).

Finally, we illustrate the applicability of our approach by training our GNN model using the evolution of COVID-19 in Spain between January 1st 2020 and March 27th 2021 (see Fig. 4). This dataset consists in the daily number of new cases (i.e. incidence) for each of the 50 provinces of Spain as well as Ceuta and Melilla [51]. We also use a network of the mobility flow recorded in 2018 [50] as a proxy to model the interaction network between these 52 regions. This network is multiplex—each layer corresponds to a different mode of transportation—, directed and weighted (average daily mobility flow).

We compare the performance of our approach with that of different baselines: Four data-driven techniques—three competing neural network architectures [37, 55] and a linear vector autoregressive model (VAR) [56, 57]—, and an equivalent mechanistic metapopulation model (Metapop) driven by a simple contagion mechanism [58]. Among the three neural network architectures, we used the model of Ref. [37] (KP-GNN) that has been used to predict the evolution of COVID-19 in the US. The other two GNN architectures embody the assumptions that the nodes of the networks are mutually independent (IND), or that the nodes are arbitrarily codependent (FC) in a way that is learned by the neural network. Finally,

FIG. 5. **Learning the Spain COVID-19 dataset.** (a-b) Comparison between the targets and the predictions in the in-sample and the out-of-sample datasets for our GNN model (blue) and for other models (KP-CNN in orange, IND in pink, FC in purple and VAR in green; see main text). The accuracy of the predictions is quantified by the Pearson correlation coefficient provided in the legend. (c) Forecasts by our GNN model for individual time series of the provincial daily incidence compared with the ground truth. Underestimation and overestimation are respectively shown in blue and red. Each time series has been rescaled as in Fig. 4(b) and are ordered according to mean square error of the GNN’s predictions. (d) Forecasts for the global incidence (sum of the daily incidence in every province). The solid grey line indicates the ground truth (GT); the dashed blue line, the dashed orange line and dotted green line show the forecast of our GNN model, of KP-CNN and of VAR, respectively. We also show the forecast of an equivalent metapopulation model (red dash-dotted line) which has its own scale (red axis on the right) to improve the visualization; the other lines share the same axis on the left. Similarly to Fig. 4, we differentiate the in-sample from the out-of-sample forecasts using a shaded background.

we used the parameters of Ref. [59] for the metapopulation model. Section V D provides more details on the baselines.

Figure 5 shows that all data-driven models can generate highly accurate in-sample predictions, with the exception of the KP-GNN model which appears to have a hard time leaning the dynamics, possibly because of its aggregation mechanism (see Sec. III F of the Supplementary Material). However, the other architectures do not appear to have the same capability to generalize the dynamics out-of-sample: The FC and the VAR models, especially, tend to overfit more than the GNN and the IND models. While this was expected for the linear VAR model, the FC model overfits because it is granted too much freedom in the way it learns how the nodes interact with one another. Interestingly, the IND and the GNN models seem to perform similarly, which hints to the possibility that the specifics of the mobility network might not have contributed significantly to the dynamics. This is perhaps not surprising since social distancing and confinement measures were in place during the period covered by the dataset. Indeed, our results indicate that the global effective dynamics was mostly driven by internal processes within each individual province, rather than driven by the interaction between them. This last observation suggests that our GNN model is robust to spurious connections in the interaction network.

Finally, Fig. 5(d) shows that the metapopulation model is systematically overestimating the incidence by over an order of magnitude. Again, this is likely due to the confinement measures in place during that period which were not re-

flected in the original parameters of the dynamics [59]. Additional mechanisms accounting for this interplay between social restrictions and the prevalence of the disease—e.g. complex contagion mechanisms [12] or time-dependent parameters [60]—would therefore be in order to extend the validity of the metapopulation model to the full length of the dataset. Interestingly a signature of this interplay is encoded in the daily incidence data and our GNN model appears to be able to capture it to some extent.

III. DISCUSSION

We introduced a data-driven approach that learns effective mechanisms governing the propagation of diverse dynamics on complex networks. We proposed a reliable training protocol, and we validated the projections of our GNN architecture on simple, complex, interacting contagion and metapopulation dynamics using synthetic networks. Interestingly, we found that our approach performs better when trained on data whose underlying network structure is heterogeneous, which could prove useful in real-world applications of our method given the ubiquitousness of scale-free networks [61].

By recovering the bifurcation diagram of various dynamics, we illustrated how our approach can leverage time series from an unknown dynamical process to gain insights about its properties—e.g. the existence of a phase transition and its order. We have also shown how to use this framework on real

datasets, which in turn could then be used to help build better effective models. In a way, we see this approach as the equivalent of a numerical Petri dish—offering a new way to experiment and gain insights about an unknown dynamics—that is complementary to traditional mechanistic modeling to design better intervention procedures, containment countermeasures and to perform model selection.

Although we focused the presentation of our method on contagion dynamics, its potential applicability reaches many other realms of complex systems modeling where intricate mechanisms are at play. We believe this work establishes solid foundations for the use of deep learning in the design of realistic effective models of complex systems.

Gathering detailed epidemiological datasets is a complex and labor-intensive process, meaning that datasets suitable for our approach are currently the exception rather than the norm. The current COVID-19 pandemic has, however, shown how an adequate international reaction to an emerging infectious pathogen critically depends on the free flow of information. New initiatives like Global.health [62] are good examples on how the international epidemiological community is coming together to share data more openly and to make available comprehensive datasets to all researchers. Thanks to such initiatives, it is likely that future pandemics will see larger amount of data available to the scientific community in real time. It is therefore crucial for the community to start developing tools, such as the one presented here, to leverage these datasets so that we are ready for the next pandemic.

IV. ACKNOWLEDGEMENTS AND SUPPORT

The authors are grateful to Emily N. Cyr and Guillaume St-Onge for their comments and to Vincent Thibeault, Xavier Roy-Pomerleau, François Thibault, Patrick Desrosiers, Louis J. Dube', Simon Hardy and Laurent He'bert-Dufresne for fruitful discussions. They also want to thank the anonymous reviewers for their insightful comments which led to substantial improvements of the manuscript. This work was supported by the Sentinelle Nord initiative from the Fonds d'excellence en recherche Apogée Canada (CM, EL, AA), the Conseil de recherches en sciences naturelles et en gé'nie du Canada (CM, AA) and the Fonds de recherche du Que'bec-Nature et tech-nologie (EL).

V. MATERIAL AND METHODS

A. Graph neural network and training details

In this section, we briefly present our GNN architecture, the training settings, the synthetic data generation procedure and the hyperparameters used in our experiments.

1. Architecture

We use the GNN architecture shown in Fig. 6 and detailed in Tab. I. First, we transform the state x_i of every node with a shared multilayer perception (MLP), denoted $f_{in}: S \rightarrow \mathbb{R}^d$ where d is the resulting number of node features, such that

$$\leftarrow i = f_{in}(x_i). \quad (8)$$

We concatenate the node attributes \square_i to x_i , when these attributes are available, in which case $f_{in}: S \times \mathbb{R}^Q \rightarrow \mathbb{R}^d$. At this point, $\leftarrow i$ is a vector of features representing the state (and attributes) of node v_i . Then, we aggregate the features of the first neighbors using a modified attention mechanism f_{att} , inspired by Ref. [40] (see Sec. V A 2),

$$\otimes_i = f_{att}(\leftarrow i, \leftarrow N_i), \quad (9)$$

where we recall that $N_i = \{v_j | E_{ij} \in E\}$ is the set of nodes connected to node v_i . We also include the edge attributes \otimes_{ij} into the attention mechanism, when they are available. To do so, we transform the edge attributes \otimes_{ij} into abstract edge features, such that $\otimes_{ij} = f_{edge}(\otimes_{ij})$ where $f_{edge}: \mathbb{R}^p \rightarrow \mathbb{R}^{d_{edge}}$ is also a MLP, before they are used in the aggregation. Finally, we compute the outcome \hat{y}_i of each node v_i with another MLP $f_{out}: \mathbb{R}^d \rightarrow \mathbb{Z}$ such that

$$\hat{y}_i = f_{out}(\otimes_i). \quad (10)$$

2. Attention Mechanism

We use an attention mechanism inspired by the graph attention network architecture (GAT) [40]. The attention mechanism consists of three trainable functions $A: \mathbb{R}^d \rightarrow \mathbb{R}$, $B: \mathbb{R}^d \rightarrow \mathbb{R}$ and $C: \mathbb{R}^{d_{edge}} \rightarrow \mathbb{R}$, that combine the feature vectors $\leftarrow i$, $\leftarrow j$ and \otimes_{ij} of a connected pair of nodes v_i and v_j , where we recall that d and d_{edge} are the number of node and edge features, respectively. Then, the attention coefficient a_{ij} is computed as follows

$$a_{ij} = \frac{A(\leftarrow i) + B(\leftarrow j) + C(\otimes_{ij})}{\sum_{k \in N_i} A(\leftarrow k) + B(\leftarrow k) + C(\otimes_{ik})} \quad (11)$$

where $\sigma(x) = [1 + e^{-x}]^{-1}$ is the logistic function. Notice that, by using this logistic function, the value of the attention coefficients is constrained to the open interval $(0, 1)$, where $a_{ij} = 0$ implies that the feature $\leftarrow j$ do not change the value of \otimes_i , and $a_{ij} = 1$ implies that it maximally changes the value of \otimes_i . In principle, a_{ij} quantifies the influence of the state of node v_j over the outcome of node v_i . In reality, the representation learned by the GNN can be non-sparse, meaning that the neighbor features $\leftarrow N_i$ can be combined in such a way that their noncontributing parts are canceled out without having a_{ij} being necessarily zero. This can result in the failure of this interpretation of this attention coefficients (see the Supplementary material for further details). Nevertheless, the attention coefficients can be used to assess how connected nodes interact together.

FIG. 6. **Visualization of the GNN architecture.** The blocks of different colors represent mathematical operations. The red blocks correspond to trainable affine transformation parametrized by weights and biases. The purple blocks represent activation functions between each layer. The core of the model is the attention module [40], which is represented in blue. The orange block at the end is an exponential Softmax activation that transforms the output into properly normalized outcomes.

Dynamics	Simple	Complex	Interacting	Metapopulation	COVID-19
Input layers	Linear(1, 32) ReLU Linear(32, 32) ReLU	Linear(1, 32) ReLU Linear(32, 32) ReLU	Linear(1, 32) ReLU Linear(32, 32) ReLU Linear(32, 32) ReLU	Linear(4, 32)* ReLU Linear(32, 32) ReLU Linear(32, 32) ReLU Linear(32, 32) ReLU	RNN(2, 4; L)* ReLU RNN(4, 8; L) ReLU RNN(8, 16; L)** ReLU Linear(16, 32) ReLU
Number of attention layers	2	2	4	8†	5††
Output layers	Linear(32, 32) ReLU Linear(32, 2) Softmax	Linear(32, 32) ReLU Linear(32, 2) Softmax	Linear(32, 32) ReLU Linear(32, 32) ReLU Linear(32, 4) Softmax	Linear(32, 32) ReLU Linear(32, 32) ReLU Linear(32, 32) ReLU Linear(32, 3) Softmax	Linear(32, 16)* ReLU Linear(16, 8) ReLU Linear(8, 4) ReLU Linear(4, 1)
Number of parameters	6 698	6 698	11 188	99 883	7 190

TABLE I. **Layer by layer description of the GNN models for each dynamics.** For each sequence, the operations are applied from top to bottom. The operations represented by Linear(m, n) correspond to linear (or affine) transformations of the form $f(\mathbf{x}) = W\mathbf{x} + \mathbf{b}$, where $\mathbf{x} \in \mathbb{R}^m$ is the input, $W \in \mathbb{R}^{n \times m}$ and $\mathbf{b} \in \mathbb{R}^n$ are trainable parameters. The operation RNN($m, n; L$) corresponds to an Elman recurrent neural network module [63] with m input features and n output features applied on sequences of length L [63]. The operations ReLU and Softmax are activation functions given by $\text{ReLU}(x) = \max\{x, 0\}$ and $\text{Softmax}(\mathbf{x}) = \frac{\exp(x_i)}{\sum_j \exp(x_j)}$.

increased by one, because we aggregated to the state of the nodes x_i their rescaled and centered population size N_i . (**) Here, only the features of the last element of the sequence—those corresponding to the state X_t —are kept to proceed further into the architecture. (†) Because the networks are weighted for the metapopulation dynamics, we initially transform the edge weights into abstract feature representations using a sequence of layers, i.e. (Linear(1, 4), ReLU, Linear(4, 4)) applied from left to right, before using them in the attention modules. These layers are trained alongside all the other layers. (††) The network is also weighted in this case, hence we used the same set up as for the metapopulation GNN model to transform the edge weights. Also, note that the five attention modules are each associated to a different layer in the multiplex network.

We compute the aggregated feature vectors $\langle \mathcal{X} \rangle$ of node v as

$$\langle \mathcal{X} \rangle = f_{att}(\leftarrow \cdot, \leftarrow N_i) = \leftarrow \cdot + \chi_{v/2} \frac{\partial^{ij} \leftarrow \cdot}{N_i} \cdot j. \quad (12)$$

It is important to stress that, at this point, $\langle \mathcal{X} \rangle$ contains some information about v and all of its neighbors in a pairwise manner. In all our experiments, we fix \mathbf{A} , \mathbf{B} , and \mathbf{C} to be affine transformations with trainable weight matrix and bias vector. Also, we use multiple attention modules in parallel to increase

the expressive power of the GNN architecture, as suggested by Ref. [40].

The attention mechanism described by Eq. (11) is slightly different from the vanilla version of Ref. [40]. Similarly to other well-known GNN architectures [33, 64, 65], the aggregation scheme of the vanilla GAT is designed as an average of the feature vectors of the neighbors—where, by definition, $\sum_{v_j \in N_{v_i}} a_{ij} = 1$ for all v_i —rather than as a general weighted sum like for Eq. (12). This is often reasonable in the context of structure learning, where the node features represent some coordinates in a metric space where connected nodes are likely to be close [33]. Yet, in the general case, this type of constraint was shown to lessen dramatically the expressive power of the GNN architecture [31]. We also reached the same conclusion while using average-like GNN architectures (see the Supplementary Material). By contrast, the aggregation scheme described by Eq. (12) allows our architecture to represent various dynamic processes **on networks accurately**.

3. Training settings

In all experiments **on synthetic data**, we use the cross entropy loss as the local loss function,

$$L(y_i, \hat{y}_i) = -\sum_m y_{i,m} \log \hat{y}_{i,m}, \quad (13)$$

where $y_{i,m}$ corresponds to the m -th element of the outcome vector of node v_i , which either is a transition probability for the stochastic contagion dynamics or a fraction of people for the metapopulation dynamics. For the simple, complex and interacting contagion dynamics, we used the observed outcomes \tilde{y}_i , corresponding to the stochastic state of node v_i at the next time step, as the target in the loss function. While we noticed a diminished performance when using the observed outcomes as opposed to the true transition probabilities (see Supplementary Material), this setting is more realistic and shows what happens when the targets are noisy. The effect of noise can be tempered by increasing the size of the dataset (see the Supplementary Material). For the metapopulation dynamics, since this model is deterministic, we used the true targets without adding noise.

4. Performance measures

We use the Pearson correlation coefficient r as a global performance measure defined on a set of targets Y and predictions \hat{Y} as

$$r = \frac{\text{Cov}(Y, \hat{Y})}{\sqrt{\text{Var}(Y) \text{Var}(\hat{Y})}} = \frac{\mathbb{E}[(Y - \mathbb{E}[Y])(\hat{Y} - \mathbb{E}[\hat{Y}])]}{\sqrt{\mathbb{E}[(Y - \mathbb{E}[Y])^2] \mathbb{E}[(\hat{Y} - \mathbb{E}[\hat{Y}])^2]}} \quad (14)$$

where $\mathbb{E}[W]$ denotes the expectation of W . Also, because the maximum correlation occurs at $r = 1$, we also define $1 - r$ as the global error on the set of target-prediction pairs.

5. Synthetic data generation

We generate data from each dynamics using the following algorithm:

1. Sample a graph G from a given generative model (e.g. the Erdo's-Re'nyi $G(N, M)$ or the Baraba'si-Albert network models).
2. Initialize the state of the system $X(0) = (x_i(0))_{i=1..N}$. For the simple, complex and interacting contagion dynamics, sample uniformly the number of nodes in each state. For the metapopulation dynamics, sample the population size for each node from a Poisson distribution of average 10^4 and then sample the number of infected people within each node from a binomial distribution of parameter 10^{-5} . For instance, a network of $|V| = 10^3$ nodes will be initialized with a total of 100 infected people, on average, distributed among the nodes.
3. At time t , compute the observed outcome- Y_t for the metapopulation dynamics, and \hat{Y}_t for the three stochastic dynamics. Then, record the states X_t and Y_t .
4. Repeat step 3 until $(t \bmod t_s) = 0$, where t_s is a resampling time. At this moment, apply step 2 to reinitialize the states X_t and repeat step 3.
5. Stop when $t = T$, where T is the targeted number of samples.

The resampling step parametrized by t_s indirectly controls the diversity of the training dataset. We allow t_s to be small for the contagion dynamics ($t_s = 2$) and larger for the metapopulation dynamics ($t_s = 100$) to emphasize on the performance of the GNN rather than the quality of the training dataset, while acknowledging that different values of t_s could lead to poor training (see Supplementary Material).

We trained the simple, complex and interacting contagion GNN models on networks of size $|V| = 10^3$ nodes and on time series of length $T = 10^4$. To generate the networks, we either used Erdo's-Re'nyi (ER) random networks $G(N, M)$ or Baraba'si-Albert (BA) random networks. In both cases, the **parameters** of the generative network models are chosen such that the average degree is fixed to $\langle k \rangle = 4$.

To train our models on the metapopulation dynamics, we generated 10 networks of $|V| = 100$ nodes and generated for each of them time series of $t_s = 100$ time steps. This number of time steps roughly corresponds to the moment where the epidemic dies out. Similarly to the previous experiments, we used the ER and the BA models to generate the networks, where the parameters were chosen such that $\langle k \rangle = 4$. However, because this dynamics is not stochastic, we varied the average degree of the networks to increase the variability in the time series. This was done by randomly removing a fraction $p = 1 - \ln(1 - \mu \epsilon \mu)$ of their edges, where μ was sampled for each network uniformly between 0 and 1. In this scenario, the networks were directed and weighted, with each edge weight ω_{ij} being uniformly distributed between 0 and 1.

6. Hyperparameters

The optimization of the parameters was performed using the rectified Adam algorithm [66], which is hyperparameterized by $b_1 = 0.9$ and $b_2 = 0.999$, as suggested in Ref. [66].

To build a validation dataset, we selected a fraction of the node states randomly for each time step. More specifically, we chose node i at time t proportionally to its importance weight $m_i(t)$. For all experiments on synthetic dynamics, we randomly selected 10 nodes to be part of the validation set, on average. For all experiments, the learning rate ϵ was reduced by a factor 2 every 10 epochs with initial value $\epsilon = 0.001$. A weight decay of 10^{-4} was used as well to help regularize the training. We trained all models for 30 epochs, and selected the GNN model with the lowest loss on validation datasets. We fixed the importance sampling bias exponents for the training to $\mathcal{A} = 0.5$ in the simple, complex and interacting contagion cases, and fixed it to $\mathcal{A} = 1$ in the metapopulation case.

B. Importance weights

In this section, we show how to implement the importance weights in the different cases. Other versions of the importance weights are also available in the Supplementary Material.

1. Discrete state stochastic dynamics

When S is a finite countable set, the importance weights can be computed exactly using Eq. (7),

$$m_i(t) \propto p(k_i, x_i(t), x_{\mathcal{N}_i}(t))^{1-\mathcal{A}} \quad (15)$$

where $p(k, x, x_{\mathcal{N}})$ is the probability to observe a node of degree k in state x with a neighborhood in state $x_{\mathcal{N}}$ in the complete dataset D . The inputs can be simplified from $(k, x, x_{\mathcal{N}})$ to (k, x, \mathbf{t}) without loss of generality, where \mathbf{t} is a vector whose entries are the number of neighbors in each state. The distribution is then estimated from the complete dataset D by computing the fraction of inputs that are in every configuration

$$p(k, x, \mathbf{t}) = \frac{1}{|\mathcal{M}|} \prod_{i=1}^M I_{k_i = k} \times \prod_{\mathbf{t}=1}^{\text{ET}} I_{x_i(t) = x, I_{\mathbf{t}_i(t)} = \mathbf{t}} \quad (16)$$

where $I(\cdot)$ is the indicator function.

2. Continuous state deterministic dynamics

The case of continuous states-e.g. for metapopulation dynamics-is more challenging than its discrete counterpart,

especially if the node and edge attributes, \square_i and \boxtimes_j , need to be accounted for. One of the challenges is that we cannot count the inputs like in the previous case. As a result, the estimated distribution p cannot be estimated directly using Eq. (16), and we use instead

$$m_i(t) = [P(k_i) \wedge (\square_i, \boxtimes_j | k_i) \hat{u}(x_i^-(t))]^{1-\mathcal{A}} \quad (17)$$

where $P(k_i)$ is the fraction of nodes with degree k_i , $\wedge(\square_i, \boxtimes_j | k_i)$ is the joint probability density function (pdf) the sum of the edge attributes \boxtimes_j conditioned on the degree k_i for the node attributes \square_i and $v_j \in \mathcal{N}_i$, and where $\hat{u}(x_i^-(t))$ is the pdf for the average of node states at time t $\bar{x}(t) = \frac{1}{|\mathcal{M}|} \sum_{v \in \mathcal{N}_i} x_v(t)$. The pdf are obtained using non-parametric Gaussian kernel density estimators (KDE) [67]. Provided that the density values of the KDE are unbounded above, we normalize the pdf such that the density of each sample used to construct the KDE sum to one. Further details on how we designed the importance weights are provided in the Supplementary Material.

C. Dynamics

In what follows, we describe in detail the contagion dynamics used for our experiments. We specify the node outcome function f introduced in Eq. (3) and the parameters of the dynamics.

1. Simple contagion

We consider the simple contagion dynamics called the susceptible-infected-susceptible (SIS) dynamics for which $S = \{S, I\} = \{0, 1\}$ -we use these two representations of S interchangeably. Because this dynamics is stochastic, we let $R = [0, 1]^2$. We define the infection function $\omega(t)$ as the probability that a susceptible node becomes infected given its number of infected neighbors t

$$\Pr(S \rightarrow I | t) = \omega(t) = 1 - (1 - \gamma)^t, \quad (18)$$

where $\gamma \in [0, 1]$ is the disease transmission probability. In other words, a node can be infected by any of its infected neighbors independently with probability γ . We also define the constant recovery probability as

$$\Pr(I \rightarrow S) = \theta. \quad (19)$$

The node outcome function for the SIS dynamics is therefore

$$f(x_i, x_{\mathcal{N}_i}) = \begin{cases} (1 - \omega(t), \omega(t)) & \text{if } x_i = 0, \\ (0, 1 - \theta) & \text{if } x_i = 1, \end{cases} \quad (20)$$

where

$$F_{t, \theta} = \delta(x_i, 1) \quad (21)$$

is the number of infected neighbors of v_i and $\delta(x, y)$ is the Kronecker delta. Note that for each case in Eq. (20), the outcome is a two-dimensional probability vector, where the first entry is the probability that node v_i becomes/remains susceptible at the following time step, and the second entry is the probability that it becomes/remains infected. We used $(\beta_1, \beta_2) = (0.04, 0.08)$ in all experiments involving this simple contagion dynamics.

2. Complex contagion

To lift the independent transmission assumption of the SIS dynamics, we consider a complex contagion dynamics for which the node outcome function has a similar form as Eq. (20), but where the infection function $\phi(\cdot)$ has the non-monotonic form

$$\phi(\cdot) = \frac{1}{z(\mathcal{H})} e^{i/n - 1} \quad (22)$$

where $z(\mathcal{H})$ normalizes the infection function such that $\phi(\cdot) = 1$ at its global maximum \cdot^* and $\mathcal{H} > 0$ is a parameter controlling the position of \cdot^* . This function is inspired by the Planck distribution for the black-body radiation, although it was chosen for its general shape rather than for any physical meaning whatsoever. We used $(\mathcal{H}, \beta) = (8, 0.06)$ in all

experiments involving this complex contagion dynamics.

3. Interacting contagion

We define the interacting contagion as two SIS dynamics that are interacting and denote it as the SIS-SIS dynamics. In this case, we have $\mathbf{S} = \{S_1 S_2, I_1 S_2, S_1 I_2, I_1 I_2\} = \{0, 1, 2, 3\}$. Similarly to the SIS dynamics, we have $\mathbf{R} = [0, 1]^4$ and we define the infection probability functions

$$\phi_g(\cdot) = 1 - (1 - \beta_g)^{i_g} \quad \text{if } x = 0 \quad (23a)$$

$$\phi^* g(\cdot) = 1 - (1 - \beta_g)^{i_g} \quad \text{if } x = 1, 2, \quad (23b)$$

where $\beta_g > 0$ is a coupling constant and i_g is the number of neighbors infected by disease g , and also define the recovery probabilities β_g for each disease ($g = 1, 2$). The case where $\beta > 1$ corresponds to the situation in which the diseases are synergistic (i.e. being infected by one increases the probability of getting infected by the other), whereas competition is introduced if $\beta < 1$ (being already infected by one decreases the probability of getting infected by the other). The case $\beta = 1$ falls back on two independent SIS dynamics that evolve simultaneously on the network. The outcome function is composed of 16 entries that are expressed as follows

$$f(x_i, x_{Ar_i}) = \begin{cases} \beta [1 - \beta_1(i,1)][1 - \beta_2(i,2)], \beta_1(i,1)[1 - \beta_2(i,2)], [1 - \beta_1(i,1)] \beta_2(i,2), \beta_1(i,1) \beta_2(i,2) & \text{if } x_i = 0, \\ \beta_1 [1 - \beta_2^*(i,2)], [1 - \beta_1][1 - \beta_2^*(i,2)], \beta_1 \beta_2^*(i,2), [1 - \beta_1] \beta_2^*(i,2) & \text{if } x_i = 1, \\ \beta [1 - \beta_1^*(i,1)] \beta_2, \beta_1^*(i,1) \beta_2, [1 - \beta_1^*(i,1)] [1 - \beta_2], \beta_1^*(i,1) [1 - \beta_2] & \text{if } x_i = 2, \\ \beta_1 \beta_2, [1 - \beta_1] \beta_2, \beta_1 [1 - \beta_2], [1 - \beta_1] [1 - \beta_2] & \text{if } x_i = 3. \end{cases} \quad (24)$$

where we define i_g as the number of neighbors of v_i that are infected by disease g . We used $(\beta_1, \beta_2, \beta_1, \beta_2, \beta) = (0.01, 0.012, 0.19, 0.22, 50)$ in all experiments involving this interacting contagion dynamics.

4. Metapopulation

The metapopulation dynamics considered is a deterministic version of the susceptible-infection-recovered (SIR) metapopulation model [46-49]. We consider that the nodes are populated by a fixed number of people N_i , which can be in three states-susceptible (S), infected (I) or recovered (R). We therefore track the number of people in every state at each time. Furthermore, we let the network G be weighted, with the weights describing the mobility flow of people between regions. In this case, $Q_{ij} \in \mathbb{R}$ is the average number of people that are traveling from node v_j to node v_i . Finally, because

we assume that the population size is on average steady, we let $\beta_i = N_i$ be a node attribute and work with the fraction of people in every epidemiological state. More precisely, we define the state of node v_j by $x_j = (s_j, i_j, r_j)$, where s_j, i_j and r_j are the fractions of susceptible, infected and recovered people, respectively. From these definitions, we define the node outcome function of this dynamics as

$$f(x_j, x_{Ar_j}, G) = \begin{pmatrix} 0 \\ \beta \end{pmatrix} \begin{matrix} s_j - s_j \phi^j \\ i_j - \frac{i_j}{T_r + s_j} \phi^j \\ r_j + \frac{i_j}{T_r} \phi^j \end{matrix} \quad (25)$$

where

$$\phi^j = \phi(i_j, N_j) + \frac{k_j Q_{ij} \phi(i_l, N_l)}{\sum_{v_l \in Ar_j} Q_{jl} N_l},$$

and k_j is the degree of node v_j . The function $\phi(i, N)$ corresponds to the infection rate, per day, at which an individual is

infected by someone visiting from a neighboring region with iN infected people in it, and is equal to

$$ce(i, N) = 1 - \frac{1 - R_0}{T_r N} \left(1 - e^{-\frac{R_0}{T_r} i} \right) \quad (27)$$

where R_0 corresponds to the reproduction number and, T_r is the average recovery time in days. In all experiments with this metapopulation dynamics, we used $(R_0, T_r) = (8.31, 7.5)$.

D. COVID-19 outbreak in Spain

1. Dataset

The dataset is composed of the daily incidence of the 52 Spanish provinces (including Ceuta and Melilla) monitored for 450 days between January 1st 2020 and March 27th 2021 [51]. The dataset is augmented with the origin-destination (OD) network of individual mobility [50]. This mobility network is multiplex, directed and weighted, where the weight of each edge e_{ij}^v represents mobility flow from province V_j and to province V_i using transportation v . The metadata associated to each node is the population of province V_i [68], noted $4^i = N_i$. The metadata associated to each edge, \mathbb{X}_{ij}^v , corresponds to the average number of people that moved from V_j to V_i using v as the main means of transportation.

2. Models

The GNN model used in Fig. 5 is very similar to the metapopulation GNN model-with node and edge attributes-, with the exception that different attention modules are used to model the different OD edge types (plane, car, coach, train and boat, see Tab. I). To combine the features of each layer of the multiplex network, we average pooled the output features of the attention modules. We also generalize our model to take in input a sequence of L states of the system, that is

$$\hat{Y}_t = \hat{M}(X_{t:t-L+1}, G; 0) \quad (28)$$

where $X_{t:t-L+1} = (X_t, X_{t-1}, \dots, x_{t-L+1})$ and L is a lag. At the local level, it reads

$$\left(\hat{y}_i(t) = f \hat{x}_i(t : t - L + 1), \right.$$

$$\left. 4^i, xN(t : t - L + 1), 4^N, \mathbb{X}^N; 0 \right) \quad (29)$$

where $x_i(t : t - L + 1)$ corresponds to the L previous state of node i from time t to time $t-L+1$. As we now feed sequences of node states to the GNN, we use Elman recurrent neural networks [63] to transform these sequences of states before aggregating them instead of linear layers, as shown in Fig. 6 and Tab. I. Additionally, because the outputs of the models are not probability vectors, like for the dynamics of Sec. V C,

but real numbers, we use the mean square error (MSE) loss to train the model:

$$L(y_i, \hat{y}_i) = (y_i - \hat{y}_i)^2. \quad (30)$$

We use five different baseline models to compare with the performance of our GNN: Three additional neural network architectures, a vector autoregressive model (VAR) [56] and an equivalent metapopulation model driven by a simple contagion mechanism. The first neural network architecture, denoted the KP-GNN model, was used in Ref. [37] to forecast the evolution COVID-19 in the US using a similar strategy as ours with respect to the mobility network. As described in Ref. [37], we used a single-layered MLP with 64 hidden units to transform the input, and then we used two graph convolutional networks (GCN) in series, each with 32 hidden units, to perform the feature aggregation. Finally, we computed the output of the model using another single-layered MLP with 32 hidden units. The layers of this model are separated by ReLU activation functions and are sampled with a dropout rate of 0.5. Because this model is not directly adapted to multiplex networks, we merged all layers together into a single network and summed the weights of the edges. Then, as prescribed in Ref. [37], we thresholded the merged network by keeping at most 32 neighbors with the highest edge weight for each node. We did not use our importance sampling procedure to train the KP-GNN model-letting $A = 0$ -to remain as close as possible to the original model.

3.

The other two neural network architectures are very similar to the GNN model we presented in Tab. I: The only different component is their aggregation mechanism. The IND model, where the nodes are assumed to be mutually independent, does not aggregate the features of the neighbors. It therefore acts like a univariate model, where the time series of each node are processed like different elements of a minibatch. In the FC model, the nodes interact via a single-layered MLP connecting all nodes together. The parameters of this MLP are learnable, which effectively allows the model to express any interaction patterns. Because the number of parameters of this MLP scales with $d|V|^2$, where d is the number of node features after the input layers, we introduce another layer of 8 hidden units to compress the input features before aggregating.

The VAR model is a linear generative model adapted for multivariate time series forecasting

$$\hat{Y}_t = \hat{M}(X_t, X_{t-1}, \dots, X_{t-L+1}) = \sum_{l=0}^{L-1} A_l X_{t-l} + b + \epsilon_t \quad (31)$$

where $A_l \in \mathbb{R}^{|V| \times |V|}$ are weight matrices, $b \in \mathbb{R}^{|V|}$ is a trend vector and ϵ_t is an error term with $E[\epsilon_t] = 0$ and $E[\epsilon_t \epsilon_s] = \delta_{t,s} E$, with E being a positive-semidefinite covariance matrix.

While autoregressive models are often used to predict stock markets [69], they have also been used recently

to forecast diverse COVID-19 outbreaks [57]. This model is fitted to the COVID-19 time series dataset also by minimizing the MSE.

The metapopulation model is essentially identical to the model presented in Sec. V C. However, because we track the incidence, i.e. the number of newly infectious cases $\square_i(t)$ in each province i , instead of the complete state (S_i, I_i, R_i) representing the number of individuals in each state, we allow the model to internally track the complete state based on the ground truth. At first, the whole population is susceptible, i.e. $S_i(1) = N_i$. Then, at each time step, we subtract the number of newly infectious cases in each node from $S_i(t)$, and add to I_i . Finally, the model allows a fraction \square_{Hofits} infected people to recover. The evolution equations of this model are as follows

$$S_i(t+1) = S_i(t) - \square_i(t), \quad (32a)$$

$$I_i(t+1) = I_i(t) + \square_i(t) - \frac{1}{\square_{\text{Hofits}}} I_i(t), \quad (32b)$$

$$R_i(t+1) = R_i(t) + I_i(t) \frac{1}{\square_{\text{Hofits}}}. \quad (32c)$$

Finally, we computed the incidence $\square_i(t)$ predicted by the metapopulation model using the current internal state as follows:

$$\square_i(t) = S_i \leftarrow \tilde{i}, \quad (33)$$

where $\leftarrow \tilde{i}$ is given by Eq. (26), using the mobility network G , Eq. (27) for $\leftarrow (i, N)$ and $ij = \lfloor Nj$. Since the

weights \square_{ij} represent the average number of people traveling from province v_j to province v_i , we assumed all layers to be equivalent and aggregated each layer into a single typeless network where $\square_{ij} = P \square_{ij}$. We fixed the parameters of the model to $R_0 = 2.5$ and $\square_{\text{Hofits}} = 7.5$, as these values were used in other contexts for modeling the propagation of COVID-19 in Spain [59].

3. Training

We trained the GNN and other neural networks for 200 epochs, while decreasing the learning rate by a factor of 2 every 20 epochs with an initial value of 10^{-3} . For our GNN, the IND and the FC models, we fixed the importance sampling bias exponent to $\square = 0.5$ and, like the models trained on synthetic data, we used a weight decay of 10^{-4} (see Sec. V A 3). We fixed the lag of these models, including the VAR model, to $L = 5$. The KP-GNN model was trained using a weight decay of 10^{-5} following Ref. [37], and we chose a lag of $L = 7$. For all models, we constructed the validation dataset by randomly selecting a partition of the nodes at each time step proportionally to their importance weights $w_i(t)$: 20% of the nodes are used for validation in this case. The test dataset was constructed by selecting the last 100 time steps of the time series of all nodes, which rough corresponds to the third wave of the outbreak in Spain.

-
- [1] W. O. Kermack and A. G. McKendrick, "A Contribution to the Mathematical Theory of Epidemics," Proc. R. Soc. A 115, 700–721 (1927).
- [2] H. W. Hethcote, "The Mathematics of Infectious Diseases," SIAM Rev. 42, 599–653 (2000).
- [3] C. I. Siettos and L. Russo, "Mathematical modeling of infectious disease dynamics," Virulence 4, 295–306 (2013).
- [4] I. Z. Kiss, J. C. Miller, and P. L. Simon, Mathematics of Epidemics on Networks (Springer, 2017) p. 598.
- [5] F. Brauer, C. Castillo-Chavez, and Z. Feng, Mathematical Models in Epidemiology (Springer, 2019).
- [6] N. C. Grassly and C. Fraser, "Mathematical models of infectious disease transmission," Nat. Rev. Microbiol. 6, 477–487 (2008).
- [7] A. Pastore y Piontti, N. Perra, L. Rossi, N. Samay, and A. Vespignani, Charting the Next Pandemic: Modeling Infectious Disease Spreading in the Data Science Age (Springer, 2019).
- [8] C. Viboud and A. Vespignani, "The future of influenza forecasts," Proc. Natl. Acad. Sci. U.S.A. 116, 2802–2804 (2019).
- [9] D. M. Morens, J. K. Taubenberger, and A. S. Fauci, "Predominant Role of Bacterial Pneumonia as a Cause of Death in Pandemic Influenza: Implications for Pandemic Influenza Preparedness," J. Infect. Dis. 198, 962–970 (2008).
- [10] J. Sanz, C.-Y. Xia, S. Meloni, and Y. Moreno, "Dynamics of Interacting Diseases," Phys. Rev. X 4, 041005 (2014).
- [0] S. Nickbakhsh, C. Mair, L. Matthews, R. Reeve, P. C. D. Johnson, F. Thorburn, B. von Wissmann, A. Reynolds, J. McMenamin, R. N. Gunson, and P. R. Murcia, "Virus–virus interactions impact the population dynamics of influenza and the common cold," Proc. Natl. Acad. Sci. U.S.A. 116, 27142–27150 (2019).
- [1] L. Hébert-Dufresne, S. V. Scarpino, and J.-G. Young, "Macroscopic patterns of interacting contagions are indistinguishable from social reinforcement," Nat. Phys. 16, 426–431 (2020).
- [2] D. Centola, "The Spread of Behavior in an Online Social Network Experiment," Science 329, 1194–1197 (2010).
- [3] S. Lehmann and Y.-Y. Ahn, eds., Complex Spreading Phenomena in Social Systems, Computational Social Sciences (Springer, 2018).
- [4] I. Iacopini, G. Petri, A. Barrat, and V. Latora, "Simplicial models of social contagion," Nat. Commun. 10, 2485 (2019).
- [5] M. Biggerstaff, M. Johansson, D. Alper, L. C. Brooks, P. Chakraborty, D. C. Farrow, S. Hyun, S. Kandula, C. McGowan, N. Ramakrishnan, R. Rosenfeld, J. Shaman, R. Tibshirani, R. J. Tibshirani, A. Vespignani, W. Yang, Q. Zhang, and C. Reed, "Results from the second year of a collaborative effort to forecast influenza seasons in the United States," Epidemics 24, 26–33 (2018).
- [6] S. L. Brunton, J. L. Proctor, and J. N. Kutz, "Discovering governing equations from data by sparse identification of nonlinear dynamical systems," Proc. Natl. Acad. Sci. USA 113, 3932–3937 (2016).

- [18] J. N. Kutz, “Deep learning in fluid dynamics,” *J. Fluid Mech.* **814**, 1–4 (2017).
- [19] J. Pathak, Z. Lu, B. R. Hunt, M. Girvan, and E. Ott, “Using machine learning to replicate chaotic attractors and calculate Lyapunov exponents from data,” *Chaos* **27**, 121102 (2017).
- [20] B. Lusch, J. N. Kutz, and S. L. Brunton, “Deep learning for universal linear embeddings of nonlinear dynamics,” *Nat. Com-mun.* **9**, 1–10 (2018).
- [21] J. Pathak, B. Hunt, M. Girvan, Z. Lu, and E. Ott, “Model-Free Prediction of Large Spatiotemporally Chaotic Systems from Data: A Reservoir Computing Approach,” *Phys. Rev. Lett.* **120**, 024102 (2018).
- [22] B. M. de Silva, D. M. Higdon, S. L. Brunton, and J. N. Kutz, “Discovery of Physics From Data: Universal Laws and Discrepancies,” *Front. Artif. Intell.* **3**, 25 (2020).
- [23] X. Chen, T. Weng, H. Yang, C. Gu, J. Zhang, and M. Small, “Mapping topological characteristics of dynamical systems into neural networks: A reservoir computing approach,” *Phys. Rev. E* **102**, 033314 (2020).
- [24] R. Dutta, A. Mira, and J.-P. Onnela, “Bayesian inference of spreading processes on networks,” *Proc. R. Soc. A* **474**, 20180129 (2018).
- [25] C. Shah, N. Dehmamy, N. Perra, M. Chinazzi, A.-L. Barabási, A. Vespignani, and R. Yu, “Finding Patient Zero: Learning Contagion Source with Graph Neural Networks,” (2020), [arXiv:2006.11913](https://arxiv.org/abs/2006.11913).
- [26] F. A. Rodrigues, T. Peron, C. Connaughton, J. Kurths, and Y. Moreno, “A machine learning approach to predicting dynamical observables from network structure,” (2019), [arXiv:1910.00544](https://arxiv.org/abs/1910.00544).
- [27] A. Salova, J. Emenheiser, A. Rupe, J. P. Crutchfield, and R. M. D’Souza, “Koopman operator and its approximations for systems with symmetries,” *Chaos* **29**, 093128 (2019).
- [28] E. Laurence, C. Murphy, G. St-Onge, X. Roy-Pomerleau, and V. Thibeault, “Detecting structural perturbations from time series with deep learning,” (2020), [arXiv:2006.05232](https://arxiv.org/abs/2006.05232).
- [29] Z. Zhang, P. Cui, and W. Zhu, “Deep Learning on Graphs: A Survey,” (2018), [arXiv:1812.04202](https://arxiv.org/abs/1812.04202).
- [30] J. Zhou, Gé Cui, Z. Zhang, C. Yang, Z. Liu, L. Wang, C. Li, and M. Sun, “Graph Neural Networks: A Review of Methods and Applications,” (2018), [arXiv:1812.08434](https://arxiv.org/abs/1812.08434).
- [31] K. Xu, W. Hu, J. Leskovec, and S. Jegelka, “How Powerful are Graph Neural Networks?” (2018), [arXiv:1810.00826](https://arxiv.org/abs/1810.00826).
- [32] B. Perozzi, R. Al-Rfou, and S. Skiena, “DeepWalk: Online Learning of Social Representations,” *Proc. ACM SIGKDD Int. Conf. Knowl. Discov. Data Min.* , 701–710 (2014), [arXiv:1403.6652](https://arxiv.org/abs/1403.6652).
- [33] W. L. Hamilton, R. Ying, and J. Leskovec, “Representation Learning on Graphs: Methods and Applications,” (2017), [arXiv:1709.05584](https://arxiv.org/abs/1709.05584).
- [34] Z. Zhang, Y. Zhao, J. Liu, S. Wang, R. Tao, R. Xin, and J. Zhang, “A general deep learning framework for network reconstruction and dynamics learning,” *Appl. Netw. Sci.* **4**, 110 (2019).
- [35] A. Fout, J. Byrd, B. Shariat, and A. Ben-Hur, “Protein Interface Prediction using Graph Convolutional Networks,” in *Adv. Neural Inf. Process. Syst.* **30** (2017) pp. 6530–6539.
- [36] M. Zitnik, M. Agrawal, and J. Leskovec, “Modeling polypharmacy side effects with graph convolutional networks,” in *Bioinformatics*, Vol. 34 (2018) pp. i457–i466.
- [37] A. Kapoor, X. Ben, L. Liu, B. Perozzi, M. Barnes, M. Blais, and S. O’Banion, “Examining covid-19 forecasting using spatio-temporal graph neural networks,” (2020), [arXiv:2007.03113](https://arxiv.org/abs/2007.03113).
- [38] C. Fritz, E. Dorigatti, and D. Rügamer, “Combining graph neural networks and spatio-temporal disease models to predict covid-19 cases in germany,” (2021), [arXiv:2101.00661](https://arxiv.org/abs/2101.00661).
- [39] J. Gao, R. Sharma, C. Qian, L. M Glass, J. Spaeder, J. Romberg, J. Sun, and C. Xiao, “STAN: spatio-temporal attention network for pandemic prediction using real-world evidence,” *J. Am. Med. Inform. Assoc.* **28**, 733–743 (2021).
- [40] P. Velickovic, G. Cucurull, A. Casanova, A. Romero, P. Lio, and Y. Bengio, “Graph Attention Networks,” *arXiv* , 1710.10903 (2017).
- [41] A.-L. Barabási, *Network Science* (Cambridge University Press, 2016) p. 474.
- [42] R. Y. Rubinstein and D. P. Kroese, *Simulation and the Monte Carlo Method*, 3rd ed. (Wiley, 2016) p. 414.
- [43] L. Isella, J. Stehlé, A. Barrat, C. Cattuto, J.-F. Pinton, and W. Van den Broeck, “What’s in a crowd? Analysis of face-to-face behavioral networks,” *J. Theor. Biol.* **271**, 166–180 (2011).
- [44] R. Mastrandrea, J. Fourmet, and A. Barrat, “Contact Patterns in a High School: A Comparison between Data Collected Using Wearable Sensors, Contact Diaries and Friendship Surveys,” *PLOS ONE* **10**, e0136497 (2015).
- [45] M. Géniois and A. Barrat, “Can co-location be used as a proxy for face-to-face contacts?” *EPJ Data Sci.* **7**, 11 (2018).
- [46] V. Colizza, R. Pastor-Satorras, and A. Vespignani, “Reaction–diffusion processes and metapopulation models in heterogeneous networks,” *Nat. Phys.* **3**, 276–282 (2007).
- [47] D. Balcan, B. Gonçalves, H. Hu, J. J. Ramasco, V. Colizza, and A. Vespignani, “Modeling the spatial spread of infectious diseases: The global epidemic and mobility computational model,” *J. Comput. Sci.* **1**, 132–145 (2010).
- [48] M. Ajelli, Q. Zhang, K. Sun, S. Merler, L. Fumanelli, G. Chow-ell, L. Simonsen, C. Viboud, and A. Vespignani, “The RAPIDD Ebola forecasting challenge: Model description and synthetic data generation,” *Epidemics* **22**, 3–12 (2018).
- [49] D. Soriano-Panós, L. Lotero, A. Arenas, and J. Gómez-Gardenes, “Spreading Processes in Multiplex Metapopulations Containing Different Mobility Networks,” *Phys. Rev. X* **8**, 031039 (2018).
- [50] “Observatorio del Transporte y la Logística en España,” <https://observatoriotransporte.mitma.gob.es/estudio-experimental> (2018), [Accessed: 11-August-2020].
- [51] “COVID-19 en España,” <https://cneccovid.isciii.es> (2020), [Accessed: 27-March-2021].
- [52] P. Eichelsbacher and A. Ganesh, “Bayesian Inference for Markov Chains,” *J. Appl. Probab.* **39**, 91–99 (2002).
- [53] I. Voitalov, P. van der Hoorn, R. van der Hofstad, and D. Krioukov, “Scale-free networks well done,” *Phys. Rev. Research* **1**, 033034 (2019).
- [54] R. Pastor-Satorras, C. Castellano, P. Van Mieghem, and A. Vespignani, “Epidemic processes in complex networks,” *Rev. Mod. Phys.* **87**, 925–979 (2015).
- [55] Y. Tian, Ish. Luthra, and . Zhang, “Forecasting covid-19 cases using machine learning models,” (2020), [10.1101/2020.07.02.20145474](https://arxiv.org/abs/2014.07.02.20145474).
- [56] W. W. S. Wei, *Multivariate time series analysis and applications* (John Wiley & Sons, 2018).
- [57] F. Rustam, A. A. Reshi, A. Mehmood, S. Ullah, B. On, W. Aslam, and G. S. Choi, “Covid-19 future forecasting using supervised machine learning models,” *IEEE Access* **8**, 101489–101499 (2020).
- [58] V. Colizza, A. Barrat, M. Barthelemy, A.-J. Valleron, and A. Vespignani, “Modeling the Worldwide Spread of Pandemic Influenza: Baseline Case and Containment Interventions,”

- PLOS Med. 4, e13 (2007).
- [59] A. Aleta and Y. Moreno, “Evaluation of the potential incidence of COVID-19 and effectiveness of containment measures in Spain: A data-driven approach,” BMC Med. 18, 157 (2020).
- [60] Q. Wang, S. Xie, Y. Wang, and D. Zeng, “Survival-Convolution Models for Predicting COVID-19 Cases and Assessing Effects of Mitigation Strategies,” Front. Public Health 8, 325 (2020).
- [61] I. Voitalov, P. van der Hoorn, R. van der Hofstad, and D. Krioukov, “Scale-free networks well done,” Phys. Rev. Research 1, 033034 (2019).
- [62] “Global.health: A Data Science Initiative,” <https://global.health> (2020), [Accessed: 21-May-2021].
- [63] I. Goodfellow, Y. Bengio, and A. Courville, *Deep Learning* (MIT Press, 2016).
- [64] T. N. Kipf and M. Welling, “Semi-Supervised Classification with Graph Convolutional Networks,” (2016), arXiv:1609.02907.
- [65] C. Morris, M. Ritzert, M. Fey, W. L. Hamilton, J. E. Lenssen, G. Rattan, and M. Grohe, “Weisfeiler and Leman Go Neural: Higher-order Graph Neural Networks,” (2018), arXiv:1810.02244.
- [66] L. Liu, H. Jiang, P. He, W. Chen, X. Liu, J. Gao, and J. Han, “On the Variance of the Adaptive Learning Rate and Beyond,” (2019), arXiv:1908.03265.
- [67] W. J. Conover, *Practical nonparametric statistics* (John Wiley & Sons, 1998) p. 350.
- [68] “Instituto Nacional de Estadística,” <https://www.ine.es> (2020), [Accessed: 11-August-2020].
- [69] C. A. Sims, “Macroeconomics and reality,” Econometrica, 1–48 (1980).

Reviewers' Comments:

Reviewer #1:

Remarks to the Author:

The main contributions of this paper have not changed in the revision, for which my summary of the paper is the same as the last review, and I further recognize the novelty in the importance sampling procedure. This paper proposes a complementary approach based on Graph Neural Networks to automatically learn the stochastic dynamics from time series data on complex networks. By predicting the local transition probabilities of each node on various states, the authors provide a data-driven approach to learn the effective mechanisms governing the propagation process of contagion. The accuracy of the proposed method is demonstrated through a lot of experiments. On generated networks from classical patterns such as Barabasi-Albert random network, the proposed method successfully fits the ground truth. On real-world temporal networks, the proposed method recovers the bifurcation diagram of different contagion dynamics based on epidemic models well. The experiment settings are rigorous and the results are convincing.

In the revision, the authors spent substantial efforts and I appreciate their works in improving their manuscript and responding to my concerns. Compared to the paper before the revision, the authors added a GNN-based baseline in the experiments and included the performance of each baseline to discuss the improvements from the proposed framework in a way of ablation study. In the rebuttal, the authors also gave a satisfying illustration on their novelty compared with other GNN models that focus on dynamic networks.

I still have one suggestion about the writing. In Reply 1.4 of the rebuttal part, the authors demonstrate the advantages of the proposed model compared with other GNN models which can also deal with dynamic networks, but this part of demonstration is not well presented in the manuscript. I believe that the potential readers of this article may be interested in GNN and have the same concern as me that why the former GNN methods cannot handle contagion dynamics in this work. Therefore, I suggest that the authors might clarify this in the introduction part or other proper position, which might help to highlight the contributions of his article.

Reviewer #3:

Remarks to the Author:

I thank sincerely the authors for their effort in improving the paper. I will recommend accepting the paper as:

- I am happy with their inclusion of the discussion about interpretability and alternative thoughts - excellent works. I know I have insisted a lot to have this and I hope the authors do see the point of this exploration, to provide the readers a proper perspective and not to be carried away and always keep in mind the drawback etc.

- Regarding the data analysis I know my comments were slightly vague as I have not provided a specific recipe to follow. But given already many other improvements done in the paper, I myself actually is happy with the data analysis already they have.

Overall I would say finally the paper have become an excellent piece of work, I will strongly recommend it for publication.

Response to Referee 1

Comment 1.1

The main contributions of this paper have not changed in the revision, for which my summary of the paper is the same as the last review, and I further recognize the novelty in the importance sampling procedure. This paper proposes a complementary approach based on Graph Neural Networks to automatically learn the stochastic dynamics from time series data on complex networks. By predicting the local transition probabilities of each node on various states, the authors provide a data-driven approach to learn the effective mechanisms governing the propagation process of contagion. The accuracy of the proposed method is demonstrated through a lot of experiments. On generated networks from classical patterns such as Barabasi-Albert random network, the proposed method successfully fits the ground truth. On real-world temporal networks, the proposed method recovers the bifurcation diagram of different contagion dynamics based on epidemic models well. The experiment settings are rigorous and the results are convincing.

In the revision, the authors spent substantial efforts and I appreciate their works in improving their manuscript and responding to my concerns. Compared to the paper before the revision, the authors added a GNN-based baseline in the experiments and included the performance of each baseline to discuss the improvements from the proposed framework in a way of ablation study. In the rebuttal, the authors also gave a satisfying illustration on their novelty compared with other GNN models that focus on dynamic networks.

Reply 1.1

We would like to thank the Reviewer once again for their thorough assessment of our work throughout this review process. Their comments and suggestions have truly helped us to improve our article significantly.

Comment 1.2

I still have one suggestion about the writing. In Reply 1.4 of the rebuttal part, the authors demonstrate the advantages of the proposed model compared with other GNN models which can also deal with dynamic networks, but this part of demonstration is not well presented in the manuscript. I believe that the potential readers of this article may be interested in GNN and have the same concern as me that why the former GNN methods cannot handle contagion dynamics in this work. Therefore, I suggest that the authors might clarify this in the introduction part or other proper position, which might help to highlight the contributions of his article.

Reply 1.2

We thank the Reviewer for this suggestion.

Action taken 1.2

The part of Reply 1.4 in our previous answer document about the models the Reviewer mentions has been adapted and added to the Supplementary Information material and is now referred to in the main text.

Response to Referee 3

Comment 3.1

I thank sincerely the authors for their effort in improving the paper. I will recommend accepting the paper as:

- I am happy with their inclusion of the discussion about interpretability and alternative thoughts - excellent works. I know I have insisted a lot to have this and I hope the authors do see the point of this exploration, to provide the readers a proper perspective and not to be carried away and always keep in mind the drawback etc.

- Regarding the data analysis I know my comments were slightly vague as I have not provided a specific recipe to follow. But given already many other improvements done in the paper, I myself actually is happy with the data analysis already they have.

Overall I would say finally the paper have become an excellent piece of work, I will strongly recommend it for publication.

Reply 3.1

We would like to thank the Reviewer for their help throughout this review process.